# DAISI: Data Assimilation with Inverse Sampling using Stochastic Interpolants

**Martin Andrae** [* 1]  **Erik Wikingsson** [* 1]  **So Takao** [* 2 3]  **Tomas Landelius** [1 4]  **Fredrik Lindsten** [1]

## Abstract

Data assimilation (DA) is a cornerstone of scientific and engineering applications, combining model forecasts with sparse and noisy observations to estimate latent system states. Classical high-dimensional DA methods, such as the ensemble Kalman filter, rely on Gaussian approximations that are violated for complex dynamics or observation operators. To address this limitation, we introduce DAISI, a scalable filtering algorithm built on flow-based generative models that enables flexible probabilistic inference using data-driven priors. The core idea is to use a stationary, pre-trained generative prior that first incorporates forecast information through a novel *inverse-sampling step*, before assimilating observations via guidance-based conditional sampling. This allows us to leverage any forecasting model as part of the DA pipeline without having to retrain or fine-tune the generative prior at each assimilation step. Experiments on challenging nonlinear systems show that DAISI achieves accurate filtering results in regimes with sparse, noisy, and nonlinear observations where traditional methods struggle. The code for DAISI is available at https://github.com/Erik-Wikingsson/DAISI

## 1. Introduction

Estimating the evolving state of a complex dynamical system is a fundamental challenge in science and engineering. In many real-world problems—such as weather forecasting, fluid mechanics, neuroscience and robotics—the states of interest are only partially observed and subject to multiple sources of uncertainty (Asch et al., 2016). DA, and in particular, *filtering*, seeks to combine imperfect model forecasts with sparse, noisy observations to reconstruct these hidden states. Mathematically, this corresponds to estimating the *filtering distribution* $p(\boldsymbol{x}_n|\boldsymbol{y}_{1:n})$ where $\boldsymbol{x}_n$ denotes the latent state and $\boldsymbol{y}_{1:n} := (\boldsymbol{y}_1, \ldots, \boldsymbol{y}_n)$, where $\boldsymbol{y}_n$ is the observation at time $n$ modeled via the likelihood $p(\boldsymbol{y}_n|\boldsymbol{x}_n)$.

Weather forecasting provides a clear illustration of the challenges present in DA. The atmosphere is chaotic, high-dimensional, and observed through nonlinear, noisy measurements. Accurate state estimation is essential for applications ranging from early warning systems and agriculture to renewable energy management (NOAA NCEI, 2025; Whitt & Gordon, 2023; IPCC, 2023; Andrade & Bessa, 2017). To meet these demands, classical DA methods, such as the Ensemble Kalman Filter (EnKF), or variational methods such as 4DVar, have been developed over decades of DA research. However, these methods have respective drawbacks; EnKF is only guaranteed to work under near-Gaussian settings (Calvello et al., 2024), and the inflation and localization parameters necessary to stabilize the filter are notoriously challenging to tune (Bannister, 2017). 4DVar, on the other hand, requires the tedious development of an adjoint model and cannot quantify uncertainty, due to it being a MAP estimation method. These methods have also been shown to struggle when applied to ML-based forecasting models (Tian et al., 2024). Particle filters (e.g., Naesseth et al. (2019)) can, in principle, solve the filtering problem, but suffers from the curse of dimensionality (Bengtsson et al., 2008). These limitations motivate the development of more flexible, data-driven approaches to high-dimensional DA that can leverage non-Gaussian priors without suffering from the same curse of dimensionality as particle filters.

Recent progress in offline inverse problems—such as those in medical imaging, astrophysics, and computer vision—has shown that flow- and diffusion-based generative models can serve as powerful priors for conditional sampling (see, e.g., Zheng et al. (2025); Chung et al. (2025); Zhao et al. (2025)). However, extending these ideas to sequential filtering presents unique challenges. One possibility is to learn a generative prior for the predictive distribution $p(\boldsymbol{x}_n|\boldsymbol{y}_{1:n-1})$, but this would require retraining the model at every time step $n$ (Bao et al., 2024b), making it impractical for oper-

*Equal contribution  [1]Division of Statistics and Machine Learning, Linköping University, Linköping, Sweden  [2]California Institute of Technology, Pasadena, USA  [3]PhysicsX, New York, USA  [4]Swedish Meteorological and Hydrological Institute, Norrköping, Sweden. Correspondence to: Martin Andrae <martin.andrae@liu.se>, Erik Wikingsson <erik.wikingsson@gmail.com>, So Takao <so.takao@physicsx.ai>.

*Proceedings of the 43rd International Conference on Machine Learning*, Seoul, South Korea. PMLR 306, 2026. Copyright 2026 by the author(s).

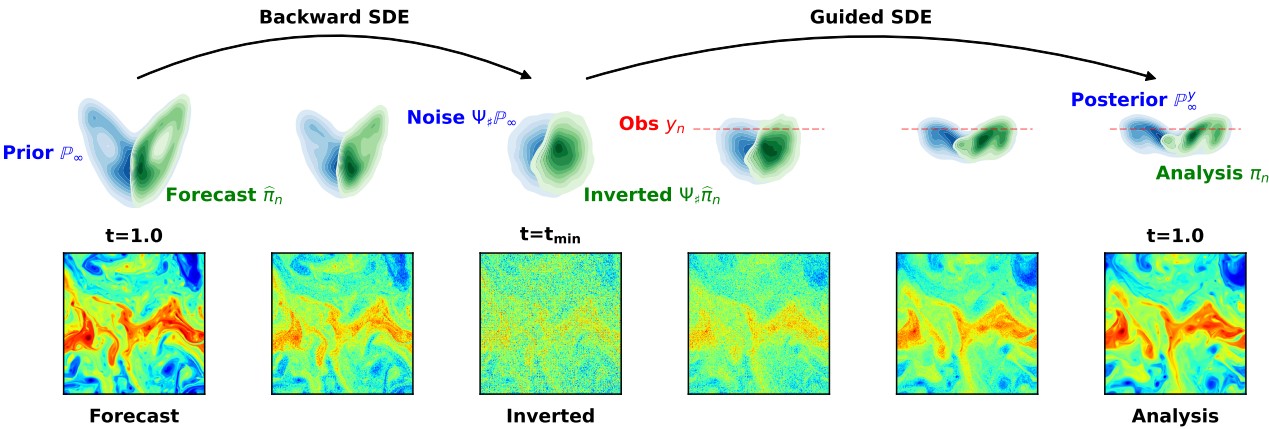

*Figure 1.* DAISI combines a flow-based unconditional prior $\mathbb{P}_\infty$ with the forecast ensemble $\hat{\pi}_n$. To condition on an observation $\boldsymbol{y}_n$, one could apply the guided SDE starting from random noise, producing $\mathbb{P}_\infty^{\boldsymbol{y}}$ (blue). However, this would ignore the information contained in the forecast. Instead, DAISI (green): (i) applies the backward SDE to the forecast ensemble, producing inverted samples $\Psi_\sharp \hat{\pi}_n$, and (ii) uses these latents as initial conditions for the guided SDE, generating approximate samples from the filtering distribution $\pi_n$.

ational use. Another strategy is to use a pre-trained generative forecast model to guide towards observations (Chen et al., 2025; Savary et al., 2026), though such models require the use of specialized dynamical models. Alternatively, approaches based on approximating the smoothing distribution $p(\boldsymbol{x}_{0:n}|\boldsymbol{y}_{1:n})$ (Rozet & Louppe, 2023) exist. However, they are memory-intensive and scale poorly with the number of time steps. We elaborate on how these approaches relate to our proposed method in Section 3.3.

### 1.1. Contributions

We present DAISI (**D**ata **A**ssimilation with **I**nverse sampling using **S**tochastic **I**nterpolants), a scalable filtering algorithm built on flow-based generative models. DAISI avoids retraining at each assimilation step by leveraging a *stationary* pre-trained generative prior to condition on new observations via guidance. To integrate dynamical information, DAISI couples this generative prior with a forecast model that advances an ensemble of states in time. Following the forecast, an *inverse-sampling* step runs the generative SDE backward from the forecast ensemble, mapping forecasted states to latent variables. These latent states then serve as initial conditions for conditional sampling under the learned prior. The full procedure is illustrated in Figure 1.

In summary, the strength and novelty of DAISI lie in its following capabilities:

(i) **Zero-shot compatibility** with both numerical and ML-based forecast and observation models.

(ii) **Modular design**, supporting any flow-based generative model and gradient-based guidance method.

(iii) **Expressive uncertainty quantification**, capturing

complex, multimodal, high-dimensional posteriors under sparse, noisy, and nonlinear observations.

## 2. Preliminaries

### 2.1. Data Assimilation

Consider the state-space model

$$\boldsymbol{x}_n = \mathcal{F}(\boldsymbol{x}_{n-1}, \boldsymbol{\omega}_n), \quad \boldsymbol{\omega}_n \sim p(\boldsymbol{\omega}) \tag{1}$$

$$\boldsymbol{y}_n = \mathcal{H}(\boldsymbol{x}_n) + \boldsymbol{\nu}_n, \quad \boldsymbol{\nu}_n \sim p(\boldsymbol{\nu}) \tag{2}$$

defined for $n = 1, \ldots, N$ and $\boldsymbol{x}_0 \sim p(\boldsymbol{x}_0)$. Here, $\mathcal{F}: \mathcal{X} \times \Omega \to \mathcal{X}$ and $\mathcal{H}: \mathcal{X} \to \mathcal{Y}$ are the dynamics and observation operators, respectively, and $\boldsymbol{\omega}_n \in \Omega, \boldsymbol{\nu}_n \in \mathcal{Y}$ are stochastic noise sources. Our general formulation of the state evolution (1) encompasses various settings such as a deterministic physics-based simulator (in which case $\boldsymbol{\omega}_n \equiv 0$) or a pre-trained generative model (in which case $\boldsymbol{\omega}_n$ corresponds to the driving noise process of the model).

DA seeks to infer the latent states $\{\boldsymbol{x}_n\}_{n=0}^N$ from the noisy observations $\{\boldsymbol{y}_n\}_{n=1}^N$. In particular, *filtering* refers to the problem of estimating the distribution $p(\boldsymbol{x}_n|\boldsymbol{y}_{1:n})$, while *smoothing* targets the full posterior $p(\boldsymbol{x}_{0:N}|\boldsymbol{y}_{1:N})$. In this work, we focus on the filtering problem, generally solved by alternating between a *forecast* and *analysis* step. Denoting by $\pi_n(\mathrm{d}\boldsymbol{x}_n) := p(\boldsymbol{x}_n|\boldsymbol{y}_{1:n})\mathrm{d}\boldsymbol{x}_n$, the measure corresponding to the filtering distribution, this proceeds abstractly as

$$\text{(Forecast)} \quad \hat{\pi}_n(\mathrm{d}\boldsymbol{x}_n) := \mathbb{E}_{\boldsymbol{\omega}_n}\left[\mathcal{F}(\cdot, \boldsymbol{\omega}_n)_\sharp \pi_{n-1}\right](\mathrm{d}\boldsymbol{x}_n) \tag{3}$$

$$\text{(Analysis)} \quad \pi_n(\mathrm{d}\boldsymbol{x}_n) \propto p(\boldsymbol{y}_n|\boldsymbol{x}_n)\hat{\pi}_n(\mathrm{d}\boldsymbol{x}_n), \tag{4}$$

where $\mathcal{F}_\sharp \pi$ denotes the pushforward of the measure $\pi$ with respect to a map $\mathcal{F}$, defined by $\mathcal{F}_\sharp \pi := \mathrm{Law}(\mathcal{F}(X))$ for $X \sim \pi$. Starting from $\pi_0$, repeated applications of (3)–(4)

yields the sequence $\pi_1, \ldots, \pi_N$. Although this recursion is well defined, computing these measures exactly is typically infeasible in practice. Our aim is therefore to develop a practical and scalable approximation to the cycle (3)–(4), capable of operating reliably in high-dimensional settings with sparse, nonlinear observations.

### 2.2. Stochastic Interpolants for Generative Modeling

Our proposed DA method leverages recent advances in generative modeling. In the static setting (i.e., ignoring time evolution), flow-based models enable sampling from a target distribution $\rho_1$ by learning a transport map from a simple latent distribution $\rho_0$. Such transports can be parameterized in multiple ways; in this work, we adopt the framework of stochastic interpolants (Albergo et al., 2023), though closely related formulations appear in the probability flow ODE for diffusion models (Song et al., 2021) and in flow matching (Lipman et al., 2023).

We define a *stochastic interpolant* as a stochastic process

$$\boldsymbol{z}_t = \alpha_t \boldsymbol{z}_0 + \beta_t \boldsymbol{z}_1, \quad t \in [0,1], \tag{5}$$

where the curves $\alpha, \beta \in C^2([0,1])$ satisfy the endpoint conditions $\alpha_0 = \beta_1 = 1$ and $\alpha_1 = \beta_0 = 0$. The data pair $(\boldsymbol{z}_0, \boldsymbol{z}_1)$ is sampled from a measure $\nu(\mathrm{d}\boldsymbol{z}_0, \mathrm{d}\boldsymbol{z}_1)$ whose marginals correspond to $\rho_0(\mathrm{d}\boldsymbol{z}_0)$ and $\rho_1(\mathrm{d}\boldsymbol{z}_1)$, respectively.

Let $\rho_t := \mathrm{Law}(\boldsymbol{z}_t)$ denote the distribution of $\boldsymbol{z}_t$ with density $p_t$, and define its *score* $\boldsymbol{s}(t, \boldsymbol{z}) := \nabla \log p_t(\boldsymbol{z})$ and *drift* $\boldsymbol{b}(t, \boldsymbol{z}) := \mathbb{E}[\dot{\boldsymbol{z}}_t | \boldsymbol{z}_t = \boldsymbol{z}]$. Albergo et al. (2023) show that for any non-negative $\epsilon \in C^0([0,1])$, the SDE

$$\mathrm{d}\boldsymbol{z}_t = (\boldsymbol{b}(t, \boldsymbol{z}_t) + \epsilon_t \boldsymbol{s}(t, \boldsymbol{z}_t))\mathrm{d}t + \sqrt{2\epsilon_t}\mathrm{d}W_t, \tag{6a}$$

$$\boldsymbol{z}_0 \sim \rho_0, \quad t \in [0,1], \tag{6b}$$

where $W_t$ is the Brownian motion, has the same marginal laws $\rho_t$ as the interpolant in (5). Furthermore, this is also true for the corresponding reverse-time process,

$$\mathrm{d}\boldsymbol{z}_t = (\boldsymbol{b}(t, \boldsymbol{z}_t) - \epsilon_t \boldsymbol{s}(t, \boldsymbol{z}_t))\mathrm{d}t + \sqrt{2\epsilon_t}\mathrm{d}\widehat{W}_t, \tag{7a}$$

$$\boldsymbol{z}_1 \sim \rho_1, \quad t \in [0,1], \tag{7b}$$

where $\widehat{W}_t$ is the Brownian motion in reverse time. In the important special case when $\rho_0 = \mathcal{N}(\boldsymbol{0}, \mathbf{I})$, the score $\boldsymbol{s}(t, \boldsymbol{z})$ can be explicitly related to the drift $\boldsymbol{b}(t, \boldsymbol{z})$. This enables sampling via (6) and (7) for arbitrary $\epsilon_t$ using only the drift, which can be learned from samples $(\boldsymbol{z}_0, \boldsymbol{z}_1) \sim \nu$; See Appendix A.1 for details.

### 2.3. Conditional Generation via Guidance

Given an observation $\boldsymbol{y} \sim p(\boldsymbol{y}|\boldsymbol{x})$ and a forward SDE (6), we can sample from the posterior measure $\rho_1^{\boldsymbol{y}}(\mathrm{d}\boldsymbol{x}) \propto p(\boldsymbol{y}|\boldsymbol{x})\rho_1(\mathrm{d}\boldsymbol{x})$ by solving the following forward SDE with

*guidance*:

$$\mathrm{d}\boldsymbol{z}_t = (\tilde{\boldsymbol{b}}(t, \boldsymbol{z}_t; \boldsymbol{y}) + \epsilon_t \tilde{\boldsymbol{s}}(t, \boldsymbol{z}_t; \boldsymbol{y}))\mathrm{d}t + \sqrt{2\epsilon_t}\mathrm{d}W_t, \tag{8a}$$

$$\boldsymbol{z}_0 \sim \rho_0, \quad t \in [0,1], \tag{8b}$$

where the *guided score* $\tilde{\boldsymbol{s}}$ is given by

$$\tilde{\boldsymbol{s}}(t, \boldsymbol{z}_t; \boldsymbol{y}) = \boldsymbol{s}(t, \boldsymbol{z}_t) + \nabla_{\boldsymbol{z}_t} \log p(\boldsymbol{y}|\boldsymbol{z}_t), \tag{9}$$

and the *guided drift* $\tilde{\boldsymbol{b}}$ by

$$\tilde{\boldsymbol{b}}(t, \boldsymbol{z}_t; \boldsymbol{y}) = \boldsymbol{b}(t, \boldsymbol{z}_t) + \lambda_t \nabla_{\boldsymbol{z}_t} \log p(\boldsymbol{y}|\boldsymbol{z}_t), \tag{10}$$

for $\lambda_t$ defined as in Appendix A.4.

In general, the guidance term is intractable as we usually only have access to $p(\boldsymbol{y}|\boldsymbol{z}_1)$ but not $p(\boldsymbol{y}|\boldsymbol{z}_t) = \mathbb{E}_{\boldsymbol{z}_1|\boldsymbol{z}_t}[p(\boldsymbol{y}|\boldsymbol{z}_1)]$. Various approaches exist to approximate this term, e.g. diffusion posterior sampling (DPS) (Chung et al., 2023) and moment matching posterior sampling (MMPS) (Rozet et al., 2024) (see Appendix A.5 for details and (Daras et al., 2024) for a survey on guidance methods).

## 3. Method

Our method, DAISI, performs filtering by combining the strengths of ensemble-based methods with the expressive priors offered by flow-based generative models. This is conceptually similar to the *hybrid ensemble-variational (EnVar) method* in classical DA (Hamill & Snyder, 2000; Lorenc, 2003), which blend a static climatological background covariance $\mathbf{P}$ with an "error-of-the-day" ensemble covariance to represent forecast uncertainty. This compensates for the limitations of either component alone: relying solely on a static background covariance ignores the dynamical evolution of errors, while using only a small ensemble fails to characterize high-dimensional uncertainties accurately.

DAISI builds on this philosophy but moves beyond Gaussian background covariances $\mathbf{P}$ used in hybrid EnVar by considering a full *background measure* $\mathbb{P}_\infty$, learned using flow-based generative models. In practice, we take $\mathbb{P}_\infty$ to be the invariant measure of the dynamical system (1), whose samples can be approximated from trajectories. While its existence and ergodicity are not generally guaranteed, it is a reasonable assumption for many systems, including geophysical flows. Given the learned prior $\mathbb{P}_\infty$, we can in principle assimilate the observation $\boldsymbol{y}_n$ by sampling from the posterior

$$\mathbb{P}_\infty^{\boldsymbol{y}}(\mathrm{d}\boldsymbol{x}_n) \propto p(\boldsymbol{y}_n|\boldsymbol{x}_n)\mathbb{P}_\infty(\mathrm{d}\boldsymbol{x}_n) \tag{11}$$

using guidance (Section 2.3). However, this alone neglects the dynamical evolution encoded in (1), which is essential for accurately tracking the latent states through time. This necessitates coupling $\mathbb{P}_\infty$ with the forecast ensemble, echoing the role of the ensembles in hybrid EnVar schemes.

To achieve this, we introduce *inverse sampling*, which transfers dynamical information from the ensemble forecast into the latent space of the generative model. Given forecast particles $\{\hat{\boldsymbol{x}}_n^{(j)}\}_{j=1}^J$, this proceeds by solving the unconditional backward SDE (7) from $t = 1$ to $t = t_{\min} \in (0, 1)$, using each $\hat{\boldsymbol{x}}_n^{(j)}$ as terminal conditions. The resulting latent variables $\{\boldsymbol{z}_{t_{\min},n}^{(j)}\}_{j=1}^J$ encode the forecast information in "noise space". We can then perform conditional sampling by integrating the guided forward SDE (8) from $t_{\min}$ to 1, initialized at these latent states. This produces updated particles $\{\boldsymbol{x}_n^{(j)}\}_{j=1}^J$ that are samples of $\mathbb{P}_\infty^{\boldsymbol{y}}$, while simultaneously containing information about the ensemble $\{\hat{\boldsymbol{x}}_n^{(j)}\}_{j=1}^J$ through its latent representations $\{\boldsymbol{z}_{t_{\min},n}^{(j)}\}_{j=1}^J$.

Taken altogether, DAISI performs filtering by alternating between the following *forecast* and *analysis* steps:

**Forecast.** Given particles $\{\boldsymbol{x}_{n-1}^{(j)}\}_{j=1}^J$ approximating the filtering distribution $\pi_{n-1}$, generate forecasts

$$\hat{\boldsymbol{x}}_n^{(j)} = \mathcal{F}(\boldsymbol{x}_{n-1}^{(j)}, \boldsymbol{\omega}_n^{(j)}), \quad j = 1, \ldots, J, \qquad (12)$$

for i.i.d. noise realizations $\boldsymbol{\omega}_n^{(j)} \sim p(\boldsymbol{\omega})$. This produces samples from the *predictive distribution* $\hat{\pi}_n$ (see Equation (3)).

**Analysis.** Assuming access to a flow-based generative model bridging $\rho_0 = \mathcal{N}(\boldsymbol{0}, \mathbf{I})$ to $\rho_1 = \mathbb{P}_\infty$, and given forecast particles $\{\hat{\boldsymbol{x}}_n^{(j)}\}_{j=1}^J$:

1. **Inverse sampling:** Solve the backward SDE (7) from $t = 1$ to $t_{\min}$ with terminal conditions $\hat{\boldsymbol{x}}_n^{(j)}$, to obtain the latent states $\boldsymbol{z}_{t_{\min},n}^{(j)}$.

2. **Forward guided sampling:** Solve the conditional forward SDE (8), from $t = t_{\min}$ to 1 with initial condition $\boldsymbol{z}_{t_{\min},n}^{(j)}$ to obtain updated particles $\boldsymbol{x}_n^{(j)}$.

We summarize this assimilation procedure in Algorithm 1.

### 3.1. Does DAISI sample from the filtering distribution?

For $0 < s \leq t \leq 1$ and $\epsilon \geq 0$, denote by $\Phi_{s,t}^\epsilon, \Phi_{s,t}^{\boldsymbol{y},\epsilon}$ the stochastic flows of the forward SDEs (6), (8), respectively. Likewise, let $\Psi_{t,s}^\epsilon, \Psi_{t,s}^{\boldsymbol{y},\epsilon}$ denote the corresponding backward-flow maps. We assume for simplicity that the noise coefficient in the SDEs are given by constants $\epsilon_t \equiv \epsilon$. For a fixed time step $n$ and some $t_{\min} \in (0, 1)$, we see that the analysis step in DAISI samples from the measure

$$\pi_{n,t_{\min},\epsilon}^{\text{DAISI}} := \mathbb{E}\left[(\Phi_{t_{\min},1}^{\boldsymbol{y},\epsilon} \circ \Psi_{1,t_{\min}}^\epsilon)_\sharp \hat{\pi}_n\right], \qquad (13)$$

where $\hat{\pi}_n$ denotes the predictive distribution at time $n$, and the expectation is over the Brownian motions used when integrating the backward and forward SDEs. For DAISI to

---

**Algorithm 1** DAISI filtering

1: In: Samples $\{\boldsymbol{x}_0^{(j)}\}_{j=1}^J \sim \pi_0$, observations $\{\boldsymbol{y}_n\}_{n=1}^N$
2: **for** $n = 1, \ldots, N$ **do**
3:    Forecast:
4:    **for** $j = 1, \ldots, J$ **do**
5:      Sample $\boldsymbol{\omega}_n^{(j)} \sim p(\boldsymbol{\omega})$
6:      $\hat{\boldsymbol{x}}_n^{(j)} \leftarrow \mathcal{F}(\boldsymbol{x}_{n-1}^{(j)}, \boldsymbol{\omega}_n^{(j)})$
7:    **end for**
8:    Inverse sampling:
9:    **for** $j = 1, \ldots, J$ **do**
10:      Integrate SDE (7) backward from $t = 1$ to $t_{\min}$ with terminal state $\hat{\boldsymbol{x}}_n^{(j)}$ to obtain $\boldsymbol{z}_{t_{\min},n}^{(j)}$
11:    **end for**
12:    Guided sampling:
13:    **for** $j = 1, \ldots, J$ **do**
14:      Integrate guided SDE (8) from $t_{\min}$ to 1 with initial state $\boldsymbol{z}_{t_{\min},n}^{(j)}$ and observation $\boldsymbol{y}_n$ to obtain $\boldsymbol{x}_n^{(j)}$
15:    **end for**
16: **end for**
17: Out: Updated particles $\{\boldsymbol{x}_n^{(j)}\}_{j=1}^J$ for $n = 1, \ldots, N$

---

function as a reliable filter, this measure should approximate the true filtering distribution $\pi_n$ closely. While there is no guarantee that (13) exactly matches $\pi_n$, our experiments show that we can achieve a good approximation by tuning the hyperparameters $t_{\min}$ and $\epsilon$. We therefore seek to understand why tuning these hyperparameters in particular can help to mitigate the bias.

In the following, let us assume that for all time $n$, the predictive distribution $\hat{\pi}_n$ is *absolutely continuous* with respect to the invariant measure $\mathbb{P}_\infty$, i.e., there exists a measurable density ratio $f_n : \mathbb{R}^{d_x} \to \mathbb{R}_{\geq 0}$ such that $\hat{\pi}_n(\mathrm{d}\boldsymbol{x}_n) \propto f_n(\boldsymbol{x}_n)\mathbb{P}_\infty(\mathrm{d}\boldsymbol{x}_n)$. Then, we can rewrite the filtering distribution as follows

$$\pi_n(\mathrm{d}\boldsymbol{x}_n) \overset{(4)}{\propto} p(\boldsymbol{y}_n|\boldsymbol{x}_n)\hat{\pi}_n(\mathrm{d}\boldsymbol{x}_n) \qquad (14)$$

$$\propto p(\boldsymbol{y}_n|\boldsymbol{x}_n)f_n(\boldsymbol{x}_n)\mathbb{P}_\infty(\mathrm{d}\boldsymbol{x}_n) \qquad (15)$$

$$\propto f_n(\boldsymbol{x}_n)\mathbb{P}_\infty^{\boldsymbol{y}}(\mathrm{d}\boldsymbol{x}_n). \qquad (16)$$

We now compare the DAISI analysis distribution (13) with the ideal target (16) in various limits to understand the role of each hyperparameter.

**Effect of $t_{\min}$.** For simplicity, assume that $\epsilon = 0$ and introduce the shorthands $\pi_n^{\text{DAISI}} := \pi_{n,0,0}^{\text{DAISI}}$, $\Phi := \Phi_{0,1}^0$ and $\Psi := \Psi_{1,0}^0$. Similarly, write $\Phi^{\boldsymbol{y}} := \Phi_{0,1}^{\boldsymbol{y},0}$ and $\Psi^{\boldsymbol{y}} := \Psi_{1,0}^{\boldsymbol{y},0}$. Note that in this case, we have $\Psi = \Phi^{-1}$ and $\Psi^{\boldsymbol{y}} = (\Phi^{\boldsymbol{y}})^{-1}$. To understand the effect of $t_{\min}$, we consider the limiting cases $t_{\min} \to 1$ and $t_{\min} \to 0$. By time-continuity of the flows on $[0, 1]$, the first limit $t_{\min} \to 1$ trivially yields $\pi_{n,t_{\min},0}^{\text{DAISI}} \to \hat{\pi}_n \propto f_n\mathbb{P}_\infty$. Comparing with (16), we ob-

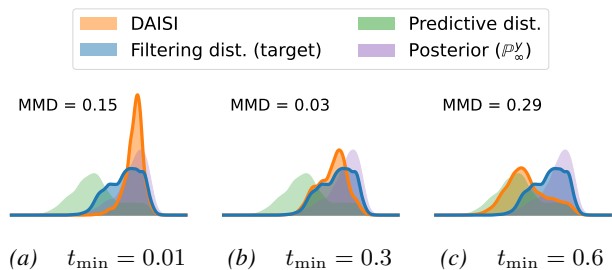

*(a)* $t_{\min} = 0.01$    *(b)* $t_{\min} = 0.3$    *(c)* $t_{\min} = 0.6$

*Figure 2.* Ablation with respect to $t_{\min}$, fixing $\epsilon = 0$. The measure $\pi_{n,t_{\min},\epsilon}^{\mathrm{DAISI}}$ (orange) is pulled towards $\hat{\pi}_n$ (green) as $t_{\min} \to 1$. There is an intermediate $t_{\min}^*$ where it matches $\pi_n$ (blue) the best.

serve that although the density ratios agree, the underlying base measures differ.

Now for the limit $t_{\min} \to 0$, first note that the inverse sampling step in DAISI produces the intermediary measure

$$\widetilde{\rho}_0(\mathrm{d}\boldsymbol{x}) := \Psi_{\sharp}\hat{\pi}_n(\mathrm{d}\boldsymbol{x}) \tag{17}$$

$$\propto f_n(\Phi(\boldsymbol{x}))\Psi_{\sharp}\mathbb{P}_{\infty}(\mathrm{d}\boldsymbol{x}) = f_n(\Phi(\boldsymbol{x}))\rho_0(\mathrm{d}\boldsymbol{x}), \tag{18}$$

where we used that $\Psi_{\sharp}\mathbb{P}_{\infty} = \rho_0$. Next applying the guided flow in the second step gives us

$$\pi_n^{\mathrm{DAISI}}(\mathrm{d}\boldsymbol{x}) = \Phi_{\sharp}^{\boldsymbol{y}}\widetilde{\rho}_0(\mathrm{d}\boldsymbol{x}) \tag{19}$$

$$\overset{(17)}{\propto} f_n(\Phi(\Psi^{\boldsymbol{y}}(\boldsymbol{x})))(\Phi^{\boldsymbol{y}})_{\sharp}\rho_0(\mathrm{d}\boldsymbol{x}) = g_n^{\boldsymbol{y}}(\boldsymbol{x})\,\mathbb{P}_{\infty}^{\boldsymbol{y}}(\mathrm{d}\boldsymbol{x}), \tag{20}$$

where $g_n^{\boldsymbol{y}}(\boldsymbol{x}) := f_n(\Phi(\Psi^{\boldsymbol{y}}(\boldsymbol{x})))$ and we used that $\Phi^{\boldsymbol{y}}$ transports $\rho_0$ to $\mathbb{P}_{\infty}^{\boldsymbol{y}}$ by construction. Comparing (20) with (16), we see that in this limit, the base measures match, while the density ratios differ. Thus, the extremes $t_{\min} \to 1$ and $t_{\min} \to 0$ each match *only one component* of the true filtering distribution, motivating the choice of an intermediate $t_{\min} \in (0, 1)$ that provides the best trade-off between these two types of mismatch. We also note that larger $t_{\min}$ reduces computational cost by shortening the integration interval.

In Figure 2, we illustrate this trade-off using a simple 1D toy problem (see Appendix C.2 for details). When $t_{\min} = 0.01$, the distribution $\pi_n^{\mathrm{DAISI}}$ (in orange) shares characteristics of $\mathbb{P}_{\infty}^{\boldsymbol{y}}$, as predicted from (20). Conversely, for a large value such as $t_{\min} = 0.6$, the distribution closely matches $\hat{\pi}_n$, as expected. The intermediate setting $t_{\min} = 0.3$ yields a distribution that interpolates between these two extremes, yielding a distribution that better aligns with the filtering distribution, which combines features of both $\mathbb{P}_{\infty}^{\boldsymbol{y}}$ and $\hat{\pi}_n$ — as reflected by the noticeably lower Maximum Mean Discrepancy (MMD) score (Gretton et al., 2012).

**Effect of $\epsilon$.** We next examine the effect of the noise parameter $\epsilon$. In the absence of observations, applying the backward process (7) to the forecast samples, followed by the forward process (6), provides a mechanism to resample the forecast ensemble. In the deterministic case $\epsilon = 0$, the

original ensemble members are exactly recovered. Introducing a small amount of noise via $\epsilon$ allows the generation of "new" forecast samples that are likely under $\mathbb{P}_{\infty}$, yet remains close to the original forecasts. As $\epsilon$ increases, the resampled ensemble progressively deviates, and in the limit $\epsilon \to \infty$, we have $\pi_{n,0,\epsilon}^{\mathrm{DAISI}} \to \mathbb{P}_{\infty}^{\boldsymbol{y}}$ and therefore all ensemble information is lost. In the conditional setting, $\epsilon$ therefore controls the extent to which conditional samples retain forecast information. In Appendix E we provide a more precise statement of this in the form of a Bakry-Émery-type entropy dissipation result. In practice, we find that taking a small but non-zero $\epsilon$ is useful for mitigating the tendency of particles from collapsing to a single state as assimilation progresses. In our 1D toy experiment (Appendix C.2), we show that tuning both $t_{\min}$ and $\epsilon$ yields better performance than adjusting $t_{\min}$ alone.

### 3.2. Complexity Analysis

Assuming the dynamics model $\mathcal{F}(\cdot)$ incurs a cost of $\mathcal{O}(f(d_x))$, the prediction step (12) costs $\mathcal{O}(Jf(d_x))$. Denoting by $\mathcal{O}(g(d_x, d_y))$ the cost of a single Euler–Maruyama (EM) step for solving (6) or (7) (with or without guidance), the analysis step costs $\mathcal{O}(JTg(d_x, d_y))$, where $T$ is the number of EM steps. Hence, the overall computational complexity of DAISI is $\mathcal{O}\big(J(f(d_x) + Tg(d_x, d_y))\big)$. For comparison, the ensemble transform Kalman filter (ETKF) has cost $\mathcal{O}\big(J(f(d_x) + (d_x + d_y)J + J^2)\big)$. Since the forward evaluation of the U-net parameterizing the drift $\boldsymbol{b}(t, \boldsymbol{z})$ scales linearly with input dimension (i.e., $g(d_x, d_y) = \mathcal{O}(d_x + d_y)$), DAISI achieves comparable cost to ETKF when $T \sim J$, while scaling linearly in ensemble size $J$—in contrast to ETKF's cubic scaling in $J$. However, in settings where $J$ is small (e.g., $\mathcal{O}(10)$), DAISI can be more expensive due to the need for $T = \mathcal{O}(10^2)$ EM steps to accurately solve the forward and backward SDEs.

We note that the `Forecast`, `Inverse sampling`, and `Guided sampling` steps in Algorithm 1 are all embarrassingly parallelizable, enabling further reductions in computational cost at the expense of increased memory use.

### 3.3. Related Works

Early work on diffusion priors for data assimilation includes Score-based Data Assimilation (SDA) (Rozet & Louppe, 2023), which learns the score of full trajectories to perform smoothing without an explicit forecast model. While effective, it is memory-intensive and scales poorly to the size of the filtering window. Adapting it to online filtering requires truncating the observation window, introducing a bias.

Closer to our setting, Yang et al. (2025) address linear filtering by guiding a pre-trained diffusion model using both forecasts and observations. Instead of solving the back-

ward SDE, they rely on SDEdit (Meng et al., 2022), which partially inverts a sample by re-noising it, and incorporate observations via RePaint (Lugmayr et al., 2022), an iterative noising-denoising procedure. Together, these steps create a trade-off between preserving forecast information and enforcing the conditioning. In Appendix C.4.6, we show that replacing the backward SDE with SDEdit leads to substantially worse results.

The Ensemble Score Filter (EnSF) (Bao et al., 2024a) replaces the learned score with an analytical approximation, enabling training-free assimilation of nonlinear observations. However, it struggles in sparse settings due to the lack of a learned structure. An extension of EnSF in latent-space (Si & Chen, 2025) partially addresses this; however, this does not easily amortize over observation models and its results depend heavily on the quality of the trained autoencoder. EnSF and its variants have been applied to the Surface Quasi-Geostrophic model (Bao et al., 2025; Yin et al., 2024; Liang et al., 2025), and extensions with Gaussian mixture score approximations (Zhang et al., 2025) and problem-dependent data coupling (Transue et al., 2025) have been proposed.

Finally, recent works explore guidance for diffusion-based forecasting models. For instance, FlowDAS (Chen et al., 2025) employs stochastic interpolants to learn a one-step forecast distribution $p(\boldsymbol{x}_{n+1}|\boldsymbol{x}_n)$, which can serve as a generative prior to condition on new observations $\boldsymbol{y}_{n+1}$ via guidance. However, this procedure effectively targets the local conditional $p(\boldsymbol{x}_{n+1}|\boldsymbol{x}_n, \boldsymbol{y}_{n+1})$, which can be arbitrarily far from the true filtering distribution $p(\boldsymbol{x}_{n+1}|\boldsymbol{y}_{1:n+1})$. The work (Savary et al., 2026)[1] proposes a correction to this mismatch using particle reweighting. However, this reintroduces particle-filter-style weight degeneracy and thus susceptibility to the curse of dimensionality.

## 4. Experiments

We evaluate DAISI on three systems: the Lorenz '63 (L63) system (Lorenz, 1963), a Surface Quasi-Geostrophic system (SQG) (Tulloch & Smith, 2009), and SEVIR, a real-world radar dataset (Veillette et al., 2020).

### 4.1. Lorenz '63 System

The goal of this experiment is to demonstrate on the L63 system that an appropriate choice of $t_{\min}$ and $\epsilon$ can approximate the filtering distribution closely. The L63 system is integrated with standard parameters from random initial conditions, and the final 500 steps are used to test the assimilation methods. For reference, we compare DAISI with the Bootstrap Particle Filter (BPF) (Gordon et al., 1993). As

---

[1]A minor difference from (Chen et al., 2025) is that it uses GenCast (Price et al., 2025) as the predictive model instead of the stochastic interpolant.

|  | RMSE ($\downarrow$) | CRPS ($\downarrow$) | SSR ($\approx 1$) |
|---|---|---|---|
| BPF | $\mathbf{1.18}_{\pm 0.50}$ | $\mathbf{1.46}_{\pm 0.49}$ | $2.57_{\pm 0.56}$ |
| DAISI (no inversion) | $5.90_{\pm 1.09}$ | $7.17_{\pm 1.38}$ | $\underline{1.40}_{\pm 0.25}$ |
| DAISI ($t_{\min} \approx 0, \epsilon = 0$) | $7.04_{\pm 0.90}$ | $11.9_{\pm 1.55}$ | $0.06_{\pm 0.02}$ |
| DAISI (tuned $t_{\min}, \epsilon = 0$) | $2.44_{\pm 2.89}$ | $3.58_{\pm 4.79}$ | $\mathbf{1.07}_{\pm 1.09}$ |
| DAISI (tuned both $t_{\min}$ & $\epsilon$) | $\underline{2.03}_{\pm 1.01}$ | $\underline{2.61}_{\pm 1.38}$ | $1.08_{\pm 0.41}$ |
| DAISI (tuned both, set $\epsilon = 0$) | $2.05_{\pm 1.31}$ | $3.19_{\pm 2.14}$ | $0.32_{\pm 0.09}$ |

*Table 1.* Summary metrics on the L63 example. We display the mean and standard deviation across 10 independent experiments, over the last 100 assimilation steps. The best score for each metric is highlighted in **bold** and the second best with an underline.

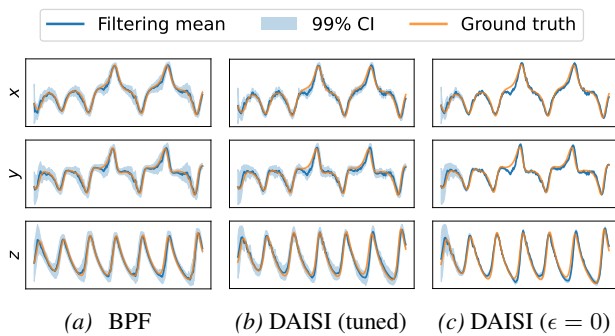

*(a)* BPF     *(b)* DAISI (tuned)     *(c)* DAISI ($\epsilon = 0$)

*Figure 3.* A comparison of filtering results from bootstrap particle filter (BPF) vs DAISI on the L63 system. We display the ground truth alongside the filtering mean and 99% credible interval.

BPF is asymptotically exact and non-degenerate for the low-dimensional L63 system, we take this as the "ground truth" filtering result to compare against. We also use an asymptotically exact guidance method for DAISI (see Appendix A.5.3) to isolate DAISI's intrinsic error.

Table 1 summarizes the RMSE and probabilistic metrics—the Continuous Ranked Probability Score (CRPS) and Spread–Skill Ratio (SSR). For comparison, we also include the results of DAISI without inverse-sampling (i.e., sampling from $\mathbb{P}_\infty^{\boldsymbol{y}}$); this performs substantially worse than BPF across all metrics, demonstrating that incorporating dynamical information is essential for accurate filtering. With $t_{\min} \approx 0$ and $\epsilon = 0$, DAISI still underperforms, and yields worse results than the version without inversion. Tuning $t_{\min}$ alone[2] already leads to substantial improvement, and jointly tuning $t_{\min}$ and $\epsilon$ produces the best performance, achieving metrics closest to those of BPF.

Visually, we see that after tuning both $t_{\min}$ and $\epsilon$, the DAISI filtering results (Figure 3b) closely resemble those from BPF (Figure 3a). To see the effect of $\epsilon$, we repeat the experiment with the same tuned $t_{\min}$ but fix $\epsilon = 0$. The results, shown in Figure 3c and summarized in the final row of Table 1, exhibit similar mean behaviour but has narrower uncertainty bands.

---

[2]The hyperparameters are tuned on a short held-out validation trajectory to minimize CRPS.

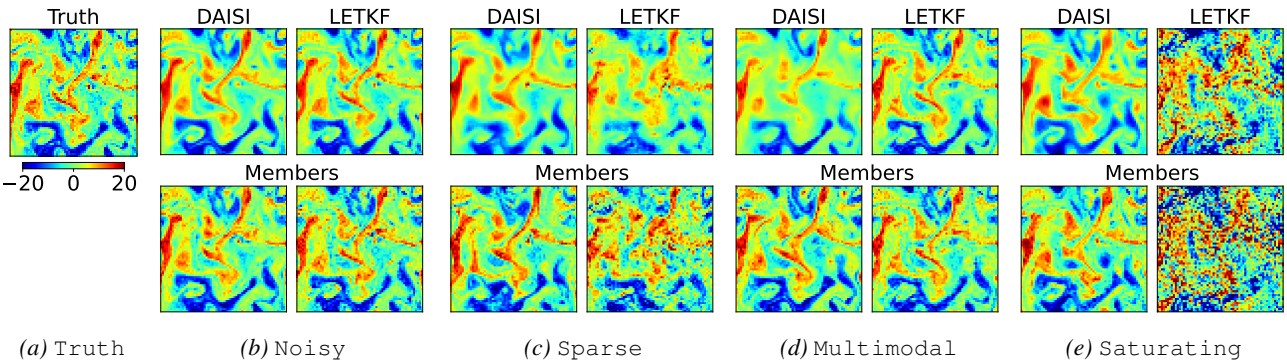

*(a)* `Truth`  *(b)* `Noisy`  *(c)* `Sparse`  *(d)* `Multimodal`  *(e)* `Saturating`

*Figure 4.* Ensemble mean and a single member for DAISI and LETKF for SQG experiments at the last step of the assimilated trajectory.

*Table 2.* Experiment configurations.

| Experiment | Data | Obs operator | $\sigma_{\mathrm{obs}}$ | Sparsity |
|---|---|---|---|---|
| Noisy | SQG-64 | $x$ | 5 | 25% |
| Sparse | SQG-64 | $x$ | 1 | 5% |
| Multimodal | SQG-64 | $(x/7)^2$ | 1 | 25% |
| Saturating | SQG-64 | $\arctan(x)$ | 0.01 | 25% |
| High-dim. | SQG-256 | $\mathrm{Avg}(x)$ | 1 | 5% |
| SEVIR | SEVIR | $x$ | 0.001 | 10% |

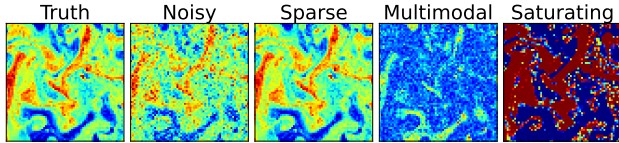

*Figure 5.* Visualization of the observations for each configuration before sparsity is applied.

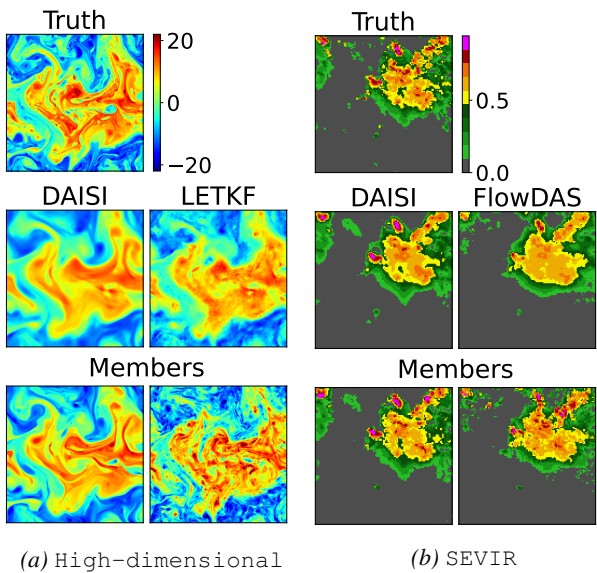

*(a)* `High-dimensional`  *(b)* `SEVIR`

*Figure 6.* Ensemble mean and a single member for DAISI and LETKF/FlowDAS for the `High-dimensional`/`SEVIR` experiments at the last step of the assimilated trajectory.

This reduction in ensemble spread leads to a much lower SSR and a degraded CRPS, despite similar RMSE. This highlights the importance of $\epsilon$ for maintaining ensemble spread, while tuning $t_{\min}$ is necessary for accuracy.

### 4.2. Surface Quasi-Geostrophic (SQG) Dynamics

We next evaluate DAISI on a Surface Quasi-Geostrophic (SQG) model, a standard benchmark for turbulent geophysical flows. The dynamics evolve a scalar field $\theta$ under nonlinear advection, combined with forcing and dissipation mechanisms including thermal relaxation and hyperdiffusion. Despite its simplicity, SQG exhibits strong sensitivity to initial conditions and multiscale turbulent behavior while remaining computationally tractable, making it a valuable benchmark for DA (Tulloch & Smith, 2009).

Following (Liang et al., 2025), we use a $64 \times 64$ grid over 100 time steps with 3-hour intervals. DAISI is tested across a range of observation operators, noise levels, and sparsity settings listed in Table 2, and visualized in Figure 5. To assess scalability we additionally include a single $256 \times 256$

experiment with an averaging observation operator similar to lower-resolution sensing. Since the guidance methods in Section 2.3 have been applied almost exclusively to Gaussian observations, we stay within this setting, but note that this is not a limitation with DAISI.

We compare DAISI to both classical and ML-based DA methods. Classical baselines include the Local Ensemble Transform Kalman Filter (LETKF), while ML baselines include FlowDAS, Score-based Data Assimilation (SDA) and the Ensemble Score Filter (EnSF). We also evaluate a DAISI variant that replaces the numerical model with the learned FlowDAS model, denoted DAISI-ML. Since SDA is originally a smoothing method, it is not directly comparable to filtering approaches. To address this, we additionally consider a filtering adaptation of SDA (see Appendix C.4.3) and report results for both filtering and smoothing.

*Table 3.* The CRPS for experiments on SQG and SEVIR. We display the mean and standard deviation across 10 independent trajectories, averaged over the last 20 (10 for SEVIR) steps. The best score for each experiment is highlighted in **bold** and the second best with an underline. Since SDA (smoothing) solves a different problem, we exclude it from the relative ranking.

| Experiment | DAISI | LETKF | FlowDAS | EnSF | SDA (filtering) | DAISI-ML | SDA (smoothing) |
|---|---|---|---|---|---|---|---|
| Noisy | 1.32$_{\pm 0.13}$ | 1.34$_{\pm 0.14}$ | 2.81$_{\pm 0.43}$ | 5.15$_{\pm 0.78}$ | **1.13**$_{\pm 0.10}$ | 1.34$_{\pm 0.13}$ | 1.21$_{\pm 0.11}$ |
| Sparse | **1.73**$_{\pm 0.18}$ | 2.35$_{\pm 0.28}$ | 3.34$_{\pm 0.53}$ | 4.20$_{\pm 0.56}$ | 2.22$_{\pm 0.25}$ | 1.72$_{\pm 0.16}$ | 1.53$_{\pm 0.11}$ |
| Multimodal | **1.81**$_{\pm 0.44}$ | 1.97$_{\pm 0.69}$ | 3.66$_{\pm 0.42}$ | 6.38$_{\pm 0.75}$ | 3.88$_{\pm 0.48}$ | 1.78$_{\pm 0.41}$ | 3.82$_{\pm 0.32}$ |
| Saturating | 1.54$_{\pm 0.09}$ | 5.24$_{\pm 0.35}$ | 2.41$_{\pm 0.39}$ | **1.33**$_{\pm 0.16}$ | 4.20$_{\pm 0.44}$ | 1.53$_{\pm 0.11}$ | 3.37$_{\pm 0.14}$ |
| SEVIR | **0.016**$_{\pm 0.01}$ | 0.018$_{\pm 0.01}$ | 0.045$_{\pm 0.01}$ | 0.075$_{\pm 0.02}$ | 0.018$_{\pm 0.01}$ | 0.016$_{\pm 0.01}$ | 0.013$_{\pm 0.00}$ |

All methods use 20 members and are tuned specifically for each setting (see hyperparameter ablation for Sparse in Figures 18–19; we observe that the performance is not highly sensitive to their precise values, requiring minimal tuning). With the exception of FlowDAS, we initialize all methods from $x_0 \sim \mathcal{N}(x_0^{\text{gt}}, \sigma_{\text{init}}^2 I)$ with $\sigma_{\text{init}} = 3$, and propagate ensemble members using the numerical model. FlowDAS instead uses its own learned autoregressive forecast model conditioned on the six previous states, so assimilation begins from samples drawn from $p(x_0 \mid x_{-1}^{\text{gt}}, \ldots, x_{-6}^{\text{gt}})$. DAISI uses a U-Net architecture based on Karras et al. (2022) with 3.5M parameters, while all other models use their original implementations.

Table 3 summarizes CRPS across all SQG experiments. DAISI consistently achieves accurate assimilation, producing temporally coherent and physically plausible ensemble members (Figures 4, 6 and 7). It matches LETKF in the noisy setting, and clearly outperforms it under sparse or nonlinear observations. DAISI also remains stable when assimilation is performed every 12 hours instead of every 3, whereas LETKF degrades (Figure 22). In multimodal settings, DAISI reliably tracks many plausible modes, whereas LETKF collapses to a single mode and typically diverges if that mode becomes inconsistent. In the high-dimensional experiment, both DAISI and LETKF track the ensemble mean accurately, but exhibit qualitatively different behaviors. DAISI produces smoother reconstructions, while LETKF tends to introduce spurious fine-scale structure. This behavior is likely due to the challenging observation setup and the lack of tuning at this resolution.

For the saturating arctan observations, DAISI performs comparably to EnSF, which handles such nonlinearities well. However, EnSF fails in other regimes due to the lack of a learned prior, and is prone to mode collapse, requiring inflation to perform well. LETKF diverges under these nonlinear observations, and FlowDAS underperforms in all settings, as it samples from $p(x_{n+1}|x_n, y_{n+1})$ rather than $p(x_{n+1}|y_{1:n+1})$ at each step, leading to errors accumulating over time. Replacing the numerical model with the FlowDAS forecast model within DAISI (DAISI-ML) shows no performance degradation, confirming FlowDAS's failures arise from the filtering scheme rather than its forecast

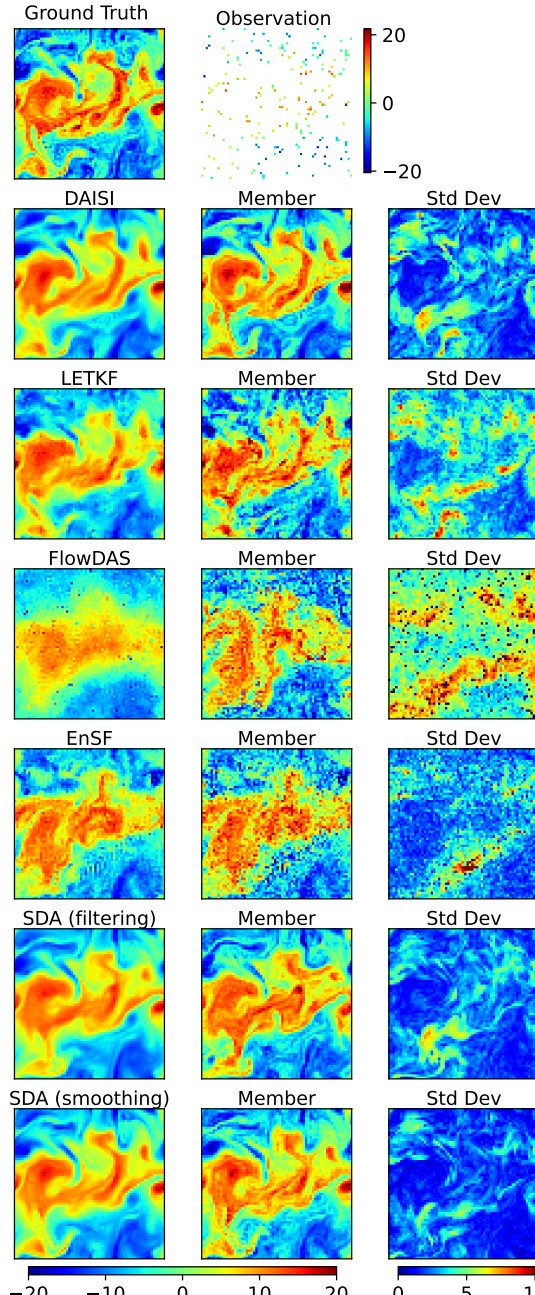

*Figure 7.* Ensemble mean, members, and standard deviation for each method at final assimilation step for the Sparse experiment.

model. Additional results, including RMSE and SSR scores, are provided in Appendix C.4.5. We also display results for more realistic observations simulating moving satellite tracks (refered to as `Non-stationary`), showing comparable performance to LETKF.

On the linear problems (`Noisy`, `Sparse` and `SEVIR`), SDA smoothing consistently outperforms DAISI. However, this is likely due to the effect of smoothing, which improves past state estimation. Evidently, the results at the last time step are similar to DAISI (Figures 23, 24 and 28). The filtering variant of SDA is slightly worse than DAISI, except on `Noisy`. On the nonlinear problems (`Multimodal` and `Saturating`), both the smoothing and filtering variants of SDA struggle to achieve strong performance.

### 4.3. Precipitation Nowcasting using SEVIR

To evaluate on a real-world dataset, we apply DAISI to the Storm EVent Imagery and Radar (SEVIR) dataset (Veillette et al., 2020), a radar observation dataset of convective storms over the United States. We use the vertically integrated liquid on a $384 \times 384$ km grid at $2$ km resolution available every $10$ min for $250$ min (Gao et al., 2023).

This dataset has been applied to DA using the FlowDAS method by Chen et al. (2025). Thus, we mirror their setting and consider a linear Gaussian observation with standard deviation 0.001 and 10% sparsity. To generate the forecasts, we use the pre-trained forecasting model of (Chen et al., 2025). This takes six consecutive frames as input and predicts the state 10 min into the future.

As shown in Figure 6, both DAISI and FlowDAS are able to accurately reconstruct the state, although the peaks are better represented in DAISI. This is also reflected by the much lower CRPS, as shown in Table 3.

## 5. Conclusions

We introduced DAISI, a robust and flexible filtering framework built on flow-based generative priors. A key component is the inversion of the generative SDE: running the flow backward from the forecast ensemble recovers latent representations that serve as informative initialization for conditional sampling, effectively combining the prior, the forecast model, and observational information in a unified way. Empirically, DAISI delivers accurate filtering results across a spectrum of challenging settings, including sparse, noisy, nonlinear, and multimodal observations.

## 6. Limitations & Future Work

A fundamental limitation of DAISI is that it does not sample exactly from the true filtering distribution. While our experiments show that tuning $t_{\min}$ and $\epsilon$ can mitigate this

discrepancy, developing principled correction schemes for debiasing remains an important direction for future work.

DAISI also inherits the high inference cost of ODE/SDE-based generative models, driven by the high number of function evaluations required to integrate generative SDEs. Approaches such as performing data assimilation in latent spaces (Andry et al., 2025) or distilling the flow model (Boffi et al., 2026) may help reduce these costs, and we leave these extensions for future work.

Its performance is further limited by the guidance mechanism. Although MMPS performed well even under strongly nonlinear observations, it is inherently biased. A promising direction is to reduce this bias by replacing MMPS with more accurate estimators of the guidance term, for example using stochastic flow maps (Potaptchik et al., 2026).

Finally, our experimental setup is simplified relative to operational data assimilation systems. Given recent progress in ML-based weather forecasting (Alet et al., 2025; Andrae et al., 2025; Larsson et al., 2025), a natural next step is to scale DAISI to realistic large-scale forecasting settings with complex observation operators (Huang et al., 2024; Andry et al., 2025; Savary et al., 2026).

## Acknowledgements

This research is financially supported by the Swedish Research Council (grant no: 2024-05011) the Wallenberg AI, Autonomous Systems and Software Program (WASP) funded by the Knut and Alice Wallenberg Foundation, and the Excellence Center at Linköping–Lund in Information Technology (ELLIIT). Our computations were enabled by the Berzelius resource at the National Supercomputer Centre, provided by the Knut and Alice Wallenberg Foundation. Landelius was financially supported by the Swedish Foundation for Strategic Research. ST acknowledges support by a Department of Defense Vannevar Bush Faculty Fellowship held by Prof. Andrew Stuart, and by the SciAI Center, funded by the Office of Naval Research (ONR), under Grant Number N00014-23-1-2729.

## Impact Statement

Improving the accuracy and efficiency of initial state assimilation can significantly enhance prediction reliability in high-dimensional nonlinear systems. In weather forecasting, better assimilation of observations into initial conditions is key to improving short- and medium-term forecasts, with downstream benefits for sectors such as agriculture, energy, transportation, and disaster preparedness. Developing robust probabilistic data assimilation methods is also crucial to avoid overconfident or miscalibrated state estimates in safety-critical settings.

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

# A. Details on Stochastic Interpolants

## A.1. Stochastic Interpolants

Given a measure $\rho_1$, a (linear one-sided) stochastic interpolant is a stochastic process of the following form:

$$z_t = \alpha_t z_0 + \beta_t z_1 \tag{21}$$

where $z_1 \sim \rho_1$ and $z_0 \sim \mathcal{N}(\mathbf{0}, \mathbf{I})$. We have the following result, found in Theorem 2.6 of (Albergo et al., 2023).

**Proposition A.1.** *The probability distribution $\rho_t$ of the interpolant $z_t$ admits Lebesgue densities $p(t)$ for all times $t \in [0, 1]$ and moreover, it satisfies the endpoint conditions $\rho(0) = \mathcal{N}(\mathbf{0}, \mathbf{I}), \rho(1) = \rho_1$. In addition, the Lebesgue densities satsify the transport equation*

$$\partial_t p_t + \nabla \cdot (\boldsymbol{b}_t p_t) = 0, \tag{22}$$

*where $\boldsymbol{b}_t$ is the drift of the interpolant, defined by*

$$\boldsymbol{b}(t, z_t) = \mathbb{E}[\dot{z}_t | z_t]. \tag{23}$$

This result implies that the flow map $\{\Phi_t\}_{t \in [0,1]}$ of the ODE

$$\frac{\mathrm{d}z_t}{\mathrm{d}t} = \boldsymbol{b}(t, z_t). \tag{24}$$

transports $\mathcal{N}(\mathbf{0}, \mathbf{I})$ to $\rho_1$, i.e., $(\Phi_1)_\sharp \mathcal{N}(\mathbf{0}, \mathbf{I}) = \rho_1$.

Furthermore, (Albergo et al., 2023) shows that the drift $\boldsymbol{b}(t, z)$ can be learned by minimizing the objective

$$\mathcal{L}(\theta) = \mathbb{E}_{z_0 \sim \mathcal{N}(\mathbf{0}, \mathbf{I}), z_1 \sim \rho_1, t \sim \mathcal{U}([0,1])} \left[ \|\boldsymbol{b}_\theta(t, z_t) - (\dot{\alpha}_t z_0 + \dot{\beta}_t z_1)\|^2 \right]. \tag{25}$$

## A.2. Turning ODEs into SDEs

We now show how one can transform an ODE into an SDE that shares the same marginal laws.

By Theorem A.1 we know that the marginals of the stochastic interpolant in Equation (21) satisfy the continuity equation

$$\partial_t \rho_t + \nabla \cdot (\boldsymbol{b}_t \rho_t) = 0. \tag{26}$$

By noticing that for any non-negative $\epsilon_t$, we have the identity

$$\epsilon_t \Delta p_t = \epsilon_t \nabla \cdot (p_t \nabla \log p_t) = \nabla \cdot (\epsilon_t \boldsymbol{s} p_t), \tag{27}$$

we can add and subtract a term from Equation (26), giving us the equivalent expression

$$\partial_t p_t = -\nabla \cdot ((\boldsymbol{b} + \epsilon_t \boldsymbol{s}) p_t) + \epsilon_t \Delta p_t. \tag{28}$$

We recognize that this is the Fokker-Planck equation whose samples satisfies the forward/backward SDEs

$$\mathrm{d}z_t = (\boldsymbol{b}(t, z_t) \pm \epsilon_t \boldsymbol{s}(t, z_t)) \mathrm{d}t + \sqrt{2\epsilon_t} \mathrm{d}W_{\pm t}, \tag{29}$$

$$z_0 \sim \rho_0, \quad z_1 \sim \rho_1, \quad t \in [0, 1]. \tag{30}$$

Thus, we have recovered another family of generative models based on SDEs that share the same marginals as the generative ODE Equation (24).

## A.3. Identities for drifts and score

We now derive some important relationships between the drift $\boldsymbol{b}(t, z)$, score $\boldsymbol{s}(t, z) := \nabla \log p_t(z)$, and $\mathbb{E}[z_1 | z_t]$.

Taking the time derivative of (21), we have

$$\dot{z}_t = \dot{\alpha}_t z_0 + \dot{\beta}_t z_1 \tag{31}$$

and therefore by definition of the drift, we have

$$\boldsymbol{b}(t, \boldsymbol{z}) = \dot{\alpha}_t \mathbb{E}[\boldsymbol{z}_0 | \boldsymbol{z}_t] + \dot{\beta}_t \mathbb{E}[\boldsymbol{z}_1 | \boldsymbol{z}_t]. \tag{32}$$

Now, since $\boldsymbol{z}_0 \sim \mathcal{N}(\boldsymbol{0}, \mathbf{I})$, the interpolant expression implies that $\boldsymbol{z}_t | \boldsymbol{z}_1 \sim \mathcal{N}(\beta_t \boldsymbol{z}_1, \alpha_t^2 I)$. Thus, we can write down the following Tweedie-type estimate for the score function

$$\boldsymbol{s}(t, \boldsymbol{z}_t) := \nabla_{\boldsymbol{z}_t} \log p(\boldsymbol{z}_t) \tag{33}$$

$$= \frac{\nabla_{\boldsymbol{z}_t} p(\boldsymbol{z}_t)}{p(\boldsymbol{z}_t)} \tag{34}$$

$$= \frac{\nabla_{\boldsymbol{z}_t} \int p(\boldsymbol{z}_t | \boldsymbol{z}_1) p(\boldsymbol{z}_1) \mathrm{d}\boldsymbol{z}_1}{p(\boldsymbol{z}_t)} \tag{35}$$

$$= \frac{\int \nabla_{\boldsymbol{z}_t} p(\boldsymbol{z}_t | \boldsymbol{z}_1) p(\boldsymbol{z}_1) \mathrm{d}\boldsymbol{z}_1}{p(\boldsymbol{z}_t)} \tag{36}$$

$$= \frac{\int (\nabla_{\boldsymbol{z}_t} \log p(\boldsymbol{z}_t | \boldsymbol{z}_1)) p(\boldsymbol{z}_t | \boldsymbol{z}_1) p(\boldsymbol{z}_1) \mathrm{d}\boldsymbol{z}_1}{p(\boldsymbol{z}_t)} \tag{37}$$

$$= \int (\nabla_{\boldsymbol{z}_t} \log p(\boldsymbol{z}_t | \boldsymbol{z}_1)) p(\boldsymbol{z}_1 | \boldsymbol{z}_t) \mathrm{d}\boldsymbol{z}_1. \tag{38}$$

Using that $\boldsymbol{z}_t | \boldsymbol{z}_1 \sim \mathcal{N}(\beta_t \boldsymbol{z}_1, \alpha_t^2 I)$, we get

$$\nabla_{\boldsymbol{z}_t} \log p(\boldsymbol{z}_t | \boldsymbol{z}_1) = -\frac{\boldsymbol{z}_t - \beta_t \boldsymbol{z}_1}{\alpha_t^2}, \tag{39}$$

which allows us to obtain

$$\boldsymbol{s}(t, \boldsymbol{z}_t) = -\frac{\boldsymbol{z}_t - \beta_t \mathbb{E}[\boldsymbol{z}_1 | \boldsymbol{z}_t]}{\alpha_t^2}. \tag{40}$$

Now, taking the conditional expectation $\mathbb{E}[\cdot | \boldsymbol{z}_t]$ of (21) we get that

$$\boldsymbol{z}_t = \alpha_t \mathbb{E}[\boldsymbol{z}_0 | \boldsymbol{z}_t] + \beta_t \mathbb{E}[\boldsymbol{z}_1 | \boldsymbol{z}_t], \tag{41}$$

which implies

$$\mathbb{E}[\boldsymbol{z}_0 | \boldsymbol{z}_t] = -\alpha_t \boldsymbol{s}(t, \boldsymbol{z}_t). \tag{42}$$

Finally, combining these with the expression for $\boldsymbol{b}$ found earlier, we arrive at

$$\boldsymbol{s}(t, \boldsymbol{z}_t) = \frac{\beta_t \boldsymbol{b}(t, \boldsymbol{z}_t) - \dot{\beta}_t \boldsymbol{z}_t}{\alpha_t \gamma_t}, \tag{43}$$

and

$$\mathbb{E}[\boldsymbol{z}_1 | \boldsymbol{z}_t] = \frac{\alpha_t \boldsymbol{b}(t, \boldsymbol{z}_t) - \dot{\alpha}_t \boldsymbol{z}_t}{\gamma_t}, \tag{44}$$

where $\gamma_t := \dot{\beta}_t \alpha_t - \beta_t \dot{\alpha}_t$.

### A.4. Conditional drift and score

Now replace the data distribution $p_1$ with the posterior distribution

$$p_1^{\boldsymbol{y}}(\boldsymbol{z}_1) := \frac{p(\boldsymbol{y} | \boldsymbol{z}_1) p_1(\boldsymbol{z}_1)}{p(\boldsymbol{y})}, \tag{45}$$

and again consider a stochastic interpolant (21), where now $z_1 \sim p_1^{\boldsymbol{y}}$. Then the law of the interpolant is given by

$$p_t^{\boldsymbol{y}}(z_t) := \int p(z_t|z_1) p_1^{\boldsymbol{y}}(z_1) \mathrm{d}z_1 \tag{46}$$

$$= \frac{1}{p(\boldsymbol{y})} \int p(z_t|z_1) p_1(z_1) p(\boldsymbol{y}|z_1) \mathrm{d}z_1 \tag{47}$$

$$= \frac{1}{p(\boldsymbol{y})} \int p(z_1|z_t) p_t(z_t) p(\boldsymbol{y}|z_1) \mathrm{d}z_1 \tag{48}$$

$$= \frac{1}{p(\boldsymbol{y})} p_t(z_t) \int p(z_1|z_t) p(\boldsymbol{y}|z_1) \mathrm{d}z_1 \tag{49}$$

$$= \frac{1}{p(\boldsymbol{y})} p_t(z_t) p(\boldsymbol{y}|z_t), \tag{50}$$

where $p_t(z_t)$ is the law of the **unconditional** interpolant. We note that this derivation relies on the fact that the interpolant $z_t$ has the same conditional structure $p(z_t|z_s)$, regardless of whether the target is $p_1(z_1)$ or $p_1^{\boldsymbol{y}}(z_1)$.

Thus, the conditional score is given by

$$\boldsymbol{s}^{\boldsymbol{y}}(t, z_t) := \nabla_{z_t} \log p_t^{\boldsymbol{y}}(z_t) \tag{51}$$

$$= \nabla_{z_t} \log p_t(z_t) + \nabla_{z_t} \log p(\boldsymbol{y}|z_t) \tag{52}$$

$$= \boldsymbol{s}(t, z_t) + \nabla_{z_t} \log p(\boldsymbol{y}|z_t), \tag{53}$$

where $\boldsymbol{s}(t, z_t)$ denotes the score of the interpolant that samples from the original measure $p_1$. Next, using our relation between score and drift, which holds for *arbitrary* data measures and therefore also the case of sampling from the posterior, we have the following corresponding drift

$$\boldsymbol{b}^{\boldsymbol{y}}(t, z_t) = \frac{\dot{\beta}_t z_t + \alpha_t \gamma_t \boldsymbol{s}^{\boldsymbol{y}}(t, z_t)}{\beta_t} \tag{54}$$

$$= \frac{\dot{\beta}_t z_t + \alpha_t \gamma_t (\boldsymbol{s}(t, z_t) + \nabla_{z_t} \log p(\boldsymbol{y}|z_t))}{\beta_t} \tag{55}$$

$$= \boldsymbol{b}(t, z_t) + \lambda_t \nabla_{z_t} \log p(\boldsymbol{y}|z_t), \tag{56}$$

where $\lambda_t := \alpha_t \gamma_t / \beta_t$, and $\boldsymbol{b}(t, z_t)$ denotes the drift corresponding to the original interpolant. Then, by Theorem A.1, the flow of the ODE with drift $\boldsymbol{b}^{\boldsymbol{y}}$ transports $\mathcal{N}(\boldsymbol{0}, \mathbf{I})$ to $p_1^{\boldsymbol{y}}$. This observation serves as the basis for sampling from the posterior using *guidance*, which we look at in the following section.

## A.5. Guidance Methods

As demonstrated in the previous section, conditional sampling requires access to the likelihood score $\nabla_{z_t} \log p(\boldsymbol{y}|z_t)$. Unfortunately, this is, for the most part, analytically intractable and requires approximations. In this section, we present three such approximations that are used in this work. In addition to the approximation below, we may also multiply the likelihood score with a guidance strength $\zeta > 0$, which we tune based on the problem setup.

### A.5.1. DIFFUSION POSTERIOR SAMPLING (DPS)

DPS (Chung et al., 2023) proceeds by approximating $p(\boldsymbol{y} \mid z_t) = \mathbb{E}_{z_1|z_t}[p(\boldsymbol{y} \mid z_1)]$ by simply taking the expectation inside the likelihood:

$$p(\boldsymbol{y} \mid z_t) \approx p(\boldsymbol{y} \mid \hat{z}_1), \tag{57}$$

where $\hat{z}_1(z_t) = \mathbb{E}[z_1|z_t]$. This incurs a bias, known as the Jensen gap; however, in many inverse problem settings, the approximation is known to work well. The resulting likelihood score used in DPS is thus

$$\nabla_{z_t} \log p(\boldsymbol{y} \mid z_t) \approx \nabla_{z_t} \log p(\boldsymbol{y} \mid \hat{z}_1(z_t)), \tag{58}$$

where we estimate $\hat{z}_1(z_t) = \mathbb{E}[z_1|z_t]$ explicitly from the drift via the relation (44).

When $p(\boldsymbol{y} \mid \boldsymbol{z}_1)$ is a Gaussian, we use the additional scaling factor proposed by (Chung et al., 2023) which gives the expression

$$\nabla_{\boldsymbol{z}_t} \log p(\boldsymbol{y} \mid \boldsymbol{z}_t) \approx \nabla_{\boldsymbol{z}_t}^\top \mathbb{E}[\boldsymbol{z}_1|\boldsymbol{z}_t] \mathbf{H}_t^\top (\boldsymbol{y} - \mathcal{H}(\mathbb{E}[\boldsymbol{z}_1|\boldsymbol{z}_t])) \tag{59}$$

where $\mathbf{H}_t := \nabla_{\boldsymbol{z}_1} \mathcal{H}(\boldsymbol{z}_1)|_{\boldsymbol{z}_1 = \mathbb{E}[\boldsymbol{z}_1|\boldsymbol{z}_t]}$.

### A.5.2. MOMENT-MATCHING POSTERIOR SAMPLING (MMPS)

MMPS extends DPS by considering the approximation $p(\boldsymbol{z}_1|\boldsymbol{z}_t) \approx \mathcal{N}(\mathbb{E}[\boldsymbol{z}_1|\boldsymbol{z}_t], \mathbb{V}[\boldsymbol{z}_1|\boldsymbol{z}_t])$, where $\mathbb{E}[\boldsymbol{z}_1|\boldsymbol{z}_t]$ is obtained as before, and $\mathbb{V}[\boldsymbol{z}_1|\boldsymbol{z}_t] := \mathbb{E}[\boldsymbol{z}_1 \boldsymbol{z}_1^\top|\boldsymbol{z}_t] - \mathbb{E}[\boldsymbol{z}_1|\boldsymbol{z}_t]\mathbb{E}[\boldsymbol{z}_1|\boldsymbol{z}_t]^\top$ can be computed using the following formula (Rozet et al., 2024):

$$\mathbb{V}[\boldsymbol{z}_1|\boldsymbol{z}_t] = \frac{\alpha_t^2}{\beta_t} \nabla_{\boldsymbol{z}_t}^\top \mathbb{E}[\boldsymbol{z}_1|\boldsymbol{z}_t]. \tag{60}$$

Subsequently, the likelihood score can be approximated as

$$\nabla_{\boldsymbol{z}_t} \log p(\boldsymbol{y}|\boldsymbol{z}_t) = \nabla_{\boldsymbol{z}_t} \log \left( \int_{\mathbb{R}^d} p(\boldsymbol{y}|\boldsymbol{z}_1) p(\boldsymbol{z}_1|\boldsymbol{z}_t) \mathrm{d}\boldsymbol{z}_1 \right) \tag{61}$$

$$\approx \nabla_{\boldsymbol{z}_t} \log \left( \int_{\mathbb{R}^d} p(\boldsymbol{y}|\boldsymbol{z}_1) \mathcal{N}(\boldsymbol{z}_1 \mid \mathbb{E}[\boldsymbol{z}_1|\boldsymbol{z}_t], \, \mathbb{V}[\boldsymbol{z}_1|\boldsymbol{z}_t]) \mathrm{d}\boldsymbol{z}_1 \right) \tag{62}$$

$$\approx \nabla_{\boldsymbol{z}_t}^\top \mathbb{E}[\boldsymbol{z}_1|\boldsymbol{z}_t] \mathbf{H}_t^\top \left( \sigma_{\boldsymbol{y}}^2 \mathbf{I} + \frac{\alpha_t^2}{\beta_t} \mathbf{H}_t \nabla_{\boldsymbol{z}_t}^\top \mathbb{E}[\boldsymbol{z}_1|\boldsymbol{z}_t]) \mathbf{H}_t^\top \right)^{-1} (\boldsymbol{y} - \mathcal{H}(\mathbb{E}[\boldsymbol{z}_1|\boldsymbol{z}_t])), \tag{63}$$

where $\mathbf{H}_t := \nabla_{\boldsymbol{z}_1} \mathcal{H}(\boldsymbol{z}_1)|_{\boldsymbol{z}_1 = \mathbb{E}[\boldsymbol{z}_1|\boldsymbol{z}_t]}$. The last line becomes exact when $p(\boldsymbol{y}|\boldsymbol{z}_1) = \mathcal{N}(\boldsymbol{y} \mid \mathcal{H}\boldsymbol{z}_1, \sigma_{\boldsymbol{y}}^2 \mathbf{I})$, where $\mathcal{H}$ is a linear operator.

### A.5.3. MONTE CARLO GUIDANCE

On problems with smaller dimensions, we can develop an asymptotically exact guidance method using Monte Carlo integration. This follows from the following straightforward computation

$$\nabla_{\boldsymbol{z}_t} \log p(\boldsymbol{y}|\boldsymbol{z}_t) = \nabla_{\boldsymbol{z}_t} \log \left( \int_{\mathbb{R}^d} p(\boldsymbol{y}|\boldsymbol{z}_1) p(\boldsymbol{z}_1|\boldsymbol{z}_t) \mathrm{d}\boldsymbol{z}_1 \right) \tag{64}$$

$$= \nabla_{\boldsymbol{z}_t} \log \left( \int_{\mathbb{R}^d} p(\boldsymbol{y}|\boldsymbol{z}_1) \frac{p(\boldsymbol{z}_t|\boldsymbol{z}_1) p_1(\boldsymbol{z}_1)}{\int_{\mathbb{R}^d} p(\boldsymbol{z}_t|\boldsymbol{z}_1) p_1(\boldsymbol{z}_1) \mathrm{d}\boldsymbol{z}_1} \mathrm{d}\boldsymbol{z}_1 \right) \tag{65}$$

$$\stackrel{\mathrm{MC}}{\approx} \nabla_{\boldsymbol{z}_t} \log \left( \frac{\sum_{i=1}^{J} p(\boldsymbol{y}|\boldsymbol{z}_1^{(i)}) p(\boldsymbol{z}_t|\boldsymbol{z}_1^{(i)})}{\sum_{j=1}^{J} p(\boldsymbol{z}_t|\boldsymbol{z}_1^{(j)})} \right), \quad \text{for} \quad \boldsymbol{z}_1^{(i)}, \boldsymbol{z}_1^{(j)} \sim p_1. \tag{66}$$

Using the formulation of the stochastic interpolant, we have $p(\boldsymbol{z}_t|\boldsymbol{z}_1) = \mathcal{N}(\boldsymbol{z}_t|\beta_t \boldsymbol{z}_1, \alpha_t^2 I)$ and provided we have a closed form expression for $p(\boldsymbol{y}|\boldsymbol{z}_1)$, we can compute the log-density $\log p(\boldsymbol{y}|\boldsymbol{z}_t)$ via Monte Carlo and use automatic differentiation to calculate its $\boldsymbol{z}_t$-gradient.

### A.6. Rescaling of trained interpolant

Often, the model is trained using normalized data $\boldsymbol{w}_t = (\boldsymbol{z}_t - \mu)/\sigma$ to stabilize training. If we have an interpolant in the normalized space,

$$\boldsymbol{w}_t = \alpha_t \boldsymbol{\xi} + \beta_t \boldsymbol{w}_1, \quad \boldsymbol{\xi} \sim \mathcal{N}(\mathbf{0}, \mathbf{I}), \tag{67}$$

then, in the original space, we have the following un-normalized interpolant and its time derivative:

$$\boldsymbol{z}_t = \varphi(\boldsymbol{w}_t) := \sigma \boldsymbol{w}_t + \mu \tag{68}$$

$$= \sigma \alpha_t \boldsymbol{\xi} + \beta_t \boldsymbol{z}_1 + (1 - \beta_t)\mu, \tag{69}$$

$$\dot{\boldsymbol{z}}_t = \sigma \dot{\alpha}_t \boldsymbol{\xi} + \dot{\beta}_t \boldsymbol{z}_1 - \dot{\beta}_t \mu, \tag{70}$$

where $\boldsymbol{z}_1 = \sigma \boldsymbol{w}_1 + \mu$. Noting that

$$p(\boldsymbol{z}_t | \boldsymbol{z}_1) \overset{(69)}{=} \frac{1}{(2\pi\sigma^2\alpha_t^2)^{d/2}} \exp\left(-\frac{1}{2\sigma^2\alpha_t^2}\|(\boldsymbol{z}_t - \mu) - \beta_t(\boldsymbol{z}_1 - \mu)\|^2\right) \tag{71}$$

$$= \frac{1}{\sigma^d} \frac{1}{(2\pi\alpha_t^2)^{d/2}} \exp\left(-\frac{1}{2\alpha_t^2}\left\|\left(\frac{\boldsymbol{z}_t - \mu}{\sigma}\right) - \beta_t\left(\frac{\boldsymbol{z}_1 - \mu}{\sigma}\right)\right\|^2\right) \tag{72}$$

$$= \frac{1}{\sigma^d} \frac{1}{(2\pi\alpha_t^2)^{d/2}} \exp\left(-\frac{1}{2\alpha_t^2}\|\boldsymbol{w}_t - \beta_t \boldsymbol{w}_1\|^2\right) \tag{73}$$

$$\overset{(67)}{=} \frac{1}{\sigma^d} p(\boldsymbol{w}_t | \boldsymbol{w}_1) \tag{74}$$

and $\frac{\mathrm{d}\mathbb{P}_Z}{\mathrm{d}\mathbb{P}_W} = \det(D\varphi) = \sigma^d$, then by change of variables, we get

$$\boldsymbol{b}_Z(t, \boldsymbol{z}_t) = \int \dot{\boldsymbol{z}}_1 \mathbb{P}(\mathrm{d}\boldsymbol{z}_1 | \boldsymbol{z}_t) \tag{75}$$

$$= \frac{\int \dot{\boldsymbol{z}}_1 p(\boldsymbol{z}_t | \boldsymbol{z}_1) \mathbb{P}_Z(\mathrm{d}\boldsymbol{z}_1)}{\int p(\boldsymbol{z}_t | \boldsymbol{z}_1) \mathbb{P}_Z(\mathrm{d}\boldsymbol{z}_1)} \tag{76}$$

$$= \frac{\cancel{(\sigma^d/\sigma^d)} \int (\sigma \dot{\boldsymbol{w}}_1) p(\boldsymbol{w}_t | \boldsymbol{w}_1) \mathbb{P}_W(\mathrm{d}\boldsymbol{w}_1)}{\cancel{(\sigma^d/\sigma^d)} \int p(\boldsymbol{w}_t | \boldsymbol{w}_1) \mathbb{P}_W(\mathrm{d}\boldsymbol{w}_1)} \tag{77}$$

$$= \sigma \mathbb{E}[\dot{\boldsymbol{w}}_1 | \boldsymbol{w}_t] \tag{78}$$

$$= \sigma \boldsymbol{b}_W(t, (\boldsymbol{z}_t - \mu)/\sigma). \tag{79}$$

Similarly, for the score, we get

$$\boldsymbol{s}_Z(t, \boldsymbol{z}_t) = \nabla_{\boldsymbol{z}_t} \log p_Z(\boldsymbol{z}_t) \tag{80}$$

$$= D\varphi^{-1}(\boldsymbol{w}_t) \nabla_{\boldsymbol{w}_t}\left(\log \underbrace{\frac{\mathrm{d}\mathbb{P}_Z}{\mathrm{d}\mathbb{P}_W}(\boldsymbol{w}_t)}_{=\sigma^d} + \log p_W(\boldsymbol{w}_t)\right) \tag{81}$$

$$= \sigma^{-1} \nabla_{\boldsymbol{w}_t} \log p_W(\boldsymbol{w}_t) \tag{82}$$

$$= \sigma^{-1} \boldsymbol{s}_W(t, (\boldsymbol{z}_t - \mu)/\sigma). \tag{83}$$

From the relation (43), we can write down the scaled score in terms of the scaled drift as

$$\boldsymbol{s}_Z(t, \boldsymbol{z}_t) = \frac{\beta_t \boldsymbol{b}_Z(t, \boldsymbol{z}_t) - \dot{\beta}_t(\boldsymbol{z}_t - \mu)}{\sigma^2 \alpha_t \gamma_t}. \tag{84}$$

Similarly, we can express $\mathbb{E}[\boldsymbol{z}_1 | \boldsymbol{z}_t]$ and $\mathbb{E}[\boldsymbol{\xi} | \boldsymbol{z}_t]$ in terms of $\boldsymbol{b}_W$:

$$\mathbb{E}[\boldsymbol{z}_1 \mid \boldsymbol{z}_t] = \mu + \sigma \frac{\alpha_t \boldsymbol{b}_W(t, \frac{\boldsymbol{z}_t - \mu}{\sigma}) - \dot{\alpha}_t \frac{\boldsymbol{z}_t - \mu}{\sigma}}{\gamma_t}, \tag{85}$$

$$\mathbb{E}[\boldsymbol{\xi} \mid \boldsymbol{z}_t] = -\frac{\beta_t \boldsymbol{b}_W(t, \frac{\boldsymbol{z}_t - \mu}{\sigma}) - \dot{\beta}_t \frac{\boldsymbol{z}_t - \mu}{\sigma}}{\gamma_t} \tag{86}$$

### A.7. Probability flow ODE

Diffusion models work with SDEs instead of ODEs but can still be used within our framework through the so-called *Probability Flow ODE* (Song et al., 2021), which we demonstrate here.

Consider a forward (noising) SDE

$$\mathrm{d}\boldsymbol{z}_t = f(t, \boldsymbol{z}_t)\mathrm{d}t + g(t)\mathrm{d}W_t. \tag{87}$$

This induces a reverse-time SDE given by

$$\mathrm{d}\boldsymbol{z}_t = (f(t, \boldsymbol{z}_t) - g(t)^2 \boldsymbol{s}(t, \boldsymbol{z}_t))\mathrm{d}t + g(t)\mathrm{d}W_t \tag{88}$$

which, after learning $\boldsymbol{s}$, can be used to generate samples. The probability flow ODE sharing the same marginals as this SDE is given by

$$\frac{\mathrm{d}\boldsymbol{z}_t}{\mathrm{d}t} = f(t, \boldsymbol{z}_t) - \frac{1}{2}g(t)^2 \boldsymbol{s}(t, \boldsymbol{z}_t). \tag{89}$$

With the same argument as for the stochastic interpolant, this shares the same marginals as

$$\mathrm{d}\boldsymbol{z}_t = \left( f(t, \boldsymbol{z}_t) + \left( \epsilon_t - \frac{1}{2}g(t)^2 \right) \boldsymbol{s}(t, \boldsymbol{z}_t) \right) \mathrm{d}t + \sqrt{2\epsilon_t}\mathrm{d}W_t. \tag{90}$$

## B. Model Details

In this work, we make the standard choice of a linear scheduler $\alpha_t = 1 - t, \beta_t = t$, which implies a cross-term $\gamma_t = 1$ and the guided drift scaling $\lambda_t = (1 - t)/t$. This gives the score

$$\boldsymbol{s}(t, \boldsymbol{z}_t) = \frac{t\boldsymbol{b}(t, \boldsymbol{z}_t) - \boldsymbol{z}_t}{1 - t}, \tag{91}$$

and the expectation

$$\mathbb{E}[\boldsymbol{z}_1 | \boldsymbol{z}_t] = \boldsymbol{z}_t + (1 - t)\boldsymbol{b}(t, \boldsymbol{z}_t). \tag{92}$$

We note that this choice of schedule leads to an expectation given by a single Euler step to the final time. We also note that the score diverges as $t \to 1$. To avoid numerical issues, we let $\epsilon_t = \epsilon(1 - t)$ for some $\epsilon \geq 0$. To solve the SDE, we use Euler–Maruyama; we do not use any additional Langevin correction steps, as these did not improve results.

## C. Experimental details

### C.1. Metrics

Given an ensemble of assimilated states $\{\widehat{\boldsymbol{x}}_n^{(j)}\}_{j=1}^J$, and a true state $\boldsymbol{x}_n$ at step $n$ we define the Root Mean Squared Error (RMSE) as

$$\mathrm{RMSE}_n = \sqrt{\langle (\bar{\boldsymbol{x}}_n - \boldsymbol{x}_n)^2 \rangle}, \tag{93}$$

where $\langle \cdot \rangle$ denotes the averaging over variable and spatial dimensions and

$$\bar{\boldsymbol{x}}_n = \frac{1}{J} \sum_{j=1}^J \widehat{\boldsymbol{x}}_n^{(j)} \tag{94}$$

is the ensemble mean of the assimilated state. We also evaluate the RMSE of each ensemble member $j$

$$\mathrm{RMSE}_{n,j} = \sqrt{\langle (\widehat{\boldsymbol{x}}_n^{(j)} - \boldsymbol{x}_n)^2 \rangle}. \tag{95}$$

To measure the calibration of the ensemble forecasts, we measure the Continuous Ranked Probability Score (CRPS) (Gneiting & Raftery, 2007). We follow and compute the fair unbiased CRPS estimate (Ferro, 2014; Zamo & Naveau, 2018) for our ensemble, which reads

$$\mathrm{CRPS}_n = \frac{1}{J} \sum_{j=1}^J ||\widehat{\boldsymbol{x}}_n^{(j)} - \boldsymbol{x}_n||_{L_1} - \frac{1}{2J(J-1)} \sum_{j=1}^J \sum_{j^*=1}^J ||\widehat{\boldsymbol{x}}_n^{(j)} - \widehat{\boldsymbol{x}}_n^{(j^*)}||_{L_1}. \tag{96}$$

Additionally, we measure the Spread Skill Ratio (SSR) to evaluate the ensemble calibration, where a well calibrated ensemble should have SSR $\approx 1$ (Fortin et al., 2014). This is defined as

$$\text{SSR}_n = \sqrt{\frac{J+1}{J}} \frac{\text{Spread}_n}{\text{RMSE}_n}, \tag{97}$$

where

$$\text{Spread}_n = \sqrt{\langle (\widehat{\boldsymbol{x}}_n^{(j)} - \bar{\boldsymbol{x}}_n)^2 \rangle}. \tag{98}$$

In addition to ensuring that the ensemble mean and individual members are accurate and that the ensemble is well calibrated, we also require each ensemble member to be physically realistic and to reproduce the energy spectrum of the ground truth. To evaluate this, we compute the Log Spectral Distance (LSD). Given a field $X$ defined on a 2D grid, we define its 2D power spectrum as

$$P_X(\mathbf{k}) = |\mathcal{F}\{X\}(\mathbf{k})|^2, \tag{99}$$

where $\mathcal{F}\{\cdot\}$ denotes the 2D Fourier transform and $\mathbf{k} = (k_x, k_y)$ is the wavenumber vector. We obtain an isotropic (radially averaged) power spectrum by averaging over all Fourier coefficients with the same radial wavenumber $r = \|\mathbf{k}\|$:

$$\bar{P}_X(r) = \frac{1}{N(r)} \sum_{\|\mathbf{k}\|=r} P_X(\mathbf{k}), \tag{100}$$

where $N(r)$ is the number of coefficients at radius $r$.

The Log Spectral Distance (LSD) between a forecast field $\hat{X}$ and the ground truth $X$ at time $t$ is then defined as

$$\text{LSD}^t = \sqrt{\frac{1}{R} \sum_{r=1}^{R} \left( \log\big(\bar{P}_{\hat{X}}(r) + \varepsilon\big) - \log\big(\bar{P}_X(r) + \varepsilon\big) \right)^2}, \tag{101}$$

where $R$ is the maximum resolved radial wavenumber and $\varepsilon$ is a small constant included for numerical stability.

When we have access to samples $\{\boldsymbol{x}_i\}_{i=1}^N$, $\{\boldsymbol{y}_j\}_{j=1}^M$ from two measures $\pi_1$ and $\pi_2$, respectively, we can compute a particular notion of distance between them, using the Maximum Mean Discrepancy (MMD) (Gretton et al., 2012), defined as the distance between the kernel embeddings of the two measures in the corresponding reproducing kernel Hilbert space (RKHS) $\mathcal{H}$. In practice, given a kernel $k(\cdot, \cdot) : \mathbb{R}^d \times \mathbb{R}^d \to \mathbb{R}$, this is computed as

$$\text{MMD}^2(\pi_1, \pi_2) \approx \left\| \frac{1}{N} \sum_{i=1}^N k(\boldsymbol{x}_i, \cdot) - \frac{1}{M} \sum_{j=1}^M k(\boldsymbol{y}_j, \cdot) \right\|_{\mathcal{H}}^2 \tag{102}$$

$$= \frac{1}{N(N-1)} \sum_{i=1}^N \sum_{j \neq i}^N k(\boldsymbol{x}_i, \boldsymbol{x}_j) - \frac{2}{NM} \sum_{i=1}^N \sum_{j=1}^M k(\boldsymbol{x}_i, \boldsymbol{y}_j) + \frac{1}{M(M-1)} \sum_{i=1}^M \sum_{j=1}^M k(\boldsymbol{y}_i, \boldsymbol{y}_j). \tag{103}$$

For the choice of $k(\cdot, \cdot)$, we use the squared-exponential kernel $k(\boldsymbol{x}, \boldsymbol{y}) = \exp\left(-\|\boldsymbol{x} - \boldsymbol{y}\|^2 / 2\sigma^2\right)$, where the bandwidth $\sigma^2$ is chosen to be the median heuristic $\sigma^2 = \text{Median}(\{\|\boldsymbol{x}_i - \boldsymbol{y}_j\|\}_{i,j})/2$.

### C.2. 1D Gaussian Mixture

In our toy 1D experiment, we considered an artificial setup whereby the invariant measure is given by a 1D Gaussian mixture $\mathbb{P}_\infty(\cdot) = \sum_{k=1}^K \phi_k \mathcal{N}(\cdot | \mu_k, \sigma_k^2)$ with $K = 3$ and

$$(\phi_1, \phi_2, \phi_3) = (0.5, 0.3, 0.2), \qquad (\mu_1, \mu_2, \mu_3) = (0.0, 3.0, -2.0), \qquad (\sigma_1, \sigma_2, \sigma_3) = (1.0, 0.5, 0.8). \tag{104}$$

We trained a stochastic interpolant model on $50,000$ i.i.d. samples of this Gaussian mixture model, where we used a standard MLP with one hidden layer of size 64 and ReLU activations to parameterize the drift $\boldsymbol{b}(\boldsymbol{x}, t)$.

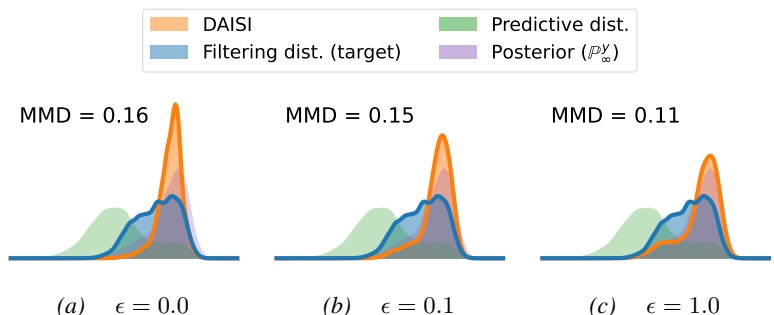

*Figure 8.* Ablation with respect to the $\epsilon$ hyperparameter in the 1D Gaussian mixture experiment, fixing $t_{\min} = 0.01$. The distribution obtained by DAISI when $\epsilon = 0$ is highly peaked, making the resulting distribution overconfident. By increasing $\epsilon$, it loses information about the predictive ensemble and pulls the distribution towards $\mathbb{P}_\infty^y$, which can help to rejuvenate sample variance.

We use this toy experiment to understand the effect of the inherent error of DAIS on sampling from the filtering distribution, as highlighted in Section 3.1, and to investigate the impact of the hyperparameters $t_{\min}$ and $\epsilon$ on reducing this error.

To this end, we simulate the analysis step of DAISI by artificially constructing a predictive distribution $\hat{\pi} \propto f \, \mathbb{P}_\infty$, where we took $f(x) = \exp\left(-\frac{1}{2\sigma^2}|x - \mu|^2\right)$ with $\mu = 0.5$ and $\sigma = 1.5$. We obtained samples $\{\hat{x}_i\}_{i=1}^N$ from $\hat{\pi}$ via samples $\{x_i\}_{i=1}^N$ of the Gaussian mixture $\mathbb{P}_\infty$ by first re-weighting the particles by $f$, then re-sampling according to the weights, i.e.,

1. Compute weights $w_i = f(x_i)$ for all $i = 1, \ldots, N$,

2. Resample particles $\hat{x}_i = x_{j_i}$, where $j_i \sim \text{Multinomial}(w_1, ..., w_N)$ for $i = 1, \ldots, N$.

Then, starting from the particles $\{\hat{x}_i\}_{i=1}^N$, we applied the analysis step of DAISI to obtain the particles $\{x_i^y\}_{i=1}^N$ post-assimilation, where, for the observation model, we took $p(y|x) = \mathcal{N}(y|x, \sigma_{\text{obs}}^2)$ with $y = 2.5$ and $\sigma_{\text{obs}} = 1.0$. For the ensemble size, we set $N = 10,000$ to accurately capture the resulting distributions. For the guidance method, we considered the asymptotically exact Monte Carlo strategy (see Section A.5.3) to isolate the sources of error as arising solely from the inexactness of DAISI (there will still be errors arising from numerical discretization and Monte Carlo estimation; however, these errors can be made arbitrarily small). For the numerical discretisation of the interpolant ODE and SDE, we used Euler-Maruyama integration with 200 time steps and for the Monte-Carlo integration used in guidance, we used $J = 10,000$ particles. As a point of reference to compare against, we also obtain samples directly from the filtering distribution $\pi \propto p(y|\cdot)\hat{\pi}$ by a similar reweight-resample strategy used before to obtain samples from $\hat{\pi}$.

We simulated the analysis step of DAISI with various combinations of hyperparameters $t_{\min}$ and $\epsilon$ and display the heatmap of the MMD between samples of the filtering distribution $\pi$ and samples of the distribution $\pi^{\text{DAISI}}$ obtained by DAISI in Figure 9. We see that the best result is achieved by the combination $t_{\min} = 0.3$ and $\epsilon = 0.1$, which yields an MMD of 0.004, indicating a very close match with the true filtering distribution. We also display ablations with respect to the individual hyperparameters in Figure 2 (for ablation with respect to $t_{\min}$, with $\epsilon$ fixed to 0) and Figure 8 (for ablation with respect to $\epsilon$, with $t_{\min}$ fixed to 0.01). The figures demonstrate that the "default" setting of $\epsilon = 0$ and $t_{\min} \approx 0$ yields a distribution $\pi^{\text{DAISI}}$ that is much more "peaked" and therefore mismatched from the filter distribution $\pi$. This error can be reduced by increasing $t_{\min}$, which pulls the profile of $\pi^{\text{DAISI}}$ towards the predictive distribution $\hat{\pi}$, while increasing $\epsilon$ pulls it towards the posterior $\mathbb{P}_\infty^y$. Since the filtering distribution $\pi \propto f \mathbb{P}_\infty^y = p(y|\cdot)\hat{\pi}$ has elements of both $\mathbb{P}_\infty^y$ and $\hat{\pi}$, we find that a slight nudge in these directions can help to match $\pi$ better.

### C.3. Lorenz '63

The Lorenz '63 (L63) model (Lorenz, 1963) is given by the following system of ODEs

$$\frac{\mathrm{d}x}{\mathrm{d}t} = \sigma(y - z) \tag{105}$$

$$\frac{\mathrm{d}y}{\mathrm{d}t} = x(\rho - z) - y \tag{106}$$

$$\frac{\mathrm{d}z}{\mathrm{d}t} = xy - \beta z \tag{107}$$

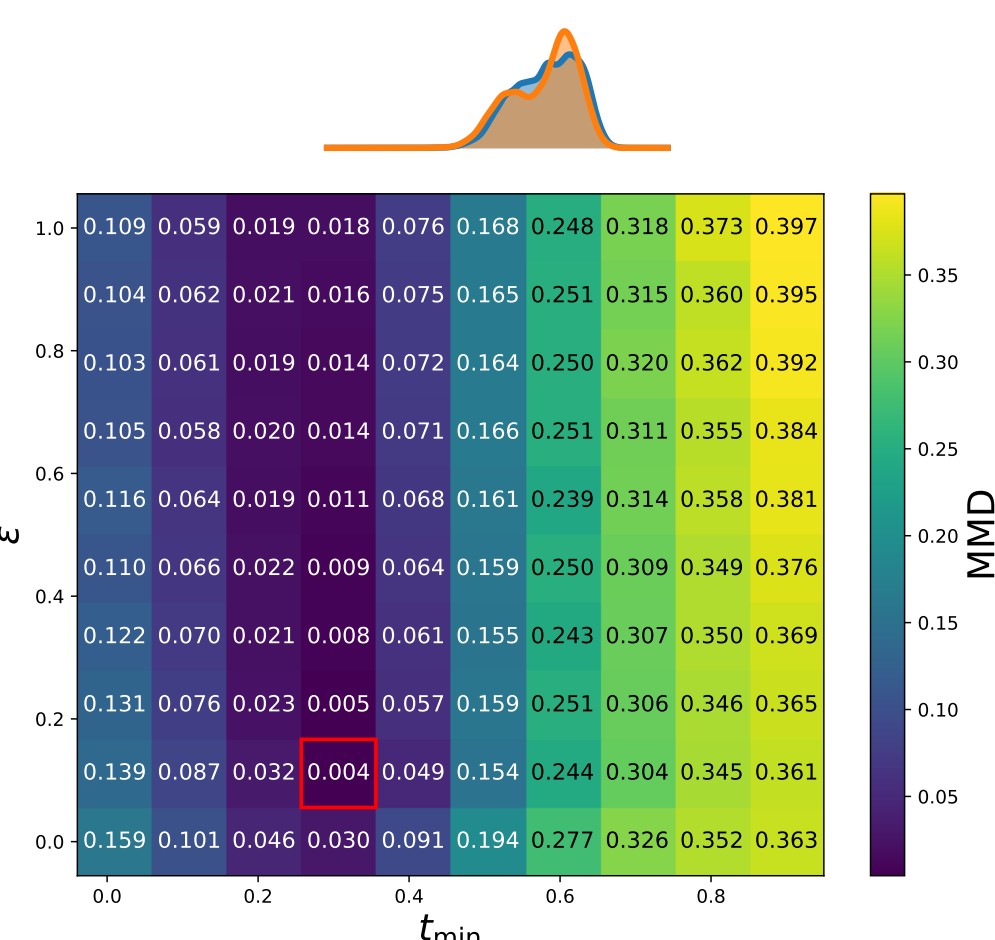

*Figure 9.* Heatmap of the maximum mean discrepancy (MMD) between the true filtering distribution $\pi$ and the DAISI analysis $\pi^{\text{DAISI}}$ using different combinations of $t_{\min}$ and $\epsilon$. The highlighted square corresponds to the $(t_{\min}, \epsilon)$-combination that yielded the lowest MMD. The corresponding distribution obtained by DAISI is plotted above (orange) against the true filtering distribution (blue). We observe a close match between the two distributions under the optimal hyperparameters.

with the parameters set to $\sigma = 10$, $\rho = 28$ and $\beta = \frac{8}{3}$. In our experiments in Section 4.1, we integrate this system forward in time using a fourth order Runge-Kutta solver with time step $\Delta t = 0.01$.

### C.3.1. MODEL TRAINING

To train the stochastic interpolant model, we generate training data $\{x_i\}_{i=0}^{N-1}$ by integrating the L63 ODE (105)–(107) starting from the initial condition $x_0 = (0, 1, 1.05)$ with $N = 10^6$. We consider an 80-20 split for training and validation. Both datasets are normalized using the empirical mean and standard deviation of the training portion:

$$w_i = \frac{x_i - \mu_{\text{train}}}{\sigma_{\text{train}}}. \tag{108}$$

To parameterize the drift $b(w, t)$, we use an MLP with two hidden layers; each with 128 neurons:

$$(w, t) \in \mathbb{R}^4 \mapsto 128 \mapsto 128 \mapsto 3,$$

and using ReLU activations. We train for 20 epochs using the Adam optimizer with learning rate $10^{-4}$ and batch size 64.

### C.3.2. DATA ASSIMILATION SETUP

To perform data assimilation with the L63 model, we generate scalar observations at each time step

$$y_n = Hx_n + \eta_n, \qquad H = \begin{bmatrix} 1 & 0 & 0 \end{bmatrix}, \qquad \eta_n \sim \mathcal{N}(0, \sigma_{\text{obs}}^2), \quad \sigma_{\text{obs}} = 5. \tag{109}$$

At initialization, we sample $J$ particles from a Gaussian ball around the ground truth,

$$x_0^{(j)} = x_0 + \sigma_{\text{init}}\, \xi^{(j)}, \quad \xi^{(j)} \sim \mathcal{N}(\mathbf{0}, \mathbf{I}), \quad \sigma_{\text{init}} = 5, \quad j = 1, \dots, J. \tag{110}$$

### C.3.3. HYPERPARAMETER TUNING

To tune the hyperparameters $t_{\min}$ and $\epsilon$ of DAISI, we use Bayesian optimization to minimize the CRPS on a separate trajectory, which we generate from the initial condition $x_0 = (0, 1, 1.05)$ and integrate for 5000 steps. We use DAISI with fixed values of $t_{\min}$ and $\epsilon$ and ensemble size $J = 100$, to assimilate data on the last 200 steps of the generated trajectory (this is to ensure the dynamics has reached statistical equilibrium), and evaluate its CRPS averaged across the last 100 steps (this is to ensure that the filter has stabilized). For Bayesian optimization, we used the following hyperpriors:

$$\epsilon \sim \text{LogUniform}(10^{-2}, 1), \qquad t_{\min} \sim \text{Uniform}(10^{-2}, 0.99), \tag{111}$$

and ran until convergence was observed.

### C.3.4. EVALUATION

After tuning the values for $t_{\min}$ and $\epsilon$, we evaluated DAISI on ten different trajectories starting from random initial conditions

$$x_0^{(i)} \sim \mathcal{N}(\mu_{\text{train}}, \sigma_{\text{train}}^2 \mathbf{I}), \quad i = 1, \dots, 10, \tag{112}$$

and integrated for 5000 time steps. We perform data assimilation with DAISI on the last 500 steps of the generated trajectories, and evaluated the RMSE, CRPS and SSR, averaged across the last 100 steps of the assimilation window. As with the previous example, we used the Monte Carlo strategy for guidance with $J = 10,000$ particles. For the bootstrap particle filter (BPF) baseline, we used $N_p = 10,000$ particles to ensure high accuracy of the obtained results.

### C.3.5. ABLATION

In Figures 10b–10e, we plot the results of filtering using DAISI under different $(t_{\min}, \epsilon)$-combinations. For reference, we also plot the result when no inverse sampling is performed in Figure 10a (i.e., it just samples from the posterior $\mathbb{P}_\infty^{\boldsymbol{y}}$ at every time step). This shows the importance of the inverse sampling step in DAISI to firstly, reduce the excessive sample variance when just using $\mathbb{P}_\infty^{\boldsymbol{y}}$ for assimilation, and secondly, for establishing time-continuity of the filter.

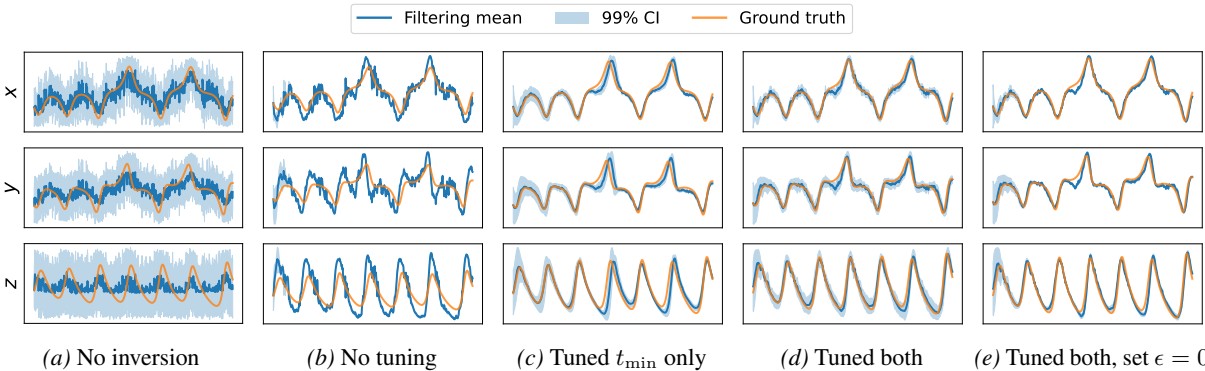

*Figure 10.* Filtering results on the L63 system with DAISI under various hyperparameter settings.

In Figure 10b, we plot the results of DAISI with the default setting $t_{\min} = 0.01$ and $\epsilon = 0$. We see that while the filter is able to roughly track the overall pattern of the ground truth, the result is not very accurate, with narrow uncertainty bars that do not cover the ground truth consistently. This demonstrates the importance of tuning the hyperparameters $t_{\min}$ and $\epsilon$ to obtain an accurate filter. For example, Figure 10c displays the result of when just $t_{\min}$ is tuned, with $\epsilon$ fixed to 0 and Figure 10d displays the result when both hyperparameters are tuned. Both results show higher accuracy than the filter in Figure 10b, highlighting the importance of tuning $t_{\min}$. Furthermore, the result in Figure 10d is more accurate than that of Figure 10c, which shows how tuning $\epsilon$ can lead to further improvements in performance. Finally, in Figure 10e, we display the results where we use the same value of $t_{\min}$ as used in Figure 10d (see Table 4), but setting $\epsilon$ to 0 to better observe the effect of $\epsilon$ on the filter. We clearly see that when $\epsilon$ is set to 0, the uncertainty bars become visibly narrower, making the predictions overconfident.

|  | $t_{\min}$ | $\epsilon$ |
|---|---|---|
| Figure 10c | 0.75 | – |
| Figure 10d | 0.65 | 0.15 |

*Table 4.* Tuned hyperparameters used in the L63 experiments.

## C.4. Surface Quasi-Geostrophic (SQG)

For the SQG experiments, we use the surface quasi-geostrophic model presented in (Tulloch & Smith, 2009). This formulation evolves potential temperature at both the lower boundary ($z = 0$) and an upper lid ($z = H$), capturing the influence of surface and tropopause-level dynamics on the interior flow. The three-dimensional geostrophic velocity field is diagnosed via a nonlocal spectral inversion, which couples the boundaries through stratified Green's functions. Compared to classical SQG, which assumes decay into the deep interior ($z \to \infty$), the two-layer model confines dynamics to a finite slab and supports richer vertical structure, including baroclinic interactions. The corresponding PDE is given by Equation (113).

$$\underbrace{\frac{\partial \theta}{\partial t}}_{\text{time tendency}} = \underbrace{-J(\psi, \theta)}_{\text{nonlinear advection}} + \underbrace{\frac{1}{t_{\text{diab}}}(\bar{\theta} - \theta)}_{\text{thermal relaxation}} + \underbrace{r\nabla^2\psi}_{\text{Ekman damping}} \underbrace{- \nu(-\nabla^2)^{n/2}\theta}_{\text{hyperdiffusion}}, \tag{113}$$

The nonlinear advection term represents the advection of $\theta$ by the geostrophic velocity field and is responsible for vortex formation and turbulent cascades, serving as the main nonlinear mechanism in the dynamics. The thermal relaxation term acts as a large-scale restoring forcing toward a prescribed equilibrium profile $\bar{\theta}$ corresponding to a background jet. This term maintains a statistically steady turbulent state by continuously driving the system. The Ekman damping term represents frictional interaction with a boundary layer and damps the streamfunction, dissipating energy primarily at large scales. Following (Liang et al., 2025), we set this term to zero in our experiments. The hyperdiffusion term is a high-order dissipation operator that selectively damps the smallest resolved scales. It does not correspond to physical diffusion but is introduced for numerical stability to prevent energy accumulation at the grid scale due to the turbulent cascade.

The equations are solved numerically by first applying a fast Fourier transform (FFT) to map model variables to spectral

*Table 5.* Optimizer Hyperparameters.

| Optimizer hyperparameters | |
| --- | --- |
| Optimiser | AdamW (Loshchilov & Hutter, 2017a) |
| Initialization | Xavier Uniform (Glorot & Bengio, 2010) |
| LR decay schedule | Cosine (Loshchilov & Hutter, 2017b) |
| Peak LR | 1e-3 |
| Weight decay | 1e-4 |
| Warmup steps | 1e3 |
| Epochs | 50 |
| Batch size | 200 (SQG 64), 2 (SQG 256), 10 (SEVIR) |

space. They are then integrated forward with a fourth order Runge-Kutta solver that uses a 2/3 dealiasing rule and implicit treatment of hyperdiffusion. For more details, we refer to the GitHub repository of the model (Whitaker, 2025).

### C.4.1. MODEL TRAINING

For the $64 \times 64$ experiments, we generate 2000 training trajectories of 100 steps each at 3-hour intervals. Evaluation is done on 10 trajectories of 100 steps, and metrics are averaged over the final 20 steps. For the $256 \times 256$ experiment, we generate 10 trajectories of length 1000 steps, but due to computational cost, evaluation is performed on a single 200-step run. All trajectories are initialized from approximate stationarity by spinning up the model for 300 days from random initial conditions. The data already has mean zero, but we normalize it with the standard deviation $\sigma_{\text{train}} = 2660$.

Our backbone model for learning the drift $b$ is a modified U-Net (Ronneberger et al., 2015) following the design of Song et al. (2021) and Karras et al. (2022) with 3.5M parameters. Since the SQG data is 2D periodic, we use circular padding to preserve spatial continuity. The same network configuration is used for both the $64 \times 64$ and $256 \times 256$ experiments, but each model is trained independently. The U-Net employs a hidden dimension of 32 throughout and consists of three hierarchical levels, with attention applied at the second level. Although we did not explore more advanced architectures, our framework is compatible with any model capable of learning a flow-matching objective.

The training process is executed in Pytorch, with setup and parameters detailed in Table 5.

### C.4.2. DATA ASSIMILATION SETUP

We perform data assimilation for all experiment setups in Table 2. The observations are given by

$$y_n = \mathcal{H}(x_n) + \eta_n, \qquad \mathcal{H}(x_n) = (A \circ h)(x_n), \qquad \eta_n \sim \mathcal{N}(0, \sigma_{\text{obs}}^2 I), \tag{114}$$

where $h$ is the operator in Table 2 and $A$ is the sparsity operator. The observation locations in $A$ are chosen uniformly at random before assimilation and remain the same across time and members and channels.

For the `High-dim.` experiments on $256 \times 256$, observations are generated through a three-step process. The state is first subsampled to $64 \times 64$, after which each grid point is replaced by the average over a $5 \times 5$ neighborhood, mimicking lower-resolution sensing. Finally, only 5% of these pixels are observed at random with added noise, yielding highly sparse observations.

**Options for initializing DAISI.** There are several options for initializing DAISI, depending on the available initial conditions. In our numerical experiments, assimilation begins from a noisy version of the true initial state $\tilde{x}_0 \sim \mathcal{N}(x_0, \sigma_{\text{init}}^2 I)$. This perturbation is needed for LETKF to work, and is thus also used by DAISI to ensure fairness. In particular, this fairness is important for the squared observations, where the initial conditions affect how multimodal the filtering becomes.

Since these initial conditions will have unphysical noise, applying DAISI directly leads to unrealistic samples for the first few iterations. To address this, we perform an additional sampling step, running the forward SDE starting in the latent $z_{t^*} = \beta_{t^*} \tilde{x}_0$, where $t^* \in [0, 1]$ is chosen such that $\alpha_{t^*}/\beta_{t^*} = \sigma_{\text{init}}$. Conceptually, this treats the noisy initial condition as a partially inverted sample and lets DAISI remove the noise. We remark that this is an experimental detail and not a part of DAISI. If ones knowledge of the initial condition was an observation $y_0$, one could start the assimilation by conditionally sampling from $\mathbb{P}_\infty^{y_0}$.

This initialization procedure is reminiscent of SDEdit (Meng et al., 2022), in which a sample is partially inverted by the stochastic interpolant $z_t = \alpha_t z_0 + \beta_t z_1$. This has been used for image editing, and in Table 14, we perform an ablation where it replaces the backward SDE in the inversion step.

### C.4.3. BASELINES

We compare DAISI against both classical and ML-based DA methods. Classical baselines include the Local Ensemble Transform Kalman Filter (LETKF), while ML baselines include Score-based Data Assimilation (SDA), FlowDAS, and the Ensemble Score Filter (EnSF).

**SDA (smoothing):** The score-based data assimilation (SDA) algorithm of (Rozet & Louppe, 2023) uses a diffusion model trained on a short window of a dynamical system's trajectory, referred to as the *Markov blanket*, to sample trajectories from the smoothing distribution $p(\boldsymbol{x}_{1:N}|\boldsymbol{y}_{1:N})$ of an arbitrary length $N$, using a guidance-based method for posterior sampling. To be consistent with the other baselines, we also conditioned on noised initial states $\tilde{\boldsymbol{x}}_0^{(j)}$ as described in Section C.4.2, yielding samples from the distribution $p(\boldsymbol{x}_{0:N}|\boldsymbol{y}_{1:N}, \tilde{\boldsymbol{x}}_0^{(j)})$. For the window length $W$ of the Markov blanket, we set $W = 5$ in all of our experiments. For the score network, we used the default U-net architecture found in the original SDA github repository (`https://github.com/francois-rozet/sda`) and trained the model for 4096 epochs using the AdamW optimizer, with learning rate $10^{-3}$, weight decay $10^{-3}$ and linear training scheduler. We use the optimal hyperparameters in Table 6, obtained by grid search.

We note, however, that SDA is a *smoothing algorithm* and therefore does not provide a completely fair comparison with DAISI and the other baselines, which are essentially filtering algorithms. For this purpose, we also propose a filtering variant of SDA that we describe in the following.

*Table 6.* Hyperparameter configurations for SDA smoothing.

| Experiment | Guidance method | Guidance strength $\zeta$ | Euler steps | Correction steps |
|---|---|---|---|---|
| Noisy | MMPS | 1 | 100 | 1 |
| Sparse | MMPS | 1 | 500 | 1 |
| Multimodal | MMPS | 10 | 250 | 2 |
| Saturating | MMPS | 5 | 100 | 2 |
| SEVIR | MMPS | 1 | 500 | 2 |

**SDA (filtering):** In order to perform approximate filtering using the spatio-temporal diffusion model trained for SDA with window size $W$, we propose to iteratively sample $\boldsymbol{x}_{n:n+W-1}^{(j)} \sim p(\boldsymbol{x}_{n:n+W-1}|\boldsymbol{y}_{n+1:n+W-1}, \tilde{\boldsymbol{x}}_n^{(j)})$ for $n = 0, \ldots, N-W+1$ and $j = 1, \ldots, J$, storing $\boldsymbol{x}_{n+W-1}^{(j)}$ as the filtered state at time step $n + W - 1$ and setting $\tilde{\boldsymbol{x}}_{n+1}^{(j)} \leftarrow \boldsymbol{x}_{n+1}^{(j)}$ for the "initial condition" in the next iteration. We summarize this in Algorithm 2. We use a window size $W = 5$ for all of our experiments and the same the score network as used in the smoothing variant of SDA. The hyperparameters used are displayed in Table 7. We also note that this setup is similar to the joint guided autoregressive model proposed in (Shysheya et al., 2024).

*Table 7.* Hyperparameter configurations for SDA filtering.

| Experiment | Guidance method | Guidance strength $\zeta$ | Euler steps | Correction steps |
|---|---|---|---|---|
| Noisy | MMPS | 1 | 100 | 1 |
| Sparse | MMPS | 1 | 100 | 1 |
| Multimodal | MMPS | 5 | 100 | 1 |
| Saturating | MMPS | 5 | 100 | 1 |
| SEVIR | MMPS | 1 | 100 | 1 |

**EnSF:** We used the Ensemble Score Filter (EnSF) as described in (Liang et al., 2025) with $\epsilon_\alpha = 0.05$ and 1000 Euler steps. We note that the EnSF implementation includes an inflation step not mentioned in the original article, which we found

---

**Algorithm 2** SDA filtering

---

1: `Inputs`: Initial ensemble $\tilde{x}_0^{(j)} \sim \mathcal{N}(\boldsymbol{x}_0, \sigma_{\text{init}}^2 \mathbf{I})$ for $j = 1, \ldots, J$, observations $\{\boldsymbol{y}_n\}_{n=1}^N$, and $A = \varnothing$
2: **for** $n = 0, \ldots, N - W + 1$ **do**
3:     Initialize $B_n = \varnothing$
4:     **for** $j = 1, \ldots, J$ **do**
5:         Sample $\boldsymbol{x}_{n:n+W-1}^{(j)} \sim p(\boldsymbol{x}_{n:n+W-1}|\boldsymbol{y}_{n+1:n+W-1}, \tilde{\boldsymbol{x}}_n^{(j)})$ via guidance
6:         Append last state $\boldsymbol{x}_{n+W-1}^{(j)}$ to $B_n$
7:         Set $\tilde{\boldsymbol{x}}_{n+1}^{(j)} \leftarrow \boldsymbol{x}_{n+1}^{(j)}$
8:     **end for**
9:     Append $B_n$ to $A$
10: **end for**
11: `Output`: Sequence of SDA filtered states $A = \{\{\boldsymbol{x}_n^{(j)}\}_{j=1}^J\}_{n=W-1}^N$

---

crucial to prevent mode collapse. After each assimlation cycle, the particles are updated as

$$\tilde{\boldsymbol{x}}_n^{(j)} = \bar{\boldsymbol{x}}_n + \frac{\sigma_{\text{init}}}{\sigma_{\boldsymbol{x}_n}}(\boldsymbol{x}_n^{(j)} - \bar{\boldsymbol{x}}_n), \tag{115}$$

where $\bar{\boldsymbol{x}}_n$ is the ensemble mean, $\sigma_{\text{init}}$ is the initial ensemble standard deviation and $\sigma_{\boldsymbol{x}_n}$ is the current ensemble standard deviation, averaged over all grid points.

**FlowDAS:** We trained a conditional stochastic interpolant model for probabilistic forecasting (Chen et al., 2024), as used in FlowDAS, to emulate the forward dynamics of SQG at $64 \times 64$ resolution. Our model conditions on the past six states for roll-out, i.e., generates samples from $p(\boldsymbol{x}_n|\boldsymbol{x}_{n-1}, \ldots, \boldsymbol{x}_{n-6})$. For the drift model $\boldsymbol{b}$, we used a U-Net, similar to the architecture used for the interpolant in DAISI, with 64 channels, and consisting of three hierarchical levels with channel multipliers $(1, 2, 2)$. Each level uses group-normalized residual blocks with 8 groups, and time embeddings are provided through learned sinusoidal features of dimension 32. Linear self-attention is applied at every resolution, together with a full multi-head self-attention block applied at the bottleneck between the encoder and decoder. The model is trained using AdamW with learning rate $2 \times 10^{-4}$, cosine learning-rate decay, gradient-norm clipping, batch size 32, and dataset normalization with standard deviation $\sigma_{\text{train}} = 2660$. Training is performed for 4096 epochs on 3-hourly SQG data.

For sampling, we use the parameters in Table 8, which are based on (Chen et al., 2025) and finetuned for the different cases.

*Table 8.* Hyperparameter configurations for FlowDAS.

| Experiment | Guidance Strength $\zeta$ | Euler steps | MC guidance members $J$ |
|---|---|---|---|
| Noisy | 1 | 500 | 25 |
| Sparse | 3 | 500 | 25 |
| Multimodal | 0.5 | 500 | 25 |
| Saturating | 5 | 500 | 25 |
| SEVIR | 0.1 | 500 | 25 |

**LETKF:** LETKF is a state-of-the-art DA method commonly used in the geosciences (Wang et al., 2021). The hyperparameters used for LETKF are shown in Table 9. These are roughly based on the ones in (Liang et al., 2025) but finetuned for the different cases. For the high-dimensional case, no parameter search is done due to computational cost.

C.4.4. HYPERPARAMETERS

To identify suitable hyperparameters for DAISI, we conducted a greedy grid search across all experiments. The final hyperparameter settings for each experiment are listed in Table 10.

*Table 9.* Hyperparameter configurations for LETKF.

| Experiment | Localization scale | Relaxation to prior spread |
|---|---|---|
| Noisy | 2500 km | 0.5 |
| Noisy 12h | 1500 km | 0.4 |
| Sparse | 1500 km | 0.5 |
| Multimodal | 1500 km | 0.5 |
| Saturating | 4500 km | 0.9 |
| High-dim. | 1500 km | 0.5 |
| Non-stationary | 1000 km | 0.2 |
| SEVIR | 3 px | 0.0 |

*Table 10.* Hyperparameter configurations for all SQG and SEVIR experiments.

| Experiment | Guide Method | Solver Steps | $\varepsilon$ | $t_{\min}$ | Guidance Strength $\zeta$ |
|---|---|---|---|---|---|
| Noisy | DPS | 100 | 0.03 | 0.4 | 10 |
| Noisy 12h | MMPS | 100 | 0.03 | 0.3 | 1 |
| Sparse | MMPS | 100 | 0.03 | 0.3 | 1 |
| Multimodal | MMPS | 100 | 0.03 | 0.3 | 1 |
| Saturating | MMPS | 100 | 0.03 | 0.3 | 20 |
| High-dim. | MMPS | 100 | 0.03 | 0.0 | 1 |
| Non-stationary | MMPS | 100 | 0.03 | 0.5 | 3 |
| SEVIR | MMPS | 100 | 0.03 | 0.3 | 1 |

### C.4.5. RESULTS

We report RMSE and SSR for the SQG and SEVIR experiments. RMSE values in Table 11 align closely with the CRPS results in Table 3. The SSR in Table 13 indicates that DAISI is slightly overdispersive, but as shown in Figure 8, SSR is sensitive to the choice of $\epsilon$ and tuning this parameter more could further improve calibration. Examples of assimilated states for all settings and methods are in Figures 11–16

### C.4.6. ABLATION

We assess the contribution of each component in DAISI through a series of ablations summarized in Table 14.

**SDEdit backward step:** One can replace the inversion step with a single SDEdit (Meng et al., 2022) step, in which the forecast $z_1$ is partially noised to the latent $z_{t_{\min}} = \alpha_{t_{\min}} z_0 + \beta_{t_{\min}} z_1$, $z_0 \sim \mathcal{N}(\mathbf{0}, \mathbf{I})$. Using a tuned value of $t_{\min} = 0.4$, this approach performs worse than using the backward SDE. The reason is that $t_{\min}$ simultaneously controls the strength of the corrective guidance and how much of the forecast information is preserved. This coupling introduces a trade-off, starting later preserves more forecast information but leaves less room for correction.

**No inversion:** Removing the inversion entirely results in a large degradation, underscoring the importance of injecting dynamical information from the forecast into the latent space. As illustrated in Figure 17, the inversion step is also critical

*Table 11.* The RMSE for experiments on SQG and SEVIR. We display the mean and standard deviation across 10 independent trajectories, averaged over the last 20 (10 for SEVIR) steps. The best score for each experiment is highlighted in **bold** and the second best with an underline. Since SDA (smoothing) solves a different problem, we exclude it from the relative ranking.

| Experiment | DAISI | LETKF | FlowDAS | EnSF | SDA (filtering) | SDA (smoothing) |
|---|---|---|---|---|---|---|
| Noisy | $\underline{2.46}_{\pm 0.23}$ | $2.51_{\pm 0.26}$ | $5.01_{\pm 0.69}$ | $8.07_{\pm 1.01}$ | $\mathbf{2.11}_{\pm 0.18}$ | $2.15_{\pm 0.20}$ |
| Sparse | $\mathbf{3.40}_{\pm 0.36}$ | $4.55_{\pm 0.53}$ | $6.29_{\pm 0.90}$ | $7.09_{\pm 0.74}$ | $\underline{4.17}_{\pm 0.44}$ | $2.88_{\pm 0.20}$ |
| Multimodal | $\underline{3.71}_{\pm 0.95}$ | $\mathbf{3.62}_{\pm 1.12}$ | $6.93_{\pm 0.64}$ | $9.64_{\pm 0.89}$ | $6.78_{\pm 0.70}$ | $7.88_{\pm 0.52}$ |
| Saturating | $\underline{2.96}_{\pm 0.17}$ | $9.65_{\pm 0.69}$ | $4.71_{\pm 0.69}$ | $\mathbf{2.67}_{\pm 0.35}$ | $7.52_{\pm 0.73}$ | $6.73_{\pm 0.34}$ |
| SEVIR | $\mathbf{0.049}_{\pm 0.02}$ | $0.053_{\pm 0.02}$ | $0.082_{\pm 0.02}$ | $0.132_{\pm 0.03}$ | $\underline{0.051}_{\pm 0.01}$ | $0.041_{\pm 0.01}$ |

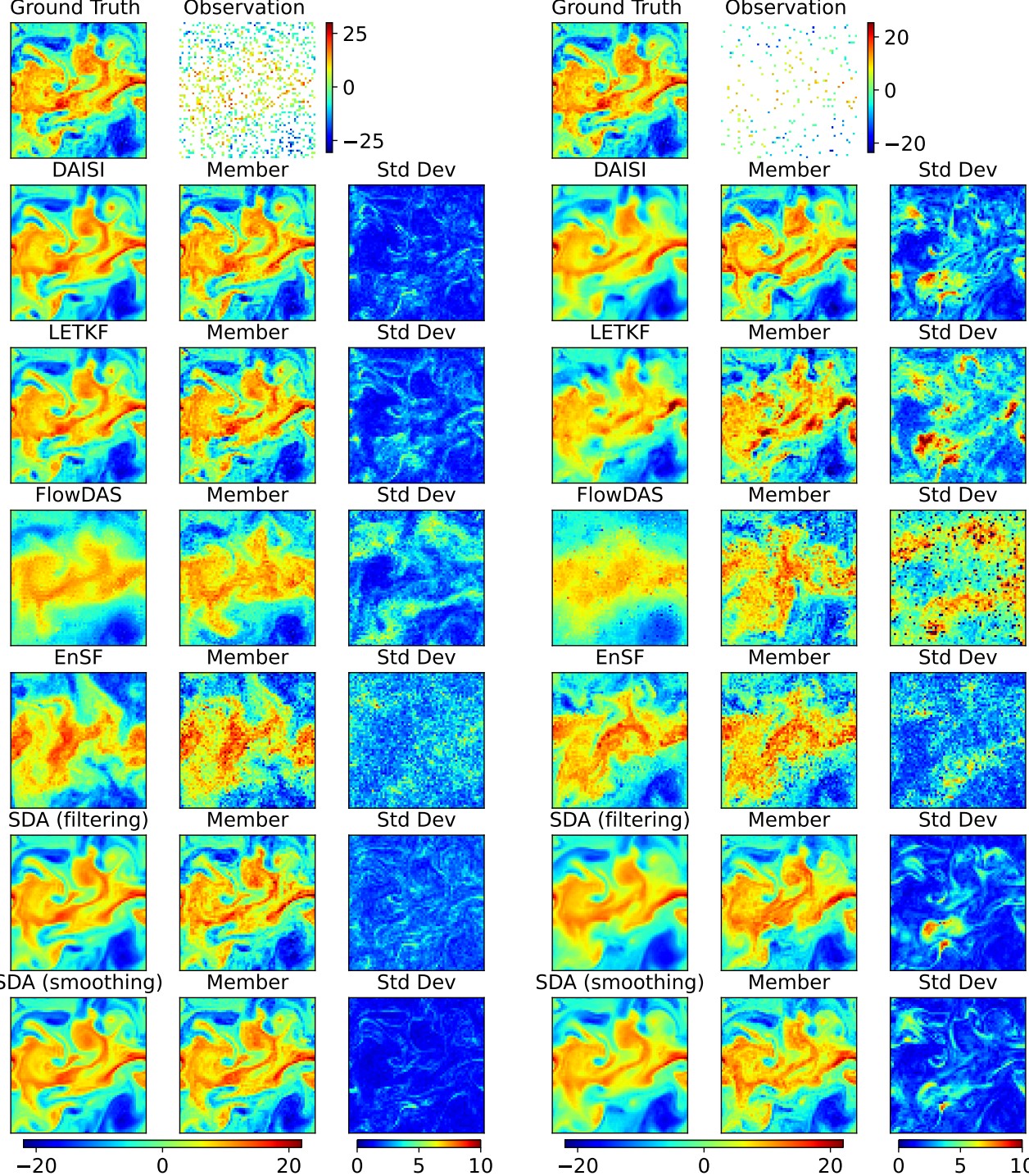

*Figure 11.* A comparison of the ensemble mean, individual members, and ensemble standard deviation for each method at the last step of the assimilated trajectory for the `Noisy` experiment.

*Figure 12.* A comparison of the ensemble mean, individual members, and ensemble standard deviation for each method at the last step of the assimilated trajectory for the `Sparse` experiment.

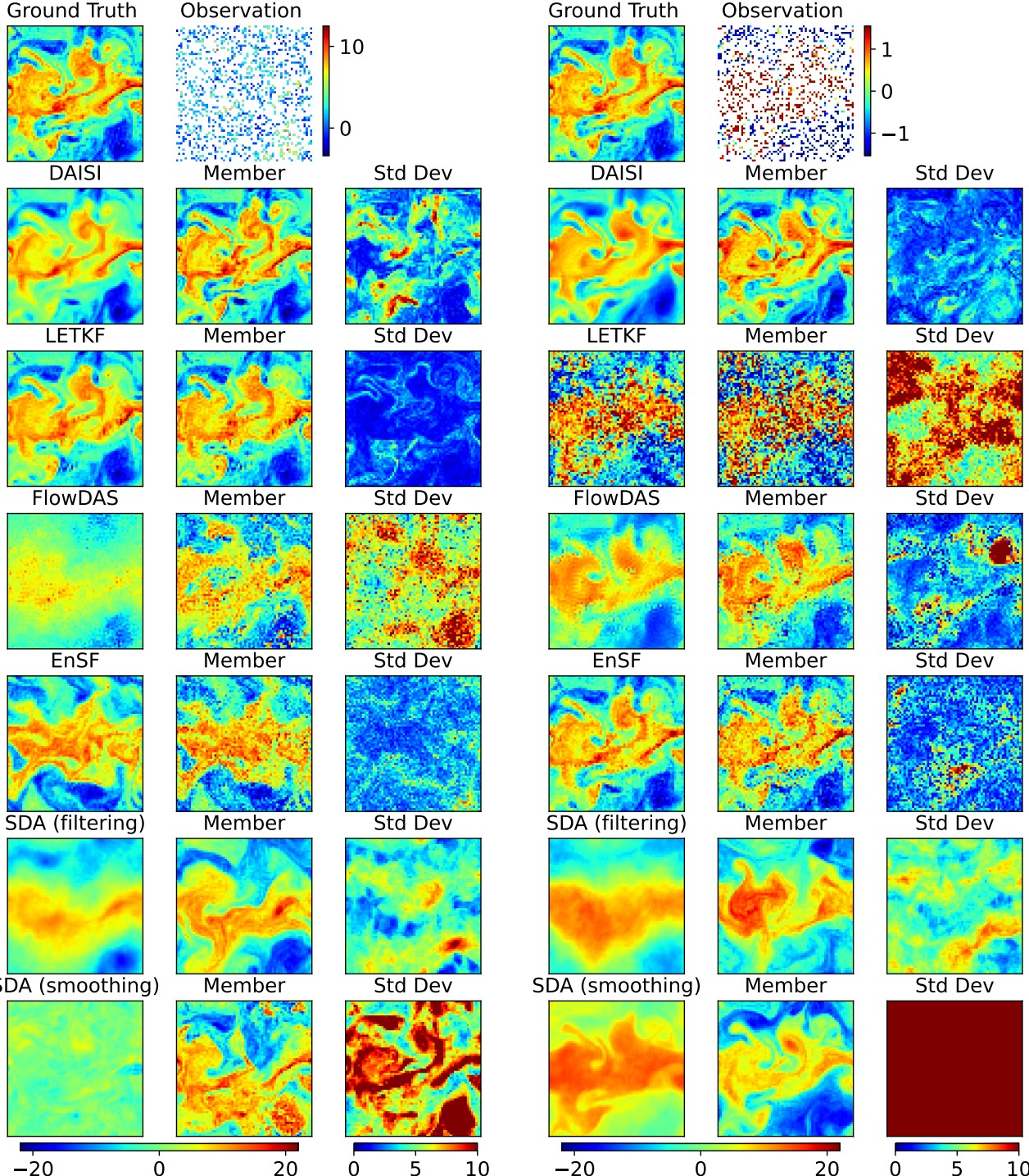

*Figure 13.* A comparison of the ensemble mean, individual members, and ensemble standard deviation for each method at the last step of the assimilated trajectory of the `Multimodal` experiment.

*Figure 14.* A comparison of the ensemble mean, individual members, and ensemble standard deviation for each method at the last step of the assimilated trajectory of the `Saturating` experiment.

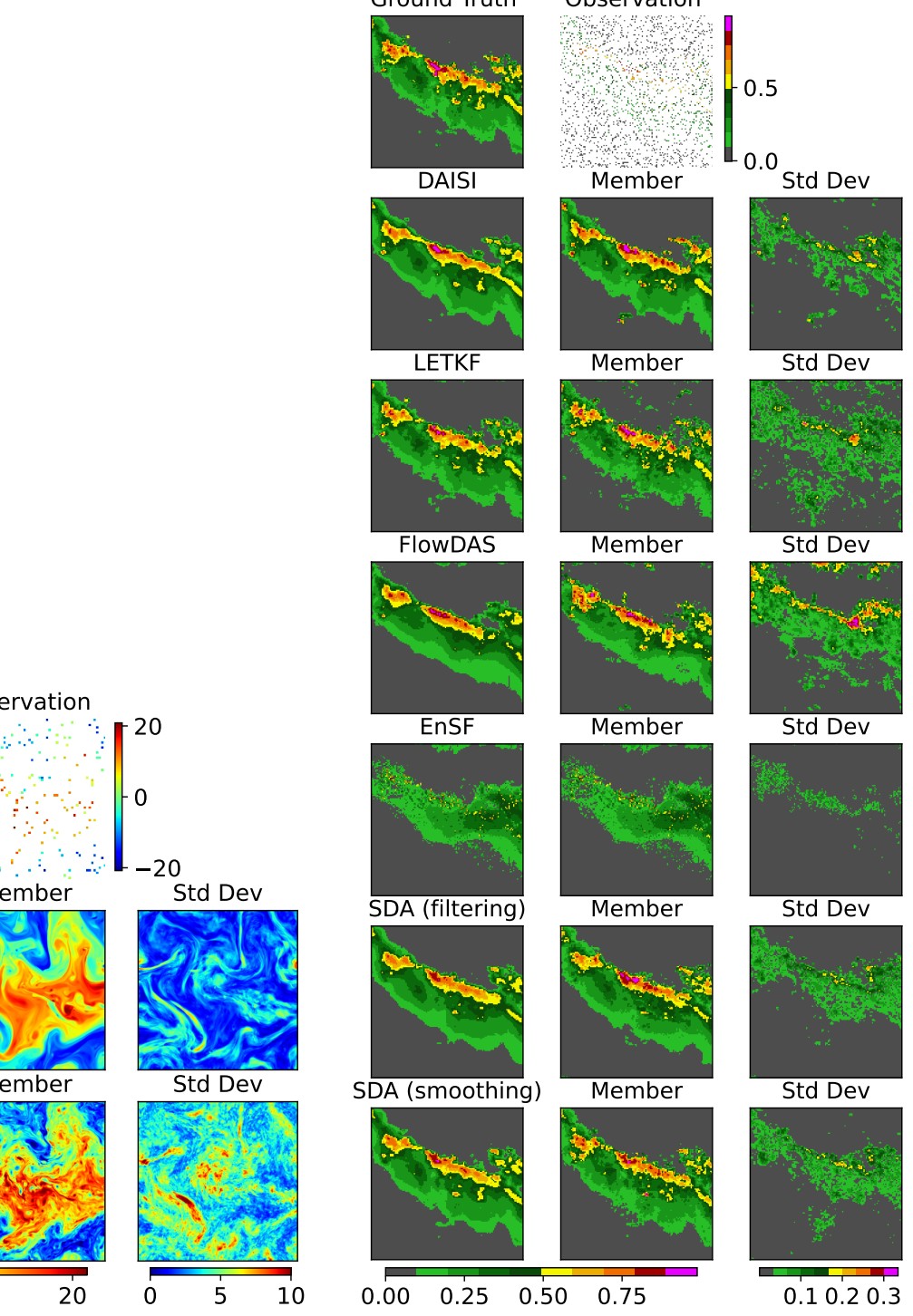

*Figure 15.* A comparison of the ensemble mean, individual members, and ensemble standard deviation for each method at the last step of the assimilated trajectory of the `High-dim.` experiment. The left colorbar is valid for the truth, samples and observation.

*Figure 16.* A comparison of the ensemble mean, individual members, and ensemble standard deviation for each method at the last step of the assimilated trajectory of the `SEVIR` experiment. The left colorbar is valid for the truth, samples and observation.

*Table 12.* The CRPS for experiments on SQG and SEVIR. We display the mean and standard deviation across 10 independent trajectories, averaged over the last 20 (10 for SEVIR) steps. The best score for each experiment is highlighted in **bold** and the second best with an underline. Since SDA (smoothing) solves a different problem, we exclude it from the relative ranking.

| Experim. | DAISI | LETKF | FlowDAS | EnSF | SDA (filtering) | SDA (smoothing) |
|---|---|---|---|---|---|---|
| Noisy | $\underline{1.32}_{\pm 0.13}$ | $1.34_{\pm 0.14}$ | $2.81_{\pm 0.43}$ | $5.15_{\pm 0.78}$ | $\mathbf{1.13}_{\pm 0.10}$ | $1.21_{\pm 0.11}$ |
| Sparse | $\mathbf{1.73}_{\pm 0.18}$ | $2.35_{\pm 0.28}$ | $3.34_{\pm 0.53}$ | $4.20_{\pm 0.56}$ | $\underline{2.22}_{\pm 0.25}$ | $1.53_{\pm 0.11}$ |
| Multimodal | $\mathbf{1.81}_{\pm 0.44}$ | $\underline{1.97}_{\pm 0.69}$ | $3.66_{\pm 0.42}$ | $6.38_{\pm 0.75}$ | $3.88_{\pm 0.48}$ | $3.82_{\pm 0.32}$ |
| Saturating | $\underline{1.54}_{\pm 0.09}$ | $5.24_{\pm 0.35}$ | $2.41_{\pm 0.39}$ | $\mathbf{1.33}_{\pm 0.16}$ | $4.20_{\pm 0.44}$ | $3.37_{\pm 0.14}$ |
| SEVIR | $\mathbf{0.016}_{\pm 0.01}$ | $\underline{0.018}_{\pm 0.01}$ | $0.045_{\pm 0.01}$ | $0.075_{\pm 0.02}$ | $\underline{0.018}_{\pm 0.01}$ | $0.013_{\pm 0.00}$ |

*Table 13.* The SSR for experiments on SQG and SEVIR. We display the mean and standard deviation across 10 independent trajectories, averaged over the last 20 (10 for SEVIR) steps. The best score for each experiment is highlighted in **bold** and the second best with an underline. Since SDA (smoothing) solves a different problem, we exclude it from the relative ranking.

| Experiment | DAISI | LETKF | FlowDAS | EnSF | SDA (filtering) | SDA (smoothing) |
|---|---|---|---|---|---|---|
| Noisy | $\mathbf{1.06}_{\pm 0.03}$ | $\underline{1.21}_{\pm 0.05}$ | $0.78_{\pm 0.07}$ | $0.49_{\pm 0.05}$ | $1.44_{\pm 0.04}$ | $0.78_{\pm 0.02}$ |
| Sparse | $1.24_{\pm 0.04}$ | $1.23_{\pm 0.10}$ | $\underline{1.13}_{\pm 0.06}$ | $0.57_{\pm 0.05}$ | $\mathbf{0.88}_{\pm 0.04}$ | $0.83_{\pm 0.03}$ |
| Multimodal | $1.66_{\pm 0.12}$ | $0.76_{\pm 0.15}$ | $\underline{1.21}_{\pm 0.07}$ | $0.42_{\pm 0.04}$ | $\mathbf{0.91}_{\pm 0.07}$ | $1.19_{\pm 0.04}$ |
| Saturating | $1.17_{\pm 0.03}$ | $0.96_{\pm 0.07}$ | $\mathbf{1.02}_{\pm 0.11}$ | $1.60_{\pm 0.18}$ | $\underline{1.03}_{\pm 0.08}$ | $4.81_{\pm 0.25}$ |
| SEVIR | $1.19_{\pm 0.12}$ | $1.35_{\pm 0.14}$ | $1.09_{\pm 0.17}$ | $0.19_{\pm 0.08}$ | $\mathbf{1.03}_{\pm 0.10}$ | $1.23_{\pm 0.09}$ |

for producing temporally smooth, physically consistent trajectories.

**Hyperparameters:** We further study the sensitivity to the hyperparameters $\epsilon$ and $t_{\min}$ in the Sparse experiment. DAISI is generally robust to the choice of $\epsilon$, however, setting $\epsilon > 0$ improves the probabilistic metrics (Figure 18). In contrast, $t_{\min}$ has a substantial impact on performance (Figure 19), suggesting its importance for minimizing the bias inherent in DAISI.

**Non-stationary observations:** Finally, we verify that DAISI can handle non-stationary observations, such as those arising from remote sensing instruments. The observation operator is linear with $\sigma_{\text{obs}} = 1$ and consists of a band of width 4 px that shifts 4 px to the right at each timestep. We find that DAISI performs comparably to LETKF (Figure 21) while maintaining meaningful uncertainty (Figure 20).

**Assimilation interval:** We also examine the effect of reduced assimilation frequency. Increasing the interval between observations amplifies forecast nonlinearities, which poses difficulties for LETKF. In the Noisy experiment, both LETKF and DAISI perform similarly with 3-hour assimilation intervals, but when the interval is extended to 12 hours, DAISI clearly outperforms LETKF (Figure 22).

## C.5. SEVIR

Here, we consider a real-life large-scale weather forecasting task using the Storm EVent Imagery and Radar (SEVIR) dataset (Veillette et al., 2020), which provides observations of severe convective storms across the continental United States. This specific experiment focuses on the Vertically Integrated Liquid (VIL) product, which serves as a 2-D proxy for precipitation intensity. Each data sample is a $128 \times 128$ grid covering a $384 \times 384$ km area at 2 km resolution, with snapshots recorded every 10 min, for a total of 250 min (i.e., 25 snapshots per sample).

For the forecast model, we used the checkpoint available in the official FlowDAS GitHub repository https://github.

*Table 14.* The CRPS for ablations on SQG. We display the mean and standard deviation across 10 independent trajectories, averaged over the last 20 (10 for SEVIR) steps.

| Experiment | DAISI | with SDEdit backward step | without Inversion |
|---|---|---|---|
| Noisy | $1.32_{\pm 0.13}$ | $2.12_{\pm 0.27}$ | $2.63_{\pm 0.26}$ |
| Sparse | $1.73_{\pm 0.18}$ | $2.17_{\pm 0.25}$ | $2.40_{\pm 0.22}$ |
| Multimodal | $1.81_{\pm 0.44}$ | $2.72_{\pm 0.45}$ | $4.35_{\pm 0.33}$ |
| Saturating | $1.54_{\pm 0.09}$ | $1.85_{\pm 0.20}$ | $2.17_{\pm 0.19}$ |

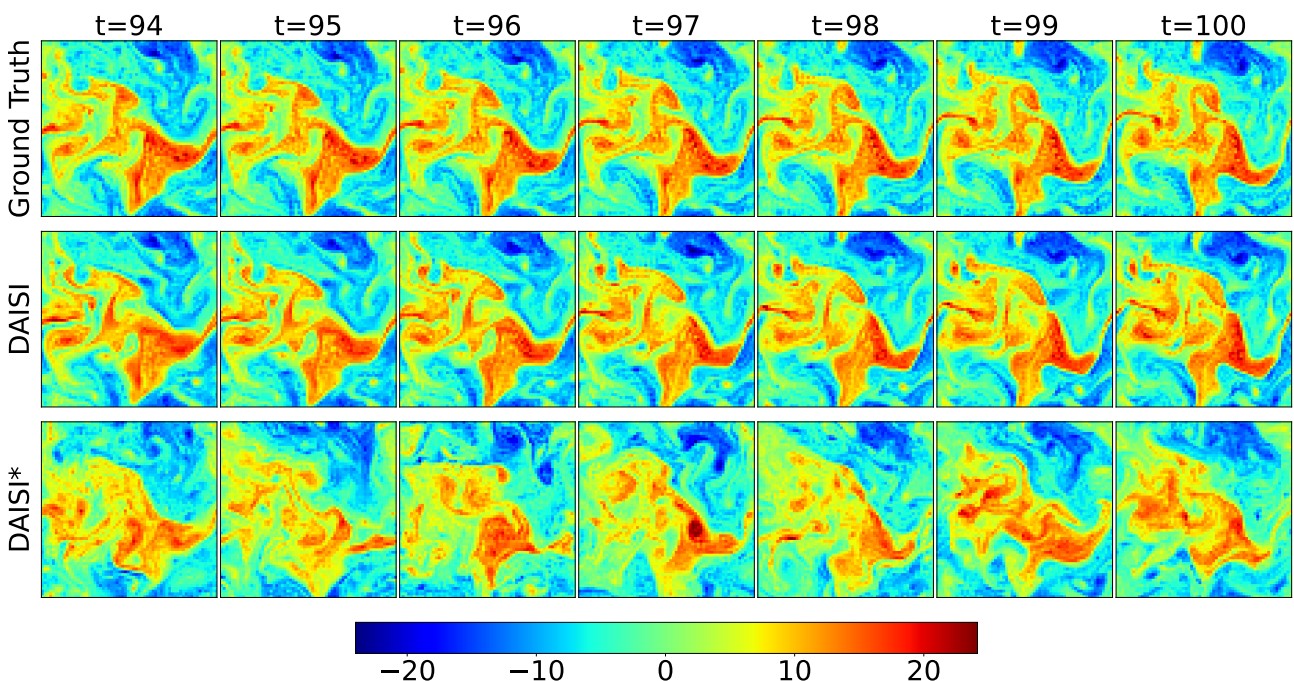

*Figure 17.* Comparison of an ensemble member trajectory on the `Noisy` experiment between DAISI and DAISI* (without inversion). The DAISI* variant exhibits discontinuities between time steps $t - 1$ and $t$, as forecast information is not propagated forward, leading to worse reconstructions and incoherent temporal evolution.

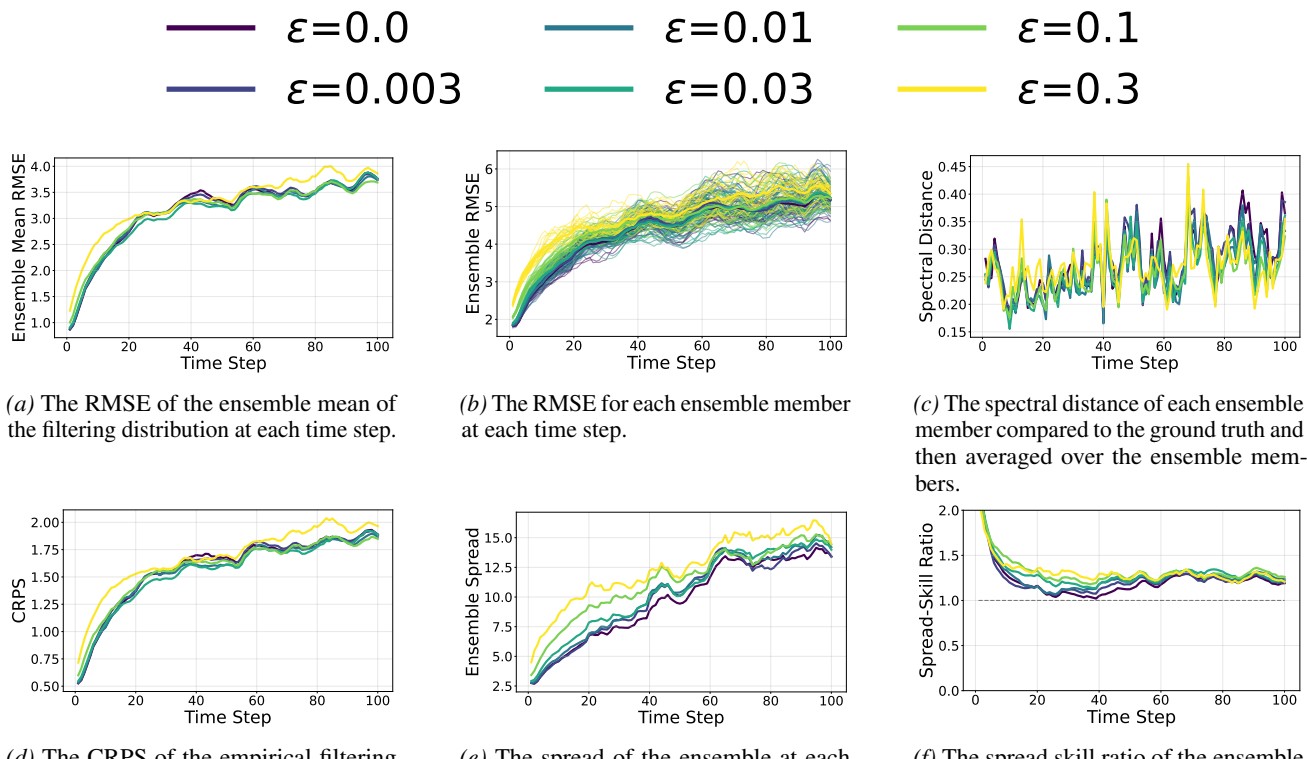

*(a)* The RMSE of the ensemble mean of the filtering distribution at each time step.

*(b)* The RMSE for each ensemble member at each time step.

*(c)* The spectral distance of each ensemble member compared to the ground truth and then averaged over the ensemble members.

*(d)* The CRPS of the empirical filtering distribution at each time step.

*(e)* The spread of the ensemble at each time step.

*(f)* The spread skill ratio of the ensemble at each time step.

*Figure 18.* The results for the $\epsilon$ ablation.

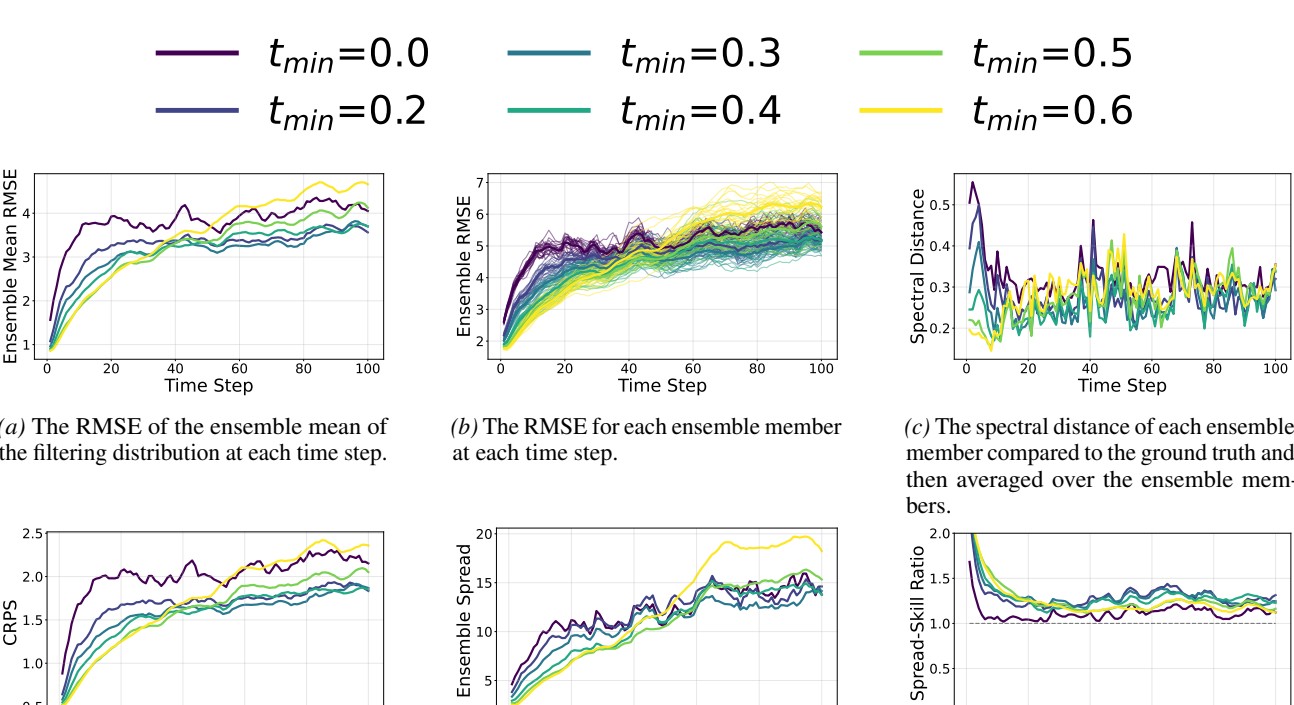

*(a)* The RMSE of the ensemble mean of the filtering distribution at each time step.

*(b)* The RMSE for each ensemble member at each time step.

*(c)* The spectral distance of each ensemble member compared to the ground truth and then averaged over the ensemble members.

*(d)* The CRPS of the empirical filtering distribution at each time step.

*(e)* The spread of the ensemble at each time step.

*(f)* The spread skill ratio of the ensemble at each time step.

*Figure 19.* The results for the $t_{\min}$ ablation.

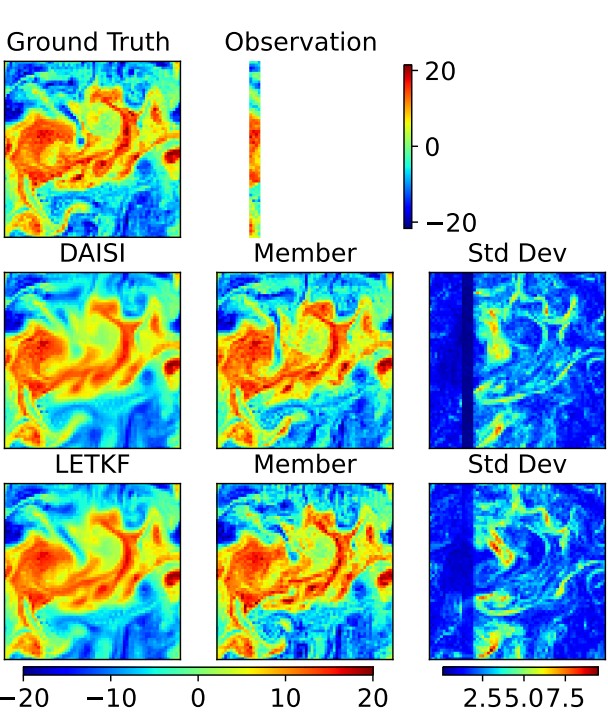

*Figure 20.* A comparison of the ensemble mean, individual members, and ensemble standard deviation for each method at the last step of the assimilated trajectory for the `Non-stationary` experiment.

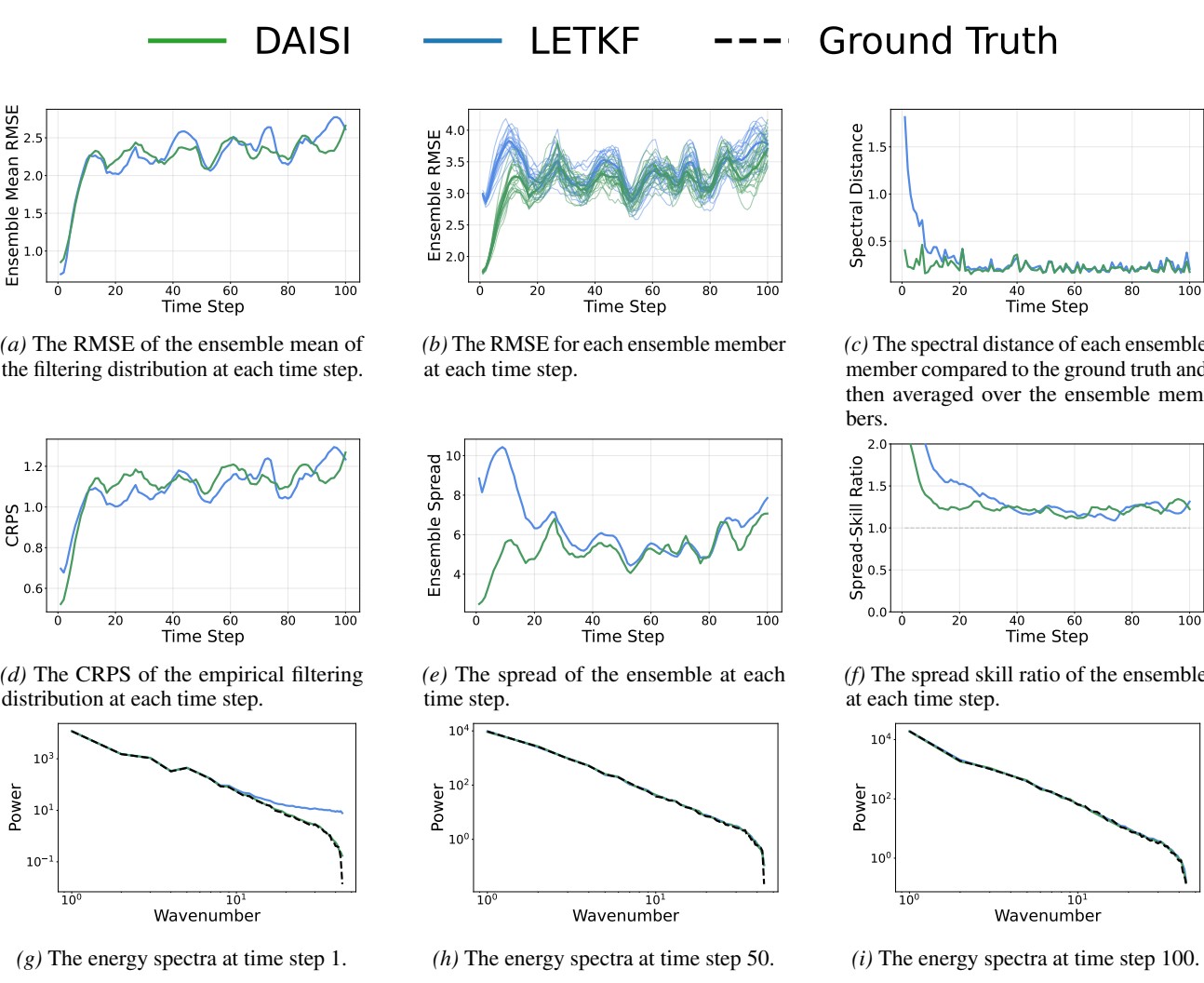

*(a)* The RMSE of the ensemble mean of the filtering distribution at each time step.

*(b)* The RMSE for each ensemble member at each time step.

*(c)* The spectral distance of each ensemble member compared to the ground truth and then averaged over the ensemble members.

*(d)* The CRPS of the empirical filtering distribution at each time step.

*(e)* The spread of the ensemble at each time step.

*(f)* The spread skill ratio of the ensemble at each time step.

*(g)* The energy spectra at time step 1.

*(h)* The energy spectra at time step 50.

*(i)* The energy spectra at time step 100.

*Figure 21.* The results for the `Non-stationary` experiment.

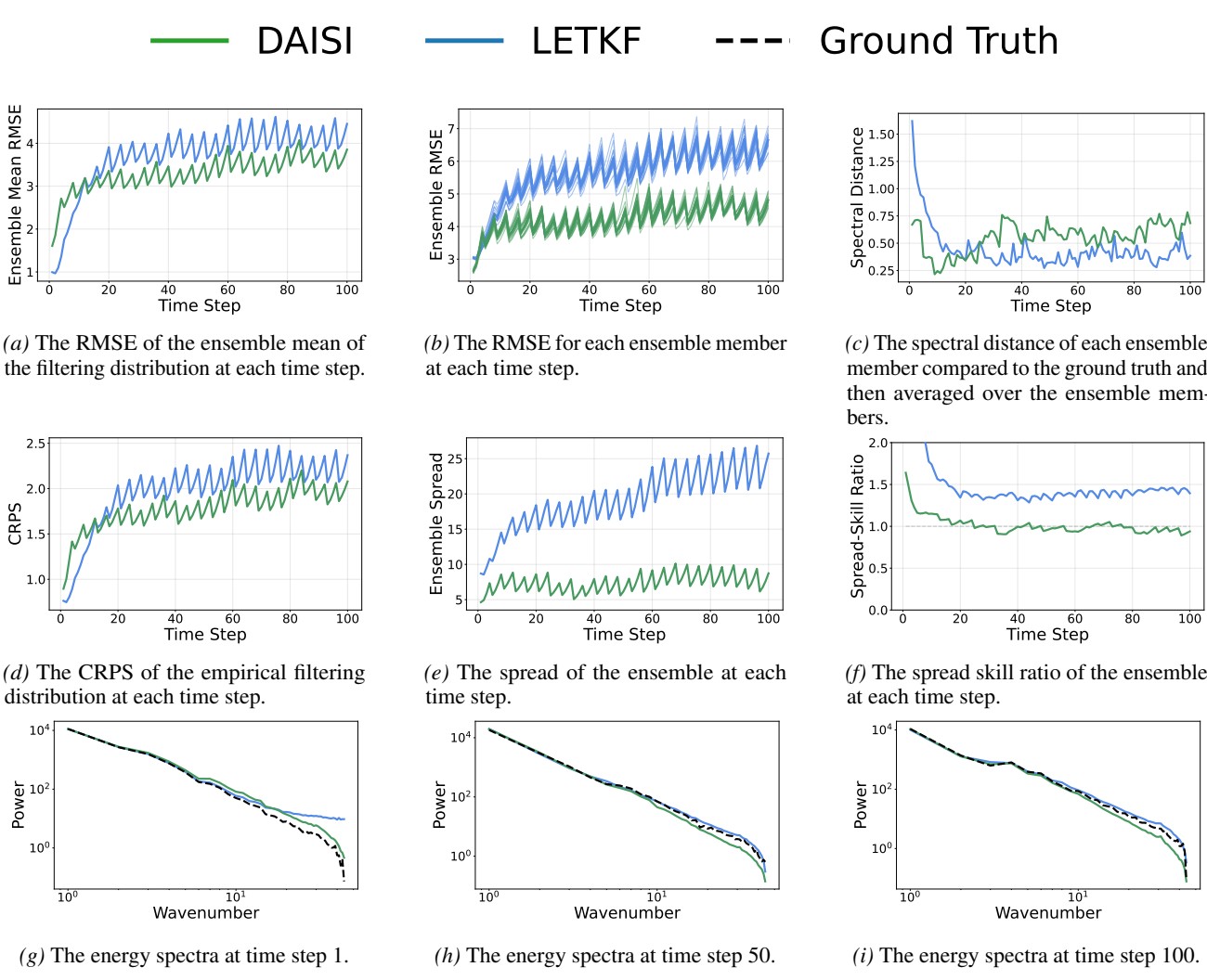

*(a)* The RMSE of the ensemble mean of the filtering distribution at each time step.

*(b)* The RMSE for each ensemble member at each time step.

*(c)* The spectral distance of each ensemble member compared to the ground truth and then averaged over the ensemble members.

*(d)* The CRPS of the empirical filtering distribution at each time step.

*(e)* The spread of the ensemble at each time step.

*(f)* The spread skill ratio of the ensemble at each time step.

*(g)* The energy spectra at time step 1.

*(h)* The energy spectra at time step 50.

*(i)* The energy spectra at time step 100.

*Figure 22.* The results for the `Noisy 12h` experiment.

 This is based on a stochastic interpolant-based generative SDE for probabilistic forecasting, proposed in (Chen et al., 2024), with a U-Net backbone used for the drift. The model makes predictions autoregressively with inputs given by the past 6 timesteps. For the guided FlowDAS baseline, we take the exact setup as in (Chen et al., 2025), with parameters listed in Table 8. When beginning the assimilation, we use the first six ground truth states as initial conditions. For DAISI we use the same U-Net as for the SQG experiments, without the circular padding. We do not scale the data as it is already in the range $[0, 1]$.

## D. Additional figures

In Figures 23–28 we show additional metrics for one of the assimilated trajectories.

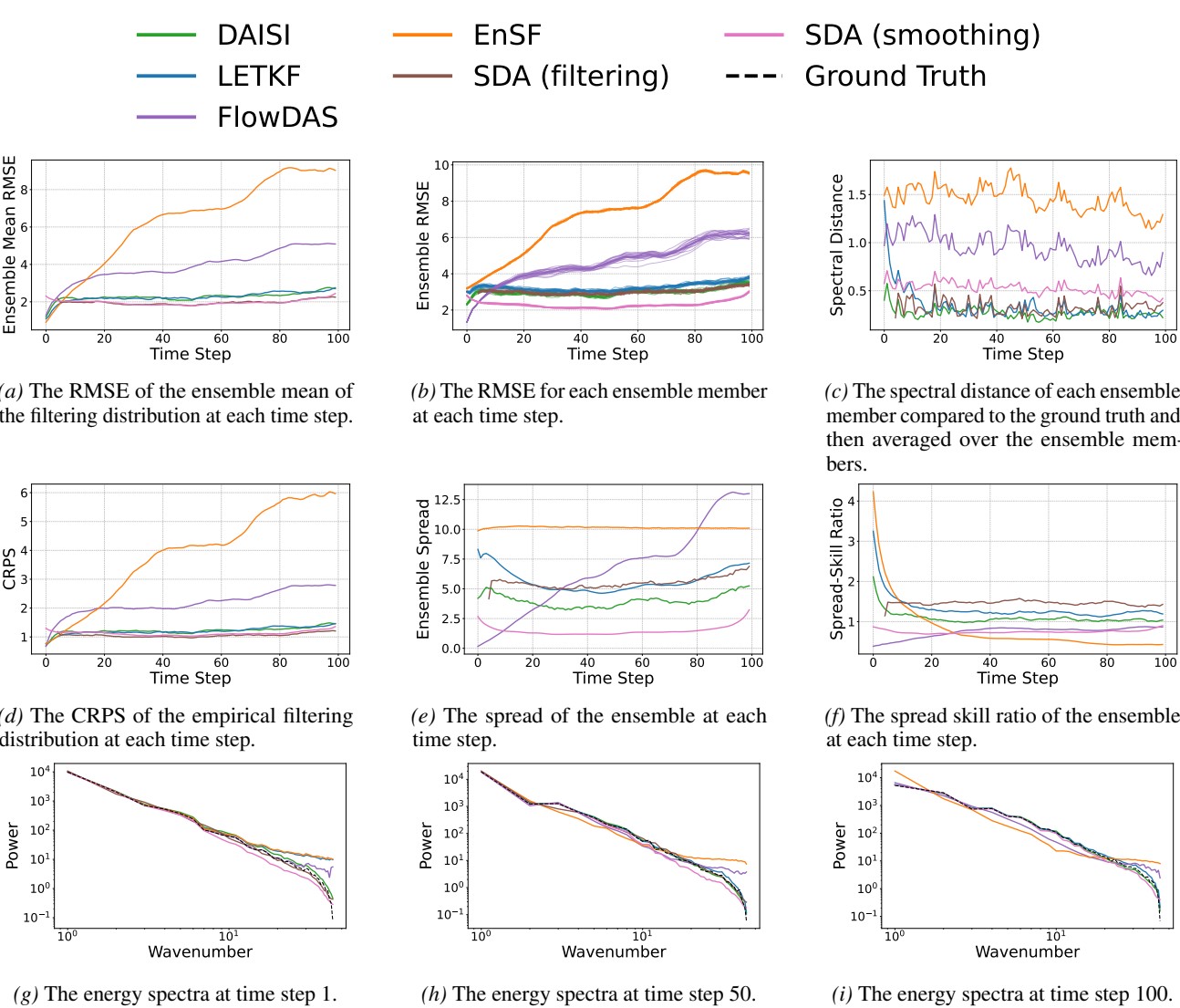

*(a)* The RMSE of the ensemble mean of the filtering distribution at each time step.

*(b)* The RMSE for each ensemble member at each time step.

*(c)* The spectral distance of each ensemble member compared to the ground truth and then averaged over the ensemble members.

*(d)* The CRPS of the empirical filtering distribution at each time step.

*(e)* The spread of the ensemble at each time step.

*(f)* The spread skill ratio of the ensemble at each time step.

*(g)* The energy spectra at time step 1.

*(h)* The energy spectra at time step 50.

*(i)* The energy spectra at time step 100.

*Figure 23.* The results for the `Noisy` experiment.

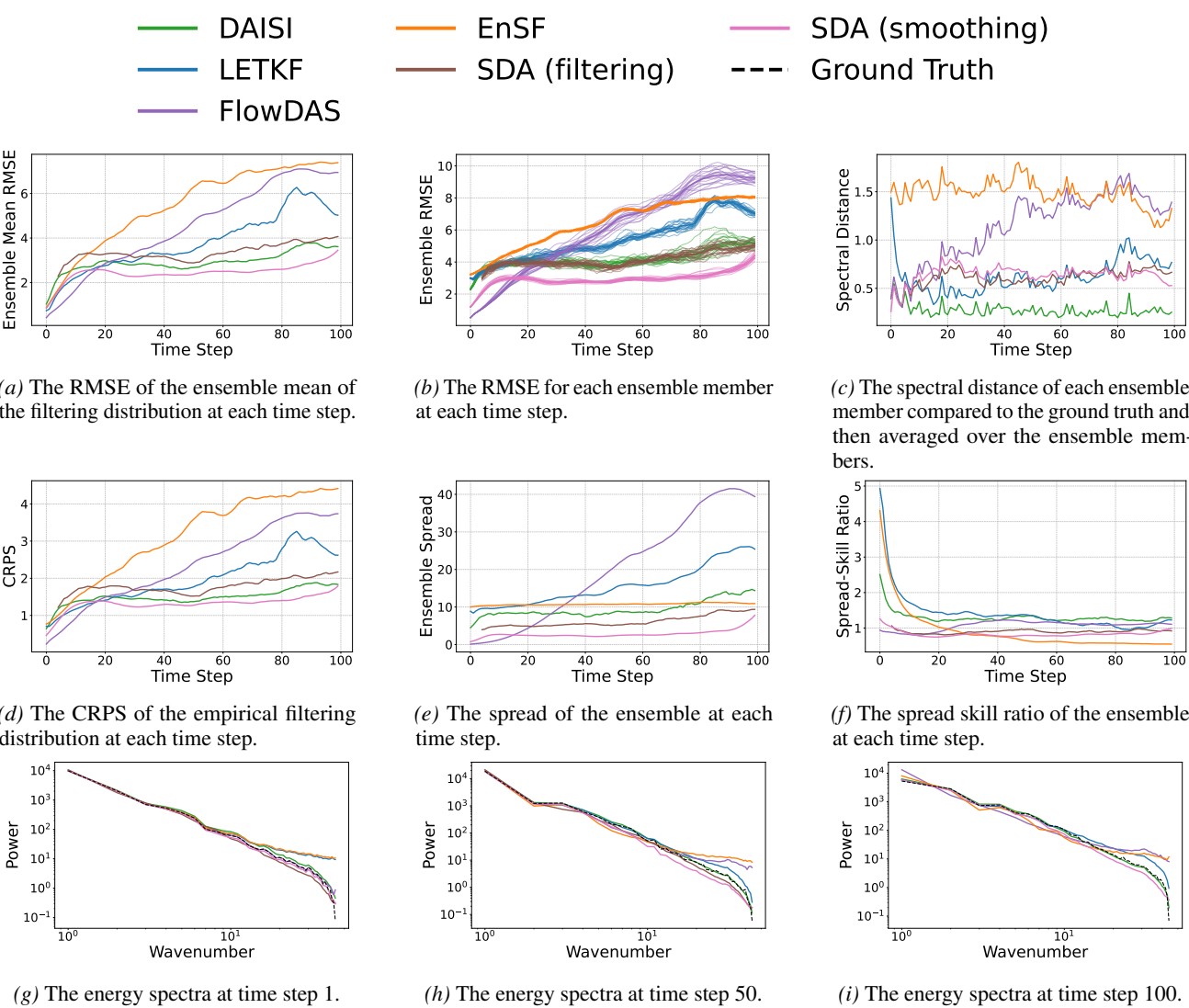

*(a)* The RMSE of the ensemble mean of the filtering distribution at each time step.

*(b)* The RMSE for each ensemble member at each time step.

*(c)* The spectral distance of each ensemble member compared to the ground truth and then averaged over the ensemble members.

*(d)* The CRPS of the empirical filtering distribution at each time step.

*(e)* The spread of the ensemble at each time step.

*(f)* The spread skill ratio of the ensemble at each time step.

*(g)* The energy spectra at time step 1.

*(h)* The energy spectra at time step 50.

*(i)* The energy spectra at time step 100.

*Figure 24.* The results for the `Sparse` experiment.

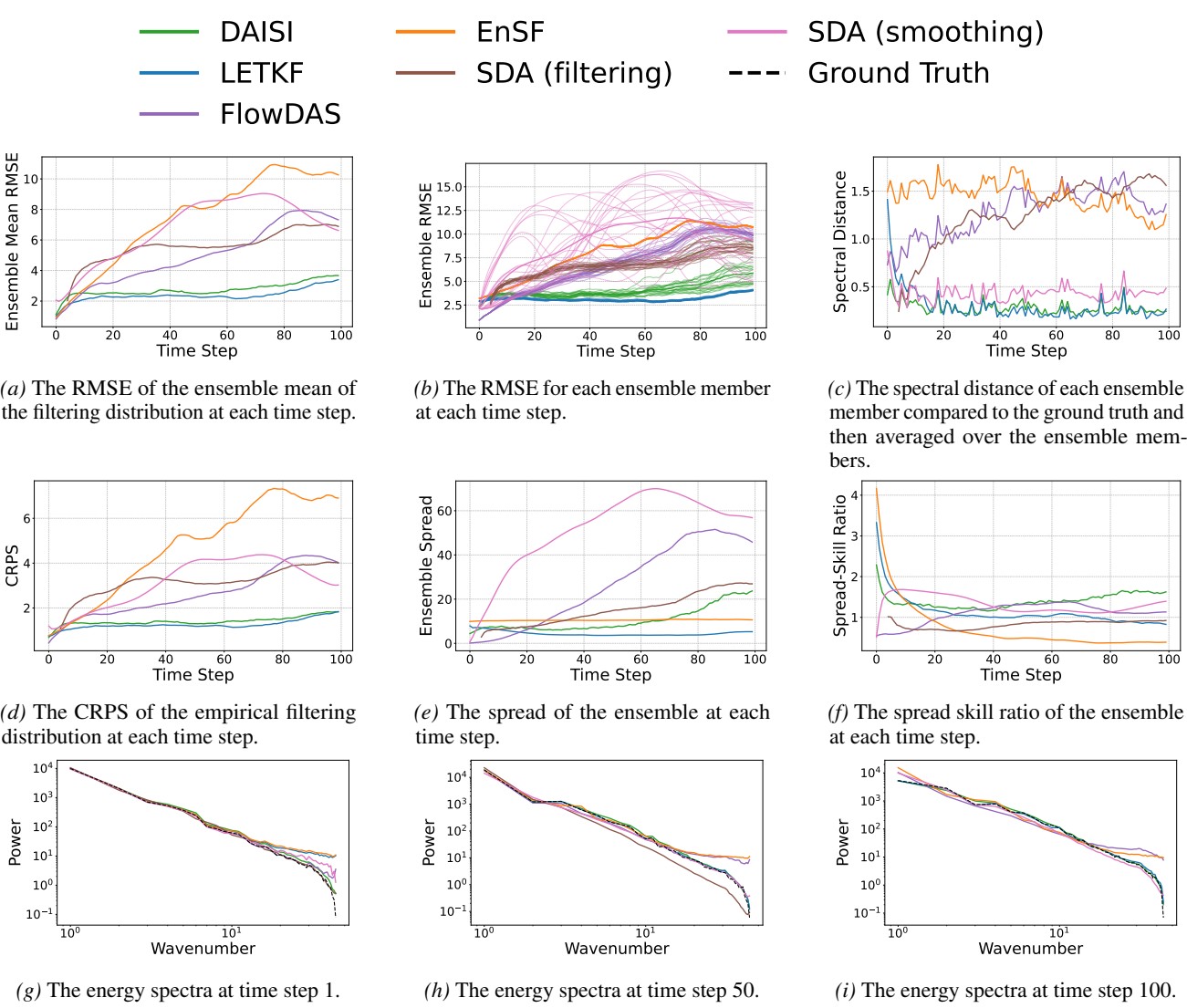

*(a)* The RMSE of the ensemble mean of the filtering distribution at each time step.

*(b)* The RMSE for each ensemble member at each time step.

*(c)* The spectral distance of each ensemble member compared to the ground truth and then averaged over the ensemble members.

*(d)* The CRPS of the empirical filtering distribution at each time step.

*(e)* The spread of the ensemble at each time step.

*(f)* The spread skill ratio of the ensemble at each time step.

*(g)* The energy spectra at time step 1.

*(h)* The energy spectra at time step 50.

*(i)* The energy spectra at time step 100.

*Figure 25.* The results for the `Multimodal` experiment.

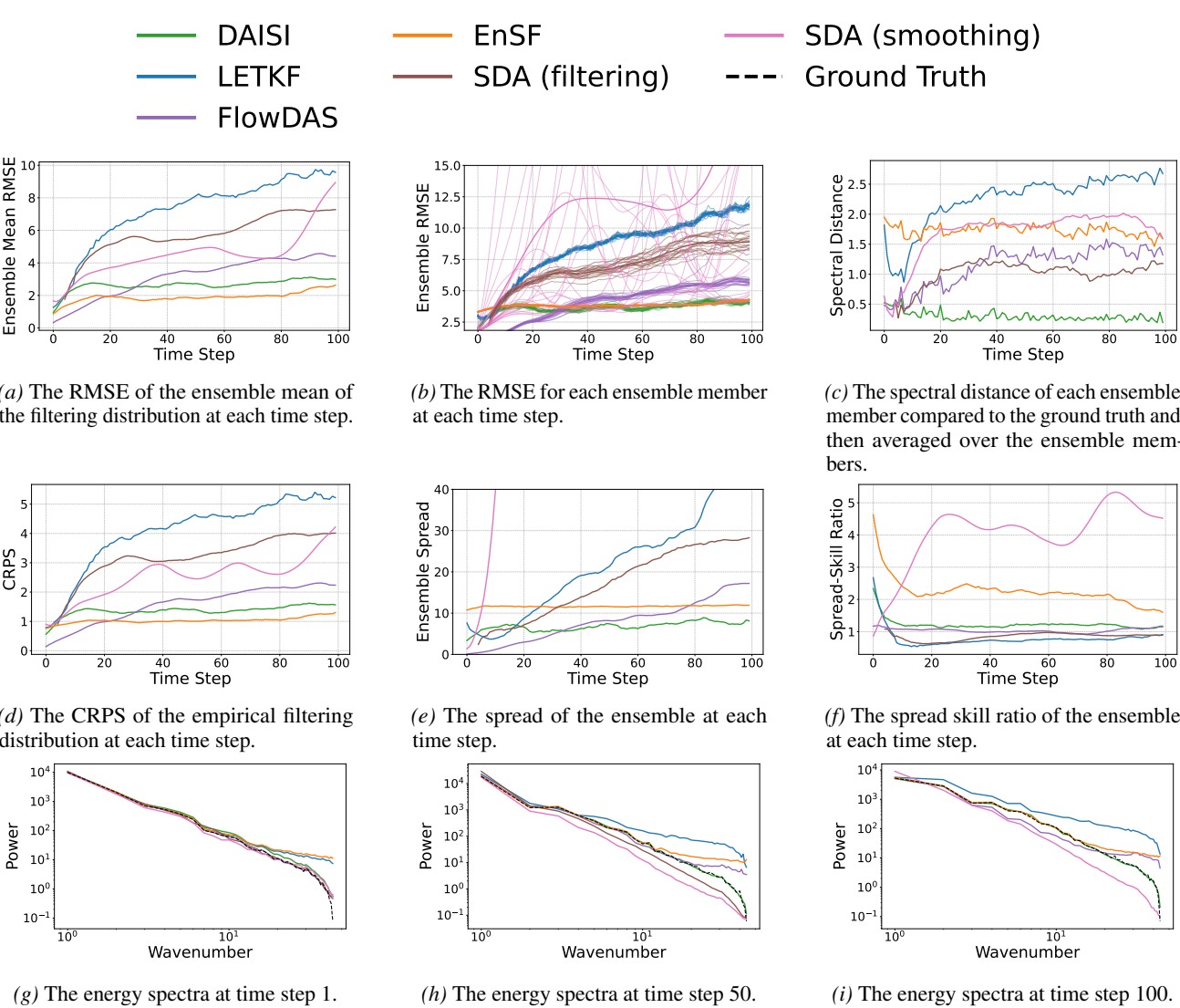

*(a)* The RMSE of the ensemble mean of the filtering distribution at each time step.

*(b)* The RMSE for each ensemble member at each time step.

*(c)* The spectral distance of each ensemble member compared to the ground truth and then averaged over the ensemble members.

*(d)* The CRPS of the empirical filtering distribution at each time step.

*(e)* The spread of the ensemble at each time step.

*(f)* The spread skill ratio of the ensemble at each time step.

*(g)* The energy spectra at time step 1.

*(h)* The energy spectra at time step 50.

*(i)* The energy spectra at time step 100.

*Figure 26.* The results for the `Saturating` experiment.

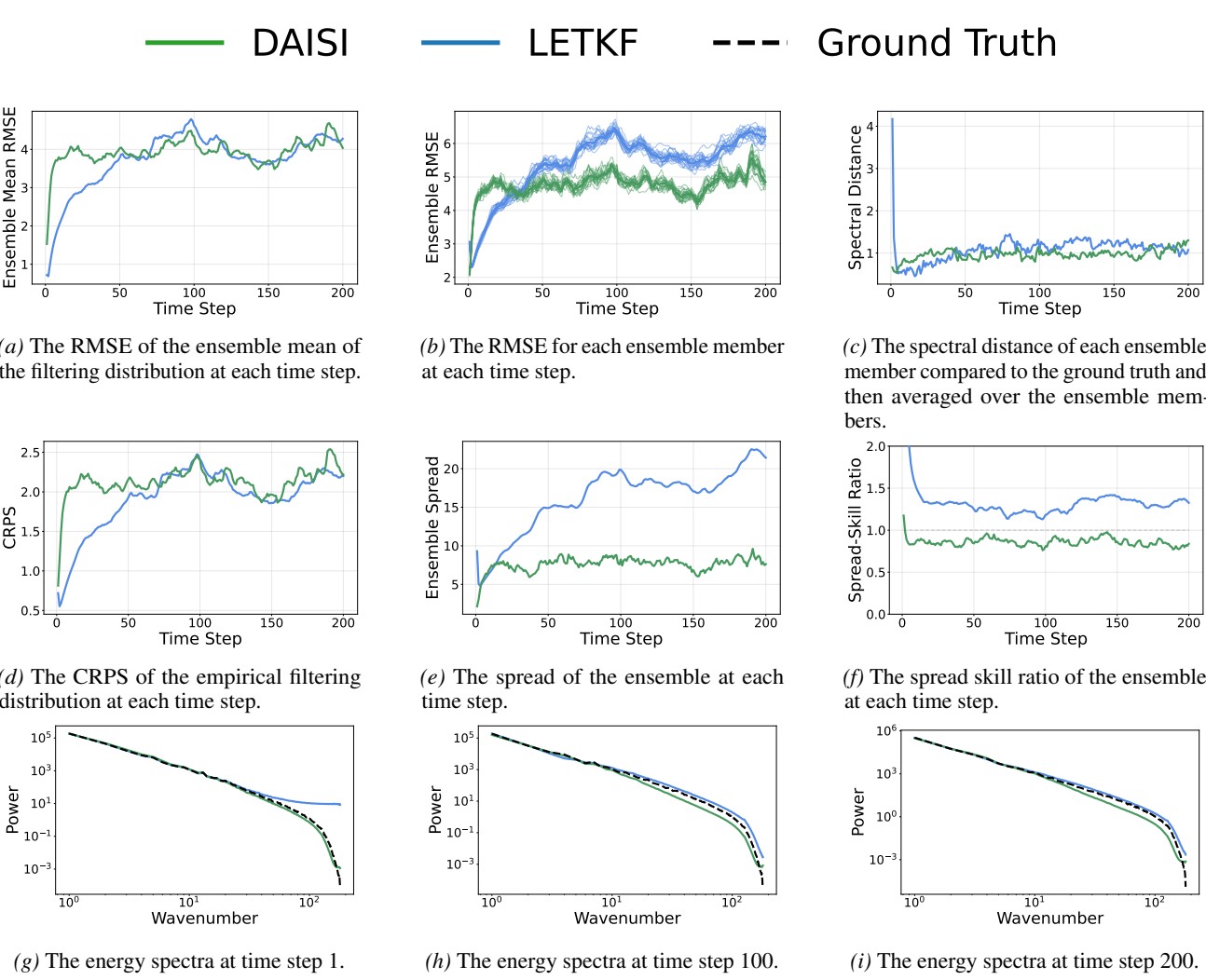

*(a)* The RMSE of the ensemble mean of the filtering distribution at each time step.

*(b)* The RMSE for each ensemble member at each time step.

*(c)* The spectral distance of each ensemble member compared to the ground truth and then averaged over the ensemble members.

*(d)* The CRPS of the empirical filtering distribution at each time step.

*(e)* The spread of the ensemble at each time step.

*(f)* The spread skill ratio of the ensemble at each time step.

*(g)* The energy spectra at time step 1.

*(h)* The energy spectra at time step 100.

*(i)* The energy spectra at time step 200.

*Figure 27.* The results for the `High-dim.` experiment.

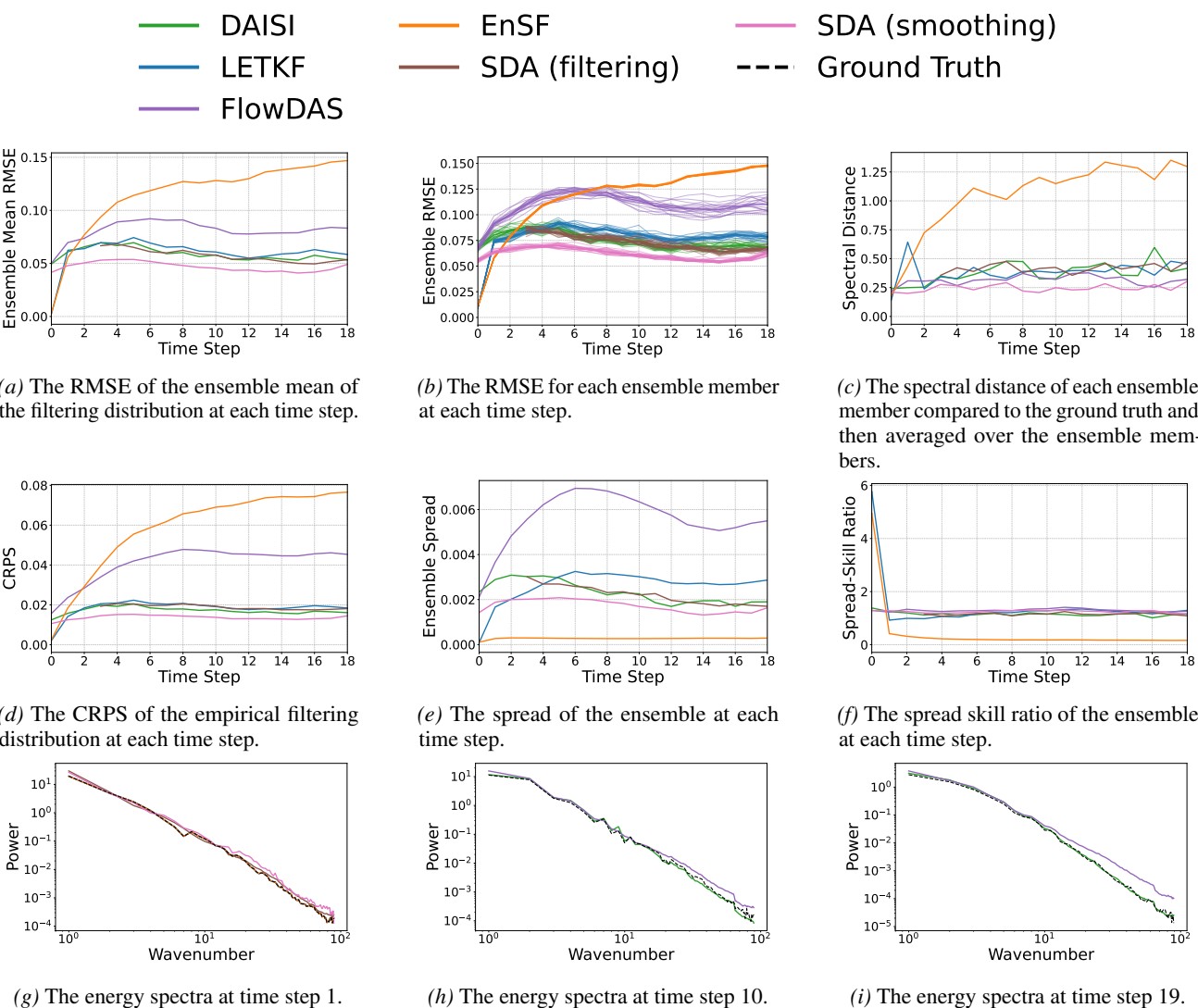

*(a)* The RMSE of the ensemble mean of the filtering distribution at each time step.

*(b)* The RMSE for each ensemble member at each time step.

*(c)* The spectral distance of each ensemble member compared to the ground truth and then averaged over the ensemble members.

*(d)* The CRPS of the empirical filtering distribution at each time step.

*(e)* The spread of the ensemble at each time step.

*(f)* The spread skill ratio of the ensemble at each time step.

*(g)* The energy spectra at time step 1.

*(h)* The energy spectra at time step 10.

*(i)* The energy spectra at time step 19.

*Figure 28.* The results for the `SEVIR` experiment.

## E. Entropy dissipation

In this section, we prove a Bakry-Émery-type entropy dissipation result (Proposition E.1) (Bakry & Émery, 2006), which describes the influence of the stochastic parameter $\epsilon$ on the information loss of the forecast ensembles when assimilating data with DAISI.

**Proposition E.1.** *Assume that the marginal laws $\{\rho_t\}_{t \in [0,1]}$ and $\{\rho_t^{\boldsymbol{y}}\}_{t \in [0,1]}$ of the interpolants bridging $\rho_0 = \mathcal{N}(\boldsymbol{0}, \mathbf{I})$ with $\rho_1 = \mathbb{P}_\infty$ and $\rho_1^{\boldsymbol{y}} = \mathbb{P}_\infty^{\boldsymbol{y}}$, respectively, satisfy a log-Sobolev inequality with constant $\lambda > 0$, uniform in t. That is,*

$$\mathcal{KL}(\mathbb{P} || \rho_t) \leq \frac{1}{2\lambda} \int_{\mathbb{R}^d} \left| \nabla \log \frac{\mathrm{d}\mathbb{P}}{\mathrm{d}\rho_t}(\boldsymbol{x}) \right|^2 \mathbb{P}(\mathrm{d}\boldsymbol{x}), \tag{116a}$$

$$\mathcal{KL}(\mathbb{P} || \rho_t^{\boldsymbol{y}}) \leq \frac{1}{2\lambda} \int_{\mathbb{R}^d} \left| \nabla \log \frac{\mathrm{d}\mathbb{P}}{\mathrm{d}\rho_t^{\boldsymbol{y}}}(\boldsymbol{x}) \right|^2 \mathbb{P}(\mathrm{d}\boldsymbol{x}), \tag{116b}$$

*for any $t \in [0, 1]$ and any probability measure $\mathbb{P}$. Then, we have*

$$\mathcal{KL}\left(\pi_{n,0,\epsilon}^{DAISI} || \mathbb{P}_\infty^{\boldsymbol{y}}\right) \leq e^{-4\lambda\epsilon} \mathcal{KL}\left(\hat{\pi}_n || \mathbb{P}_\infty\right), \tag{117}$$

*with equality at $\epsilon = 0$, where $\mathcal{KL}(q || p)$ denotes the Kullback-Leibler divergence between measures q and p.*

*Proof.* Consider a stochastic interpolant $\{\boldsymbol{z}_t\}$ bridging between $\rho_0 = \mathcal{N}(\boldsymbol{0}, \mathbf{I})$ and $\rho_1 = \mathbb{P}_\infty$ and let $p_t$ denote the Lebesgue density of $\rho_t = \mathrm{Law}(\boldsymbol{z}_t)$. Then, we know from Proposition A.1 that $\{p_t\}_{t \in [0,1]}$ is a solution to the transport equation

$$\frac{\partial p_t}{\partial t}(\boldsymbol{x}) + \mathrm{div}(\boldsymbol{b}_t(\boldsymbol{x}) p_t(\boldsymbol{x})) = 0, \quad p_0(\boldsymbol{x}) = \frac{\mathrm{d}\rho_0}{\mathrm{d}\boldsymbol{x}}, \tag{118}$$

where we denoted by $\mathrm{d}\boldsymbol{x}$ the Lebesgue measure on $\mathbb{R}^d$. Now consider the backward SDE (up to time reparameterization)

$$\mathrm{d}\boldsymbol{z}_\tau = -\boldsymbol{b}_{1-\tau}(\boldsymbol{z}_\tau)\mathrm{d}\tau + \epsilon \nabla \log p_{1-\tau}(\boldsymbol{z}_\tau) + \sqrt{2\epsilon}\,\mathrm{d}W_\tau, \quad z_0 \sim \hat{\pi}_n, \tag{119}$$

solved for $\tau = 0 \to 1$, which is used in the inverse sampling step of DAISI. For simplicity, we also denote $q_\tau := p_{1-\tau}$, which satisfies

$$\frac{\partial q_\tau}{\partial \tau}(\boldsymbol{x}) = \mathrm{div}(\boldsymbol{b}_{1-\tau}(\boldsymbol{x}) q_\tau(\boldsymbol{x})), \quad q_0(\boldsymbol{x}) = \frac{\mathrm{d}\mathbb{P}_\infty}{\mathrm{d}\boldsymbol{x}}(\boldsymbol{x}), \tag{120}$$

by (118). One can check that the Fokker-Planck equation of the SDE (119) reads

$$\frac{\partial r_\tau}{\partial \tau}(\boldsymbol{x}) = \mathrm{div}\left(\boldsymbol{b}_{1-\tau}(\boldsymbol{x}) r_\tau(\boldsymbol{x}) + \epsilon\, r_\tau \nabla \log(r_\tau(\boldsymbol{x})/q_\tau(\boldsymbol{x}))\right), \quad r_0(\boldsymbol{x}) = \frac{\mathrm{d}\hat{\pi}_n}{\mathrm{d}\boldsymbol{x}}(\boldsymbol{x}) \tag{121}$$

and consider the evolution of the Kullback-Leibler divergence between the distributions given by (120) and (121)

$$\mathcal{KL}(r_\tau || q_\tau) = \int_{\mathbb{R}^{d_x}} r_\tau(\boldsymbol{x}) \log \frac{r_\tau(\boldsymbol{x})}{q_\tau(\boldsymbol{x})} \mathrm{d}\boldsymbol{x}. \tag{122}$$

Taking the time derivative of (122) yields

$$\frac{\mathrm{d}}{\mathrm{d}\tau} \mathcal{KL}(r_\tau || q_\tau) = \int_{\mathbb{R}^{d_x}} \frac{\partial r_\tau}{\partial \tau}(\boldsymbol{x}) \log \frac{r_\tau(\boldsymbol{x})}{q_\tau(\boldsymbol{x})} \mathrm{d}\boldsymbol{x} + \int_{\mathbb{R}^{d_x}} r_\tau(\boldsymbol{x}) \frac{\partial}{\partial \tau} \left( \log \frac{r_\tau(\boldsymbol{x})}{q_\tau(\boldsymbol{x})} \right) \mathrm{d}\boldsymbol{x} \tag{123}$$

$$\stackrel{(121)}{=} \int_{\mathbb{R}^{d_x}} \mathrm{div}\left(\boldsymbol{b}_{1-\tau}(\boldsymbol{x}) r_\tau(\boldsymbol{x}) + \epsilon r_\tau(\boldsymbol{x}) \nabla \log(r_\tau(\boldsymbol{x})/q_\tau(\boldsymbol{x}))\right) \log \frac{r_\tau(\boldsymbol{x})}{q_\tau(\boldsymbol{x})} \mathrm{d}\boldsymbol{x}$$

$$+ \int_{\mathbb{R}^{d_x}} r_\tau(\boldsymbol{x}) \left( \frac{1}{r_\tau(\boldsymbol{x})} \frac{\partial r_\tau}{\partial \tau}(\boldsymbol{x}) - \frac{1}{q_\tau(\boldsymbol{x})} \frac{\partial q_\tau}{\partial \tau}(\boldsymbol{x}) \right) \mathrm{d}\boldsymbol{x} \tag{124}$$

$$= -\int_{\mathbb{R}^{d_x}} \boldsymbol{b}_{1-\tau}(\boldsymbol{x}) r_\tau(\boldsymbol{x}) \nabla \log \frac{r_\tau(\boldsymbol{x})}{q_\tau(\boldsymbol{x})} \mathrm{d}\boldsymbol{x} - \epsilon \int_{\mathbb{R}^{d_x}} r_\tau(\boldsymbol{x}) \left| \nabla \log \frac{r_\tau(\boldsymbol{x})}{q_\tau(\boldsymbol{x})} \right|^2 \mathrm{d}\boldsymbol{x} + \frac{\mathrm{d}}{\mathrm{d}\tau} \cancel{\int_{\mathbb{R}^{d_x}} r_\tau(\boldsymbol{x})\mathrm{d}\boldsymbol{x}}$$

$$- \int_{\mathbb{R}^{d_x}} \frac{r_\tau(\boldsymbol{x})}{q_\tau(\boldsymbol{x})} \mathrm{div}(\boldsymbol{b}_{1-\tau}(\boldsymbol{x}) q_\tau(\boldsymbol{x})) \mathrm{d}\boldsymbol{x}, \tag{125}$$

where we used integration-by-parts and (120) to arrive at the last line. To further simplify this expression, we note that

$$-\int_{\mathbb{R}^{d_x}} \boldsymbol{b}_{1-\tau}(\boldsymbol{x}) r_\tau(\boldsymbol{x}) \nabla \log \frac{r_\tau(\boldsymbol{x})}{q_\tau(\boldsymbol{x})} \mathrm{d}\boldsymbol{x} = -\int_{\mathbb{R}^{d_x}} \boldsymbol{b}_{1-\tau}(\boldsymbol{x}) r_\tau(\boldsymbol{x}) \nabla \log r_\tau(\boldsymbol{x}) \mathrm{d}\boldsymbol{x} + \int_{\mathbb{R}^{d_x}} \boldsymbol{b}_{1-\tau}(\boldsymbol{x}) r_\tau(\boldsymbol{x}) \nabla \log q_\tau(\boldsymbol{x}) \mathrm{d}\boldsymbol{x}$$

(126)

$$= -\int_{\mathbb{R}^{d_x}} \boldsymbol{b}_{1-\tau}(\boldsymbol{x}) \nabla r_\tau(\boldsymbol{x}) \mathrm{d}\boldsymbol{x} + \int_{\mathbb{R}^{d_x}} \boldsymbol{b}_{1-\tau}(\boldsymbol{x}) r_\tau(\boldsymbol{x}) \nabla \log q_\tau(\boldsymbol{x}) \mathrm{d}\boldsymbol{x} \qquad (127)$$

$$= \int_{\mathbb{R}^{d_x}} r_\tau(\boldsymbol{x}) \mathrm{div}(\boldsymbol{b}_{1-\tau}(\boldsymbol{x})) \mathrm{d}\boldsymbol{x} + \int_{\mathbb{R}^{d_x}} \boldsymbol{b}_{1-\tau}(\boldsymbol{x}) r_\tau(\boldsymbol{x}) \nabla \log q_\tau(\boldsymbol{x}) \mathrm{d}\boldsymbol{x}, \qquad (128)$$

and also

$$-\int_{\mathbb{R}^{d_x}} \frac{r_\tau(\boldsymbol{x})}{q_\tau(\boldsymbol{x})} \mathrm{div}(\boldsymbol{b}_{1-\tau}(\boldsymbol{x}) q_\tau(\boldsymbol{x})) \mathrm{d}\boldsymbol{x} = -\int_{\mathbb{R}^{d_x}} r_\tau(\boldsymbol{x}) \mathrm{div}(\boldsymbol{b}_{1-\tau}(\boldsymbol{x})) \mathrm{d}\boldsymbol{x} - \int_{\mathbb{R}^{d_x}} \frac{r_\tau(\boldsymbol{x}) \boldsymbol{b}_{1-\tau}(\boldsymbol{x})}{q_\tau(\boldsymbol{x})} \nabla q_\tau(\boldsymbol{x}) \mathrm{d}\boldsymbol{x} \qquad (129)$$

$$= -\int_{\mathbb{R}^{d_x}} r_\tau(\boldsymbol{x}) \mathrm{div}(\boldsymbol{b}_{1-\tau}(\boldsymbol{x})) \mathrm{d}\boldsymbol{x} - \int_{\mathbb{R}^{d_x}} \boldsymbol{b}_{1-\tau}(\boldsymbol{x}) r_\tau(\boldsymbol{x}) \nabla \log q_\tau(\boldsymbol{x}) \mathrm{d}\boldsymbol{x}. \qquad (130)$$

Hence, these two terms cancel out, giving us

$$\frac{\mathrm{d}}{\mathrm{d}t} \mathcal{KL}(r_\tau || q_\tau) = -\epsilon \int_{\mathbb{R}^{d_x}} r_\tau(\boldsymbol{x}) \left| \nabla \log \frac{r_\tau(\boldsymbol{x})}{q_\tau(\boldsymbol{x})} \right|^2 \mathrm{d}\boldsymbol{x}. \qquad (131)$$

Now, from our assumption that $\{p_t\}_{t \in [0,1]}$ and therefore $\{q_\tau\}_{\tau \in [0,1]}$ satisfies the log-Sobolev inequality (116a), we obtain the estimate

$$\frac{\mathrm{d}}{\mathrm{d}\tau} \mathcal{KL}(r_\tau || q_\tau) \leq -2\lambda\epsilon \mathcal{KL}(r_\tau || q_\tau). \qquad (132)$$

Then, applying Grönwall's inequality, we arrive at

$$\mathcal{KL}(r_\tau || q_\tau) \leq e^{-2\lambda\epsilon\tau} \mathcal{KL}(r_0 || q_0) \qquad (133)$$

$$= e^{-2\lambda\epsilon\tau} \mathcal{KL}(\hat{\pi}_n || \mathbb{P}_\infty), \qquad (134)$$

where we slightly abused notation and used the same notation $\mathcal{KL}(\cdot || \cdot)$ to denote the Kullback-Leibler divergences between two measures and their corresponding Lebesgue densities. Now, taking the limit $\tau \to 1$, we get

$$\mathcal{KL}(\tilde{\rho}_0^\epsilon || \rho_0) \leq e^{-2\lambda\epsilon} \mathcal{KL}(\hat{\pi} || \mathbb{P}_\infty), \qquad (135)$$

where we note that $\rho_0(\mathrm{d}\boldsymbol{x}) = q_1(\boldsymbol{x})\mathrm{d}\boldsymbol{x}$ and we defined $\hat{\rho}_0^\epsilon(\mathrm{d}\boldsymbol{x}) := r_1(\boldsymbol{x})\mathrm{d}\boldsymbol{x}$, which may be thought of as the latent representation of the predictive distribution $\hat{\pi}$ in noise space.

Next, we apply an identical argument for the guided sampling step of DAISI. That is, we now consider an interpolant $\{\boldsymbol{z}_t\}_{t \in [0,1]}$ bridging between $\rho_0 = \mathcal{N}(\boldsymbol{0}, \boldsymbol{I})$ and $\rho_1 = \mathbb{P}_\infty^{\boldsymbol{y}}$. As per discussion in Section A.4, the corresponding law $p_t^{\boldsymbol{y}}$ satisfies the transport equation

$$\frac{\partial p_t^{\boldsymbol{y}}}{\partial t}(\boldsymbol{x}) + \mathrm{div}(\boldsymbol{b}_t^{\boldsymbol{y}}(\boldsymbol{x}) p_t^{\boldsymbol{y}}(\boldsymbol{x})) = 0, \quad p_0^{\boldsymbol{y}}(\boldsymbol{x}) = \frac{\mathrm{d}\rho_0}{\mathrm{d}\boldsymbol{x}}(\boldsymbol{x}), \qquad (136)$$

with $\boldsymbol{b}_t^{\boldsymbol{y}}$ defined in (56). Now, consider the forward guided SDE

$$\mathrm{d}\boldsymbol{z}_t = \boldsymbol{b}_t^{\boldsymbol{y}}(\boldsymbol{z}_t)\mathrm{d}t + \epsilon \nabla \log p_t^{\boldsymbol{y}}(\boldsymbol{z}_t) + \sqrt{2\epsilon}\,\mathrm{d}W_t, \quad \boldsymbol{z}_0 \sim \hat{\rho}_0^\epsilon, \qquad (137)$$

whose marginal laws can be checked to solve the following Fokker-Planck equation

$$\frac{\partial r_t^{\boldsymbol{y}}}{\partial t}(\boldsymbol{x}) = \mathrm{div}\Big(-\boldsymbol{b}_t^{\boldsymbol{y}}(\boldsymbol{x}) r_t^{\boldsymbol{y}}(\boldsymbol{x}) + \epsilon\, r_t^{\boldsymbol{y}} \nabla \log(r_t^{\boldsymbol{y}}(\boldsymbol{x})/p_t^{\boldsymbol{y}}(\boldsymbol{x}))\Big), \quad r_0^{\boldsymbol{y}}(\boldsymbol{x}) = \frac{\mathrm{d}\hat{\rho}_0^\epsilon}{\mathrm{d}\boldsymbol{x}}(\boldsymbol{x}). \qquad (138)$$

Then, by a similar computation as before, we obtain

$$\frac{\mathrm{d}}{\mathrm{d}t}\mathcal{KL}(r_t^{\boldsymbol{y}}||p_t^{\boldsymbol{y}}) = -\epsilon \int_{\mathbb{R}^{d_x}} r_t^{\boldsymbol{y}}(\boldsymbol{x}) \left| \nabla \log \frac{r_t^{\boldsymbol{y}}(\boldsymbol{x})}{p_t^{\boldsymbol{y}}(\boldsymbol{x})} \right|^2 \mathrm{d}\boldsymbol{x}, \tag{139}$$

and again assuming that the conditional laws $\{p_t^{\boldsymbol{y}}\}_{t\in[0,1]}$ satisfy the uniform log-Sobolev inequality (116b) with uniform constant $\lambda$, we get

$$\frac{\mathrm{d}}{\mathrm{d}t}\mathcal{KL}(r_t^{\boldsymbol{y}}||p_t^{\boldsymbol{y}}) \leq -2\lambda\epsilon\mathcal{KL}(r_t^{\boldsymbol{y}}||p_t^{\boldsymbol{y}}) \tag{140}$$

$$\stackrel{\text{Grönwall}}{\Longrightarrow} \mathcal{KL}(r_t^{\boldsymbol{y}}||p_t^{\boldsymbol{y}}) \leq e^{-2\lambda\epsilon t}\mathcal{KL}(\hat{\rho}_0^{\epsilon}||\rho_0). \tag{141}$$

Taking the limit $t \to 1$, we get

$$\mathcal{KL}(\pi_{n,0,\epsilon}^{\text{DAISI}}||\mathbb{P}_\infty^{\boldsymbol{y}}) \leq e^{-2\lambda\epsilon}\mathcal{KL}(\hat{\rho}_0^{\epsilon}||\rho_0) \tag{142}$$

$$\stackrel{(117)}{\leq} e^{-4\lambda\epsilon}\mathcal{KL}(\hat{\pi}_n||\mathbb{P}_\infty), \tag{143}$$

which proves the inequality (117).

Now, from (131) and (139), we also see that when $\epsilon = 0$, we have that

$$\mathcal{KL}(\pi_{n,0,0}^{\text{DAISI}}||\mathbb{P}_\infty^{\boldsymbol{y}}) = \mathcal{KL}(\tilde{\rho}_0^{0}||\rho_0) = \mathcal{KL}(\hat{\pi}_n||\mathbb{P}_\infty), \tag{144}$$

which completes the proof. $\qquad\square$

*Remark* E.2. Inequality (117) describes how DAISI exponentially contracts the KL divergence to the posterior measure $\mathbb{P}_\infty^{\boldsymbol{y}}$ as $\epsilon \to \infty$. In other words, this shows how noise in the generative SDE progressively "erases" the discrepancy between the forecast measure $\hat{\pi}_n$ and the background measure $\mathbb{P}_\infty$. On the other extreme, when $\epsilon = 0$, this discrepancy (i.e., information about the forecast) is exactly preserved. We also note that the proof relies on the assumption that $t_{\min} = 0$, however, we expect similar behaviour to hold for $t_{\min} > 0$.

*Remark* E.3. The uniform log-Sobolev assumption (116) is a strong regularity assumption, indicating that the interpolant path $\{\rho_t\}_{t\in[0,1]}$ is smooth and does not have any irregularities, e.g., develop narrow spikes or heavy tails. While this is normally an unreasonable assumption for high-dimensional distributions arising in generative modelling, the information loss behaviour as $\epsilon \to \infty$ can be observed empirically.

