# OpenReview forum: "DAISI: Data Assimilation with Inverse Sampling using Stochastic Interpolants"
_ICML.cc/2026/Conference — ICML 2026 regular_

### Official Review · Reviewer_bC67 · 2026-02-23

**Soundness:** 3
**Presentation:** 4
**Significance:** 3
**Originality:** 2
**Overall Recommendation:** 4
**Confidence:** 5

**Summary:**

This paper introduces DAISI, an iterative algorithm based on generative models to approximate the filtering distribution $p(x_n | y_{1:n})$ of a dynamical system. To do so, the authors first train a stochastic interpolant to learn a prior $\mathbb{P}_{\infty}$ of a dynamical system. The learned interpolant is then used to generate latents from forecast ensemble members, which serve as initial conditions to a guided forward stochastic differential equation that incorporates the current observation with training-free guidance methods.

**Compliance With Llm Reviewing Policy:**

Affirmed.

**Final Justification:**

The paper is very well written and presents strong experimental results. Therefore, although the resulting distribution is theoretically different from the true filtering distribution, I believe this is a strong paper and I maintain my initial score of weak accept.

**Key Questions For Authors:**

1. Have the authors tested their method on large-scale dynamical systems (i.e., with dimension ≥ $10⁶$)? If so, how were the hyperparameters ($t_{\text{min}}$ and $\epsilon$) efficiently optimized in such high-dimensional settings?
2. Would it be possible to add correction terms (in forward and/or backward equations for example) to correct the bias between the obtained distribution and the true filtering distribution?

**Limitations:**

The authors explicitly acknowledge the limitations of their approach. In particular, Section 3.1 examines the discrepancy between the distribution produced by the algorithm and the true filtering distribution.

**Strengths And Weaknesses:**

Strengths:
* The paper is very well written, easy to follow and technically sound.
* It addresses the problem of data assimilation, which aims to estimate the state of a dynamical system from observations. Although less widely known than forecasting, this problem is of paramount importance and is solved daily by weather forecasting centers around the world to produce high-quality forecasts.
* The proposed algorithm is flexible: it can be applied with both classical numerical solvers and deep learning–based forecasting models.


Weaknesses:
* The proposed algorithm does not sample from the filtering distribution $p(x_n \mid y_{1:n})$. This discrepancy can be minimized by optimizing the two hyperparameters $t_{\text{min}}$ and $\epsilon$, which can be costly in large dimensional systems.
* At each filtering step, the computational cost per ensemble member can be substantial, as it requires performing a forecast, solving a reverse equation, and solving a guided forward equation.
* The use of training-free posterior sampling methods can introduce bias, especially since the method employed in this work (MMPS, Rozet et al.) was originally developed for linear observation operators. This limitation may reduce its effectiveness when applied to real-world, nonlinear observation operators.

---

> ### Author Rebuttal · Authors · 2026-03-31
>
> We thank the reviewer for their time and thoughtful feedback. We appreciate their positive comments on the clarity, accessibility, and technical soundness of the paper. We also share the view that filtering is a problem of central importance, and we are grateful that the reviewer recognizes our contribution. Finally we are happy that the reviewer highlighted the flexibility of DAISI, where the guidance method and forecasting model can be easily swapped. We now address the points raised.
>
> __[W1] Lack of theoretical guarantees__
>
> We agree that the lack of theoretical guarantees is a limitation of our approach. However, this challenge is generally shared by other practical DA methods. As a result, our justification is primarily empirical: we evaluate DAISI across multiple systems, observation regimes, baselines, and ablations. Although the method is biased in principle, we find that this bias is often not prohibitive in the settings we consider. Please also refer to our response to reviewer D5Cb, where we discuss in further detail our theoretical limitations.
>
> __[W2] High computational cost__
>
> We agree that DAISI inherits the high inference cost of ODE/SDE-based generative models, driven by the number of function evaluations required to integrate the generative SDE. However, compared to other generative baselines, DAISI is relatively efficient.
>
> In SDA, assimilation is performed over a window of states, which is set to five in our experiments. While DAISI has a two-stage process and is therefore not directly comparable, the guidance stage in DAISI operates on a single state, avoiding the $5\times$ slowdown incurred by SDA. Moreover, the inversion step in DAISI is significantly faster than the guidance step, since it does not require computing expensive gradients, making the overall computational cost quite manageable. Part of the speedup can be attributed to having $t_{\min}>0$, which shortens the integration range, requiring fewer steps to solve.
>
> **[W1/2] *"optimizing the two hyperparameters $t_{\min}$ and $\epsilon$, which may be costly in high-dimensional systems."***
>
> While we understand the scepticism towards hyperparameter sensitivity, we would like to clarify that DAISI was not especially brittle in our experiments. We have provided an extended discussion on this in the answer to reviewer vTYg.
>
> __[W3] Biased guidance methods__
>
> We agree that training-free guidance methods such as MMPS rely on approximations (e.g., local linearization), which can introduce bias, particularly for nonlinear observation operators. However, in our experiments we find that MMPS still provides effective guidance even under strongly nonlinear observations (e.g., the saturating setting), suggesting that this approximation is not prohibitive in practice. We also note that similar Gaussian or local approximations are common across gradient-based guidance methods. A promising direction for future work is to reduce this bias by replacing the local approximation with more accurate estimators of the guidance term, for example using stochastic flow maps [1].
>
> Compared to LETKF, the advantage of DAISI lies primarily in improved assimilation performance in sparse and nonlinear settings. However, as outlined in Sec. 3.2, DAISI scales linearly with ensemble size, whereas the covariance estimation in LETKF scales cubically and cannot be parallelized.
>
> Finally, for large-scale dynamical systems, performing data assimilation in a latent space could significantly reduce computational costs, which we consider a promising direction for future work [2].
>
> __[Q1] _"Have the authors tested their method on large-scale dynamical systems (e.g., dimension $\geq 10^6$)?"___
>
> Our high-dimensional experiment has $\geq10^5$ dimensions, making it a reasonably large system. We have not yet applied DAISI to even larger systems (e.g., ERA5), but given the success of diffusion-based models in forecasting and data assimilation for such systems, we believe this is a promising direction for future work.
>
> __[Q2] _"Would it be possible to add correction terms (in the forward and/or backward equations) to reduce the bias between the obtained distribution and the true filtering distribution?"___
>
> At present, incorporating such correction terms appears technically challenging. A promising direction would be to use DAISI as a preconditioner for a more accurate posterior sampler. In this setting, DAISI could be used to initialize high-quality particles, which are then refined through a subsequent correction stage toward the true filtering distribution. We believe this is an interesting direction for future work, but it is beyond the scope of the current paper.
>
> [1] Holderrieth, Peter, et al. "Diamond Maps: Efficient Reward Alignment via Stochastic Flow Maps." arXiv:2602.05993 (2026).
>
> [2] Andry, Gérôme, et al. "Appa: Bending weather dynamics with latent diffusion models for global data assimilation." Machine Learning and the Physical Sciences Workshop, NeurIPS. (2025).

---

> > ### Author Rebuttal · Reviewer_bC67 · 2026-04-03
> >
> > I thank the authors for this detailed rebuttal. Like reviewers vXKB and D5Cb, I remain somewhat skeptical that DAISI does not sample from the true filtering distribution, and that the theoretical gap to this distribution depends on parameters that must be tuned. However, as the authors clearly explained in their response to reviewer vTYg, most filtering methods also rely on hyperparameters and, at the end of the day, if DAISI performs better than more theoretically grounded approaches, it deserves to be highlighted.
> >
> > For these reasons, I maintain my recommendation to accept the paper.

---

> > > ### Author Response · Authors · 2026-04-08
> > >
> > > We greatly appreciate the reviewer’s support and continued recommendation for acceptance.
> > >
> > > In the paper, we aim to be transparent about what DAISI samples from and how this differs from the true filtering distribution. Scalable algorithms for unbiased filtering are rare, and in high-dimensional settings some level of bias is typically unavoidable. Importantly, this also applies to the baselines we consider, which generally do not target (i.e. samples approximately from) the exact filtering distribution.
> > >
> > > To the best of our knowledge, **particle filters** (and related interacting particle methods, such as particle flow filters [1]) are the only approaches that asymptotically sample from the true filtering distribution in nonlinear, non-Gaussian settings. However, particle filters suffer from the curse of dimensionality and perform poorly in the high-dimensional regimes we consider. As a result, most practical data assimilation methods, including generative methods and EnKF-based approaches, trade asymptotic consistency for bias in order to remain computationally tractable. DAISI belongs to this class of methods.
> > >
> > > More specifically, the baselines we consider introduce approximations of a similar nature:
> > >
> > > **FlowDAS** targets the product $\prod_{s \leq t} p(x_s|y_t, x_{s-1})$. The marginal at time $t$ of this can differ significantly from the true filtering distribution $p(x_t|y_{1:t})$, and for strong dependencies between state variables it can even collapse to the prior. From a particle filtering perspective, this corresponds to sampling from the product of locally optimal proposals, but without resampling. Note that this hold even with "perfect guidance" in FlowDAS - in practice we add a further approximation error by only approximately sampling from $p(x_s|y_t, x_{s-1})$ at each time step.
> > >
> > > **SDA** was originally designed to target the smoothing distribution $p(x_{1:t}|y_{1:t})$, which coincides with the filtering distribution only at the final time. Adapting it to online filtering requires truncating the observation window , introducing a bias similar in nature (though typically less severe) to that of FlowDAS. Additionally, SDA is not guaranteed to recover the true smoothing distribution, as it relies on a learned stationary space-time prior $p(x_{1:t})$ rather than the dynamical prior $p(x_{1:t}|x_0)$ induced by the numerical model.
> > >
> > > **EnSF** constructs an empirical score from particles and performs guided sampling, similarly to DAISI. Running the dynamics to completion recovers the bootstrap particle filter [2], while early stopping (used in practice to avoid particle collapse) introduces additional bias, akin to rejuvenation techniques.
> > >
> > > **EnKF** is the only baseline with theoretical guarantees, but these are limited to near-Gaussian regimes and degrade in nonlinear, non-Gaussian settings [3]. Practical variants such as LETKF are primarily motivated by empirical performance rather than stronger guarantees.
> > >
> > > Overall, our goal is to be transparent and highlight that many practical filtering algorithms involve approximations, and to position DAISI within this broader context.
> > >
> > > [1] Daum, Fred, and Jim Huang. "Nonlinear filters with log-homotopy." Signal and Data Processing of Small Targets 2007. Vol. 6699. SPIE, 2007.
> > >
> > > [2] Transue, Taos, et al. "Flow Matching for Efficient and Scalable Data Assimilation." arXiv preprint arXiv:2508.13313 (2025).
> > >
> > > [3] Calvello, Edoardo, et al. "Accuracy of the ensemble Kalman filter in the near-linear setting." SIAM Journal on Numerical Analysis 64.2 (2026): 391-429.

---

### Official Review · Reviewer_vTYg · 2026-03-11

**Soundness:** 3
**Presentation:** 3
**Significance:** 3
**Originality:** 4
**Overall Recommendation:** 5
**Confidence:** 4

**Summary:**

The authors use a combination of a backward SDE to draw forecast distributions closer to the prior in noised space, then use a conditional observation guided SDE to then denoise the noised forecast to generate the posterior. They test on SQG simulations, comparing against FlowDAS, LETKF, and EnSF.

**Compliance With Llm Reviewing Policy:**

Affirmed.

**Final Justification:**

See comments.  I find most of my concerns sufficiently addressed, though other reviewers bring up some good points with theoretical guarantees, etc, so I'm not sure about bringing the score up to a 6, but I think it is a solid piece of contribution, so I argue for acceptance.

**Key Questions For Authors:**

See weaknesses.

[1] Questions regarding complexity of hyperparameter tuning.

[2] Question regarding sparsity.

**Limitations:**

Yes.

**Strengths And Weaknesses:**

Strengths:

The authors show that their method outperforms the other methods on CRPS metrics in the main text, and RMSE results (both sample wise and

Paper is written well and clear, idea is very interesting.

The analysis of $t_{\text{min}}$ and $\epsilon$ is solid, and the 1D distribution is a good illustration to help readers understand the point.


Weaknesses:
Personally, it might be clearer if the definition of the backward SDE and the forward SDE were flipped, given standard score-based sampling literature, where the forward SDE denotes the noising stage, and the backwards SDE indicates the denoising step.

The authors state that there is an intermediate $t_{\text{min}}$ and $\epsilon$ such that the DAISI distribution matches the filtering distribution the best. How costly is the hyperparameter tuning for this problem? Does it need to be reconfigured when applying to a new example?

Are there any results of when DAISI is applied to some extremely sparse observations, e.g. weather forecasting with minimial observations? The experiments here only go down to 5\% for sparsity, which in the scale of standard data assimilation problems is not particularly sparse. However, it is for the SQG problem, which is quite difficult in the first place, so I'm not expecting that any model would do particularly well with <1\% observation sparsity.

---

> ### Author Rebuttal · Authors · 2026-03-31
>
> We thank the reviewer for their time and positive feedback. We appreciate the comments on the clarity of the paper and the core idea behind DAISI. We are also glad that the reviewer found our analysis of $t_{\min}$ and $\epsilon$, as well as the 1D illustrations, to be helpful. We now address the questions raised in the review.
>
> __[W1] Choice of time parameterisation__
>
> We thank the reviewer for pointing out our choice of time direction in the SDEs. In this paper, we adopt the convention used in the stochastic interpolants and flow matching literature, where $t=0$ corresponds to noise and $t=1$ to data. This is the opposite of the convention commonly used in the diffusion and score-based literature, and we will clarify this choice explicitly in the paper.
>
> __[W2], [Q1] Hyperparameter tuning cost__
>
> While we understand the scepticism towards hyperparameter sensitivity, we would like to clarify that DAISI was not especially brittle in our experiments. In Fig. 11 we see that there is quite a broad range of suitable  $t_{\min}\in[0.2,0.4]$, indicating that a coarse search is sufficient to achieve good results. Further, in our SQG experiments, we did not tune $\epsilon$ at all as it did not affect performance significantly, which can be seen in Fig. 10.
>
> We also note that most practical DA methods have hyperparameters, and that DAISI is relatively robust in comparison. For example, SDA required careful tuning of guidance strength and the number of Langevin steps, to the point where we didn't get it to work well for nonlinear observations. By contrast, DAISI's tuning primarily helped to optimize performance rather than ensure stability. Similary, LETKF is known to be sensitive to its inflation and localization parameters which are challenging to tune [1].
>
> In our SQG experiments, $t_{\min}$ and the MMPS guidance strength $\zeta$ was tuned on a single 20-step trajectory using 20 ensemble members. We tested roughly 20 combinations for each experiment, with each one taking \~6.5 minutes on a single A100 GPU, for a total tuning time of just over 2 hours per experiment.
>
> __[W3], [Q2] Extremely sparse observations__
>
> We thank the reviewer for raising the question of extreme sparsity. While the main paper reports results down to 5\% observation coverage, our high-dimensional experiments are effectively much sparser. Specifically, we use an averaging observation operator similar to lower-resolution sensing.
>
> First, the $256\times256$ state is partitioned into a coarse $64\times64$ grid, where we then observe 5\% of these coarse cells at random (with added noise), corresponding to an effective observation ratio of 0.3125\% relative to the full state dimension. This number is somewhat misleading, since each observation aggregates information over a spatial region and is therefore more informative than a single pixel measurement. Nevertheless, it provides a useful indication of performance under extremely sparse sensing.
>
> As you pointed out, the SQG problem is particularly challenging, and pushing to even sparser regimes ($<$1\%) caused all models to struggle to track the state accurately. In less chaotic settings, guided diffusion models have demonstrated successful reconstructions from extremely sparse observations [2]. This suggests that DAISI, which leverages a similar generative prior, may scale favorably to such regimes, which we consider an important direction for future work.
>
> [1] Bannister, R. N. "A review of operational methods of variational and ensemble-variational data assimilation, QJ Roy. Meteor. Soc., 143, 607–633." 2017,
>
> [2] Amorós-Trepat, Marc, et al. "Guiding diffusion models to reconstruct flow fields from sparse data." Physics of Fluids 38.1 (2026).

---

> > ### Author Rebuttal · Reviewer_vTYg · 2026-04-02
> >
> > Addressed.

---

> > > ### Author Response · Authors · 2026-04-08
> > >
> > > We sincerely thank the reviewer for their support and for the constructive feedback that has helped improve the paper.

---

### Official Review · Reviewer_D5Cb · 2026-03-13

**Soundness:** 2
**Presentation:** 3
**Significance:** 3
**Originality:** 3
**Overall Recommendation:** 4
**Confidence:** 4

**Summary:**

This work examines a fundamental question: how to combine a strong stationary generative prior with sequential forecast information so that filtering remains flexible under sparse, noisy, and nonlinear observations. The paper proposes DAISI, a filtering method that first forecasts an ensemble with any chosen dynamics model, then approximately inverts each forecast sample through a backward stochastic-interpolant SDE to obtain latent states, and finally performs observation-conditioned forward sampling from those latents to produce the analysis ensemble. Overall, this manuscript's key contribution consists of turning a pre-trained unconditional generative prior into a modular sequential data-assimilation mechanism without retraining the generative model at every assimilation step. The paper also provides heuristic analysis for the roles of 𝑡_min and 𝜖, a KL-contraction proposition for one limiting regime, and empirical results on Lorenz-63, SQG, and SEVIR showing advantages over several baselines in sparse and nonlinear settings.

**Compliance With Llm Reviewing Policy:**

Affirmed.

**Final Justification:**

I find the presentation and the results delivered by this paper really interesting, and most of my concerns resolved.

**Key Questions For Authors:**

Can you formalize the approximation error to the true filtering distribution beyond the limiting intuition in Section 3.1?

How sensitive are results to hyperparameter selection, and how expensive is tuning in wall-clock and number of validation trajectories?

Can you provide a cleaner apples-to-apples comparison against FlowDAS using matched forecast initialization and matched forecast models in the main benchmark tables?

Why should the strong assumptions behind Proposition 3.1 be expected to say something useful in the actual high-dimensional settings?

Do the claimed gains persist under more realistic observation processes and modest model misspecification?

**Limitations:**

No. The paper does mention inference cost and compatibility issues with iterative guidance, which is good, but the limitations discussion should be broader and more concrete. In particular, it should explicitly discuss: dependence on strong hyperparameter tuning; the lack of a correctness guarantee for filtering; potential brittleness to prior mismatch when the learned stationary prior 𝑃∞ is inaccurate; and fairness/transfer limitations of the current real-world evaluation. The societal-impact discussion is also quite generic and mostly positive; it should acknowledge risks from overconfident or miscalibrated state estimates in safety-critical forecasting settings.

**Strengths And Weaknesses:**

The paper has a clear and useful high-level idea: instead of retraining a predictive generative model online, it keeps a stationary prior and injects forecast information by inverse sampling in latent space before guided conditional generation. That is a sensible design, and the ablation replacing the backward SDE with SDEdit makes the central mechanism more credible. The experimental section is broader than many papers in this area: low-dimensional sanity checking on Lorenz-63, controlled nonlinear PDE-style benchmarks on SQG, and a real radar nowcasting dataset. The paper also compares to both classical and generative baselines, and the appendix includes useful ablations, including no inversion, SDEdit replacement, and swapping in FlowDAS’s forecast model.

The theory is weaker than the framing sometimes suggests. The central question in Section 3.1 is literally whether DAISI samples from the filtering distribution, and the paper’s answer is essentially 'not exactly, but maybe approximately after tuning'. The limiting arguments for $t_{\min}\to 0$ and $t_{\min}\to 1$ are intuitive and helpful, but they are not a correctness result for the actual algorithm used in experiments. Proposition 3.1 is also narrow: it only covers the $t_{\min}=0$ case, assumes uniform log-Sobolev inequalities along both unconditional and conditional interpolant paths, and the appendix itself admits this is a strong and generally unrealistic assumption for high-dimensional generative-model distributions. So the theoretical section is better read as qualitative intuition about information loss versus noise injection, not as a rigorous guarantee of filtering accuracy. That substantially limits the soundness of the theoretical claims.

There seems to be a notable mismatch between the simplified analysis and the implementation details. Section 3.1 analyzes constant $\epsilon_t\equiv\epsilon$, while Appendix B says the implementation uses $\epsilon_t=\epsilon(1-t)$ to avoid numerical issues because the score diverges near $t=1$. That implementation choice is sensible, but it further weakens the connection between the stated proposition and the actual algorithm. I would have liked a clearer statement in the main paper that the proposition is only a stylized result for a modified process.

The experiments are good in breadth, but several aspects are confusing. First, the method is quite tuning-sensitive: even the L63 section shows that performance depends materially on $t_{\min}$ and $\epsilon$, and the appendix reports a per-experiment greedy grid search for DAISI hyperparameters. Second, the FlowDAS comparison is somewhat awkward. In the main SQG setup, FlowDAS is not initialized the same way as the other methods and uses its own learned autoregressive forecast model, whereas the others use the numerical model from noisy initial ensembles. The paper does try to address this with an ablation replacing DAISI’s forecast with the FlowDAS forecast model, which is helpful, but the main table still mixes baselines with unequal forecasting setups. Third, the strongest real-world experiment is still limited: SEVIR uses the same linear Gaussian observation setup as FlowDAS and uses a pre-trained forecasting model from prior work. This is a useful transfer test, but it is not yet convincing evidence that DAISI is robust in genuinely operational DA conditions with misspecification, asynchronous observations, or more realistic sensor operators. Likewise, the high-dimensional SQG experiment is only a single 256×256 setting and the paper itself attributes some artifacts to lack of tuning at that resolution.

The paper is generally well written and the high-level idea is easy to follow. Figure 1 is effective. The method section is organized sensibly. My main presentation issue is that the theory/intuition/correctness distinctions are blurrier than they should be. The paper should more explicitly separate the exact filtering target, heuristic interpolation argument, and what is actually proven under strong assumptions. Some baseline-comparison details are also deferred too far into the appendix for claims that matter in the main table.

---

> ### Author Rebuttal · Authors · 2026-03-31
>
> We thank the reviewer for their thoughtful feedback and appreciate their positive assessment of the paper’s clarity, design, and performance in sparse and nonlinear settings, as well as their recognition of the breadth of our experiments and ablations. We address the main concerns below.
>
> __[W1], [Q1], [Q4] Theoretical weakness__
>
> We thank the reviewer for this careful reading and agree that Section 3.1 should by no means be interpreted as a correctness result. Our intention was not to claim that DAISI is theoretically guaranteed to sample from the filtering distribution; rather, the purpose was to provide intuition for two points: (1) why inverse sampling followed by guided sampling is inherently biased (this was not clear to us at first), and (2) why tuning $t_{\min}$ and $\epsilon$ can affect the bias in practice. We agree that our current presentation may suggest a stronger claim than warranted, and we will make this limitation explicit in the revision, stating clearly that this section is best viewed as providing intuition for why we introduce the hyperparameters $t_{\min}$ and $\epsilon$ to help mitigate bias.
>
> We also agree that Proposition 3.1 is narrow in scope. Our main purpose for including this result was to place our intuition that adding noise via $\epsilon$ induces information loss on a more theoretical footing. However, we agree that this is not central to the discussion of the paper; hence, in the revision, we will move this proposition to the appendix and use the free space to instead bring forward experimental details that are important to the main experimental claims.
>
> That said, we believe the lack of theoretical guarantees does not undermine the main contribution of the paper. Strong correctness guarantees are typically unavailable for practical DA methods in high-dimensional settings, not least in the context of generative models, so our goal is to provide careful empirical evidence for DAISI across a broad range of regimes. Our experiments cover multiple systems, observation regimes, baseline comparisons, and ablations to assess the impact of key design choices. The resulting evidence suggests that although our method is biased in principle, this bias is not prohibitive in the settings we study and can be mitigated through tuning.
>
> __[W2], [Q2] Hyperparameter sensitivity__
>
> While we understand the scepticism towards hyperparameter sensitivity, we would like to clarify that DAISI was not especially brittle in our experiments. We have provided an extended discussion on this in our response to reviewer vTYg.
>
> __[W3], [Q3] Mismatched FlowDAS evaluation__
>
> We agree that the main-table comparison to FlowDAS is not fully apples-to-apples, as our intent was to compare each method in the form in which it is naturally used. However, it should be noted that the mismatched setting we have considered makes it _more favourable for FlowDAS_, since FlowDAS requires six frames of ground truth for initalization – a more informative start than the other models. If we chose to run all models with the FlowDAS forecast model, then we would have to use the same ground truth initialization for all models, and we did not believe this was suitable for realistic DA experiments.
> We have thus deferred this setting to the appendix, with results displayed in Table 14, which shows, as also mentioned by the reviewer, that swapping to the FlowDAS forecast model in DAISI _has almost no effect on the overall performance_. We also note that this setting _reuses the same hyperparameters_ that we used for the numerical forecast model, highlighting the stability of DAISI. We agree that this point was not explained in the main text; hence, we plan to expand on this in the updated paper.
>
> __[W4], [Q5] Realistic observations__
>
> We share the reviewer's regard for realistic operational DA, and agree that this is not fully captured in our current experiments. The SEVIR experiment was chosen to enable a direct comparison to FlowDAS, with the added benefit of incorporating real-world data. At the same time, we acknowledge that this setup remains heavily simplified compared to operational DA systems. Moving to fully realistic settings introduces additional challenges, in particular the need to obtain and work with realistic observation operators. We view this as an important direction for future work and will expand the discussion of this limitation in the revised paper, alongside a broader treatment of other limitations and societal impacts.
>
> __[Q5] Model mis-specification__
>
> We do not provide a dedicated study of model misspecification and will clarify this as a limitation. As preliminary evidence, the forecast-swap ablation where the numerical model is replaced by the learned FlowDAS forecast introduces model error, yet we observe little impact on performance. This suggests that DAISI is not highly sensitive to the choice of forecast model in our experiments, though we emphasize that this is not a general robustness guarantee.

---

> > ### Author Rebuttal · Reviewer_D5Cb · 2026-04-02
> >
> > I have read the rebuttal carefully and appreciate the thoughtful and candid response. The clarification that Section 3.1 is intended as intuition rather than a correctness result is helpful, and I agree this should be made more explicit in the paper.
> >
> > My main concerns, however, are only partially resolved. On the theoretical side, the rebuttal clarifies the intended scope of the analysis but does not change my core concern that the paper does not provide a result connecting the practical DAISI procedure to the true filtering distribution under realistic conditions. I do not view the absence of such a guarantee as fatal in this area, but it does mean the work should primarily be assessed as an empirically motivated method.
> >
> > On the experimental side, the explanation regarding the FlowDAS comparison is reasonable and the forecast-swap ablation is helpful. That said, I still believe a more directly matched comparison should be central rather than appendix-only.
> >
> > Overall, the rebuttal improves the framing of the work, but it does not materially change my assessment of the remaining core issues.

---

> > > ### Author Response · Authors · 2026-04-08
> > >
> > > We greatly appreciate the reviewer's comments and are glad that some issues were clarified. We will use their suggestions to reframe the paper and address the remaining concerns below.
> > >
> > > **[W3] Mismatched FlowDAS evaluation**
> > >
> > > After further consideration, we agree with the reviewer that a more directly matched comparison with FlowDAS is important for a fair and interpretable evaluation. To address this, we will move the DAISI forecast-swap ablation to the main paper where it can be directly compared to FlowDAS. We believe this revision will improve the clarity and fairness of the comparison, and we thank the reviewer for highlighting this point.
> > >
> > > **[W1] Theoretical weakness**
> > >
> > > In the paper, we aim to be transparent about what DAISI samples from and how this differs from the true filtering distribution. Scalable algorithms for unbiased filtering are rare, and in high-dimensional settings some level of bias is typically unavoidable. Importantly, this also applies to the baselines we consider, which generally do not target (i.e. samples approximately from) the exact filtering distribution.
> > >
> > > To the best of our knowledge, **particle filters** (and related interacting particle methods, such as particle flow filters [1]) are the only approaches that asymptotically sample from the true filtering distribution in nonlinear, non-Gaussian settings. However, particle filters suffer from the curse of dimensionality and perform poorly in the high-dimensional regimes we consider. As a result, most practical data assimilation methods, including generative methods and EnKF-based approaches, trade asymptotic consistency for bias in order to remain computationally tractable. DAISI belongs to this class of methods.
> > >
> > > More specifically, the baselines we consider introduce approximations of a similar nature:
> > >
> > > **FlowDAS** targets the product $\prod_{s \leq t} p(x_s|y_t, x_{s-1})$. The marginal at time $t$ of this can differ significantly from the true filtering distribution $p(x_t|y_{1:t})$, and for strong dependencies between state variables it can even collapse to the prior. From a particle filtering perspective, this corresponds to sampling from the product of locally optimal proposals, but without resampling. Note that this hold even with "perfect guidance" in FlowDAS - in practice we add a further approximation error by only approximately sampling from $p(x_s|y_t, x_{s-1})$ at each time step.
> > >
> > > **SDA** was originally designed to target the smoothing distribution $p(x_{1:t}|y_{1:t})$, which coincides with the filtering distribution only at the final time. Adapting it to online filtering requires truncating the observation window , introducing a bias similar in nature (though typically less severe) to that of FlowDAS. Additionally, SDA is not guaranteed to recover the true smoothing distribution, as it relies on a learned stationary space-time prior $p(x_{1:t})$ rather than the dynamical prior $p(x_{1:t}|x_0)$ induced by the numerical model.
> > >
> > > **EnSF** constructs an empirical score from particles and performs guided sampling, similarly to DAISI. Running the dynamics to completion recovers the bootstrap particle filter [2], while early stopping (used in practice to avoid particle collapse) introduces additional bias, akin to rejuvenation techniques.
> > >
> > > **EnKF** is the only baseline with theoretical guarantees, but these are limited to near-Gaussian regimes and degrade in nonlinear, non-Gaussian settings [3]. Practical variants such as LETKF are primarily motivated by empirical performance rather than stronger guarantees.
> > >
> > > Overall, our goal is to be transparent and highlight that many practical filtering algorithms involve approximations, and to position DAISI within this broader context.
> > >
> > > [1] Daum, Fred, and Jim Huang. "Nonlinear filters with log-homotopy." Signal and Data Processing of Small Targets 2007. Vol. 6699. SPIE, 2007.
> > >
> > > [2] Transue, Taos, et al. "Flow Matching for Efficient and Scalable Data Assimilation." arXiv preprint arXiv:2508.13313 (2025).
> > >
> > > [3] Calvello, Edoardo, et al. "Accuracy of the ensemble Kalman filter in the near-linear setting." SIAM Journal on Numerical Analysis 64.2 (2026): 391-429.

---

### Official Review · Reviewer_vXKB · 2026-03-16

**Soundness:** 2
**Presentation:** 4
**Significance:** 3
**Originality:** 3
**Overall Recommendation:** 5
**Confidence:** 4

**Summary:**

This work studies the use of stochastic interpolants to perform filtering of the state of dynamical systems given observations through time, a special case of the data assimilation problem. In this space, previous approaches typically use a generative/diffusion model of the dymamics (e.g. a diffusion model that forecasts the next state given the previous one(s) or one that generates trajectories) that can be guided with observations. The authors propose a different approach, where the generative model (here formulated as a stochastic interpolant) represents the background distribution over states. Filtering is then performed in two steps, resembling the classical prediction and update steps of Bayesian filtering: (1) a forecasting model (whose origin is not imposed by the method) pushes the ensemble members of the current time step to the next one, and (2) the new ensemble members are "corrected" with respect to the observation. The latter correction is performed with a novel "inverse sampling" method, which consists in encoding each ensemble member to the base distribution of the stochastic interpolant, then decoding it conditionaly on the observation (using gradient-based guidance methods). The authors argue that this method can achieve a good approximation to the target filtering distribution, by tuning its two hyperparameters, which they motivate and provide intuitive explanation for. The method is applied to three benchmarks, and demonstrates competitive results (qualitative and quantitative) against a relevant selection of baselines.

**Compliance With Llm Reviewing Policy:**

Affirmed.

**Key Questions For Authors:**

1. Are there classes of forecasting / background models for which there exist theoretical guarantees for inverse sampling?
2. In your experiments, the hyperparameters are tuned once for each dataset. Would it be possible or sensible to tune these on a per-observation basis? Maybe even online. How would you tackle this?

**Limitations:**

See strenght and weaknesses.

**Strengths And Weaknesses:**

The manuscript is very well written. The problem statement is well formalized, the contributions are explicit, the method is well motivated and put in perspective with respect to the literature, and the selection of baselines is sound and representative. My only nitpick about presentation would be the overly-mathy formalization, but that's a question of taste.

Regarding the "inverse sampling" method itself, while the authors provide some intuitions and some formal analysis of the two hyperparameters, it remains an approximation for which no guarantees of convergence towards the proper filtering distribution exist. I suggest to highlight this limitation clearly in the discussion, while avoiding vague sentences like "suggesting the existence of" (line 245). The presence of hyperparameters that should be tuned can also be seen as a limitation, which is not mentioned in the manuscript. While the authors present the compatibility of DAISI with numerical and ML-based forecasts as an advantage, the need for two models (forecasting and background) is a drawback.

Nevertheless, the approach remains interesting, novel to my knowledge, and seems to hold well against baselines, some of which present theoretical guarantees. The experiments are extensive, covering both low-dimensional systems for which a ground-truth is available and high-dimensional systems with challenging dynamics. I appreciate the extensive appendix section and additional experiments, that showed a great level of care from the authors.

For these reasons, I recommended acceptance of this submission, despite the limitations concerning the theoretical grounding of DAISI.

---

> ### Author Rebuttal · Authors · 2026-03-31
>
> We thank the reviewer for their careful reading and positive feedback. We are glad that the reviewer found the paper well-written, the idea novel, and the experimental evaluation extensive and competitive. We address the main points below.
>
> __[W1] Presentation issue__
>
> We appreciate the comment regarding the math-heavy presentation. Our goal was to make the analysis in Section 3.1 precise, but we agree that parts of the exposition can be simplified for clarity. We will revise the presentation and remove vague statements (e.g., “suggesting the existence of”) to more clearly distinguish intuition from formal results.
>
> __[W2] Lack of theoretical guarantees__
>
> We agree that DAISI is an approximate filtering procedure without general convergence guarantees to the exact filtering distribution. Hence, our main means for justifying our method is empirical in nature: we evaluate extensively across multiple systems, observation settings, baselines, and ablations to demonstrate that the bias in DAISI is not detrimental with proper tuning.
> We will make this limitation more explicit in the discussion. Please also find a longer discussion about this point in our response to reviewer D5Cb.
>
> __[W3] Presence of hyperparameters__
>
> While we understand the scepticism towards hyperparameter sensitivity, we would like to clarify that DAISI was not especially brittle in our experiments. We have provided an extended discussion on this in our response to reviewer vTYg.
>
> __[W4] Drawback of having two models__
>
> We agree that requiring two models can be viewed as a limitation and will clarify this point in the paper. At the same time, in many large-scale DA systems, a forecasting model is already available and therefore does not require additional training. Separating the two components also provides practical benefits: DAISI can directly leverage ongoing improvements in forecasting models (e.g., recent advances in AI-based weather prediction), and can flexibly adapt to changes in time-stepping without retraining. Additionally, decoupling the DA model from the forecasting model allows for the flexibility to use either physics-based or AI-based forecasting systems.
>
> We also note that approaches based on directly guiding a forecast model (e.g., FlowDAS and SDA (filtering)) are themselves approximate and generally do not sample from the filtering distribution. Empirically, we find that this separation leads to improved performance compared to single-model baselines in our experiments, particularly in nonlinear observation settings.
>
> __[Q1] _"Are there classes of forecasting / background models for which there exist theoretical guarantees for inverse sampling?"___
>
> At present, we are not aware of non-trivial and practically relevant settings (i.e., beyond simplified or limiting cases) where DAISI yields exact filtering.
>
> __[Q2] _"Would it be possible or sensible to tune these [hyperparameters] on a per-observation basis?"___
>
> Based on our experiments, we did not find clear evidence that tuning $t_{\min}$ or $\epsilon$ online would provide significant benefits. However, adapting the guidance strength based on the observation is a promising direction. To some extent, this is already explored in guidance methods such as DPS, where the guidance strength is proportional to $1/p(y|x)$.

---

> > ### Author Rebuttal · Reviewer_vXKB · 2026-04-02
> >
> > I thank the authors for their answer. My main concern, shared by reviewers D5Cb and bC67, remains that the algorithm does not sample from the filtering distribution, even in the infinite compute regime. This should be made very clear by the authors, which they agree to in the rebuttal.
> >
> > Nevertheless, I still appreciate this submission and continue to argue for acceptance.

---

> > > ### Author Response · Authors · 2026-04-08
> > >
> > > We sincerely thank the reviewer for their thoughtful feedback and continued support of our work. We greatly appreciate the positive assessment and the constructive comments provided throughout the review process.

---

### Decision · Program_Chairs · 2026-04-30

**Decision:**

Accept (regular)

**Comment:**

Two reviewers recommend Accept, and two reviewers recommend Weak Accept. The main concern raised by the reviewers is that the algorithm does not sample from the filtering distribution, yet this is true of many other similar algorithms. The authors are asked to further discuss this point of concern in the next version of the paper.